# Almost Minimax Optimal Best Arm Identification in Piecewise Stationary Linear Bandits

**Yunlong Hou**
Department of Mathematics
National University of Singapore
`yhou@u.nus.edu`

**Vincent Y. F. Tan**
Department of Mathematics
Department of Electrical and Computer Engineering
National University of Singapore
`vtan@nus.edu.sg`

**Zixin Zhong**
Data Science and Analytics Thrust
Hong Kong University of Science and Technology (Guangzhou)
`zixinzhong@hkust-gz.edu.cn`

## Abstract

We propose a *novel* piecewise stationary linear bandit (PSLB) model, where the environment randomly samples a context from an unknown probability distribution at each changepoint, and the quality of an arm is measured by its return averaged over all contexts. The contexts and their distribution, as well as the changepoints are unknown to the agent. We design *Piecewise-Stationary $\varepsilon$-Best Arm Identification$^+$* (PS$\varepsilon$BAI$^+$), an algorithm that is guaranteed to identify an $\varepsilon$-optimal arm with probability $\geq 1 - \delta$ and with a minimal number of samples. PS$\varepsilon$BAI$^+$ consists of two subroutines, PS$\varepsilon$BAI and NAÏVE $\varepsilon$-BAI (N$\varepsilon$BAI), which are executed in parallel. PS$\varepsilon$BAI actively detects changepoints and aligns contexts to facilitate the arm identification process. When PS$\varepsilon$BAI and N$\varepsilon$BAI are utilized judiciously in parallel, PS$\varepsilon$BAI$^+$ is shown to have a finite expected sample complexity. By proving a lower bound, we show the expected sample complexity of PS$\varepsilon$BAI$^+$ is optimal up to a logarithmic factor. We compare PS$\varepsilon$BAI$^+$ to baseline algorithms using numerical experiments which demonstrate its efficiency. Both our analytical and numerical results corroborate that the efficacy of PS$\varepsilon$BAI$^+$ is due to the delicate change detection and context alignment procedures embedded in PS$\varepsilon$BAI.

## 1 Introduction

In stochastic *multi-armed bandits* (MABs), an agent interacts with the environment at each time step. The agent pulls an arm and observes the corresponding return provided by the environment. The classical MAB framework assumes a *stationary* environment where the expected return of each arm remains unchanged over time. However, we usually face ever-changing environments in real life. For instance, in investment option selection and portfolio management, fund managers want to select a subset of good candidate portfolios. However, the market may be bullish, bearish, or in some other state. The transition between these states can be well-modelled as being stochastic. We wish to select portfolios that yield the best long-term option under such a piecewise stationary environment. Further examples such as one based on agriculture in the face of stochastically changing weather patterns are discussed in detail in Appendix A. These motivate us to formulate and investigate a *piecewise stationary linear bandit* (PSLB) model.

Our PSLB model is equipped with an *arm set* $\mathcal{X}$, a *context set* $\Theta$ and a deterministic but unknown sequence of *changepoints* $\mathcal{C}$. At each changepoint, the environment samples a context $\theta \in \Theta$ from an

unknown probability distribution $P_\theta$, and the returns of arms may change when the context changes. The return of each arm under each context is determined by its feature $x \in \mathcal{X}$ and the context $\theta$. In particular, the expected return of an arm is the weighted sum $\mu_x = \mathbb{E}_{P_\theta}[x^\top \theta]$. While the sequence of changepoints, as well as the distribution and latent vectors of contexts are not known, the agent samples an arm and observes the corresponding return at each time step so that it can identify the best arm $\arg\max_x \mu_x$ up to some tolerance $\varepsilon$ with probability $\geq 1 - \delta$ and with as few samples as possible. The agent's behavior does not affect the sequence of contexts that is drawn from $P_\theta$.

**Main Contributions.** We are the *first* to study the fixed-confidence *best arm identification* (BAI) problem in *piecewise stationary bandits* (PSB). Given $\delta > 0$, we say the arm with the highest expected return $\mu^*$ is the *best*, and an arm is $\varepsilon$-optimal if its expected return is at least $\mu^* - \varepsilon$. We seek to design an $(\varepsilon, \delta)$-PAC algorithm which can identify an $\varepsilon$-optimal arm with probability $\geq 1 - \delta$ in as few time steps as possible, i.e., with minimal sample complexity.

Our **first** contribution concerns the formulation of a *novel* PSLB model, where we measure the quality of an arm $x$ according to its *expected return* $\mu_x = \mathbb{E}_{\theta \sim P_\theta}[x^\top \theta]$ for the following reasons. Consider that an arm is measured by its average return across time, which is a generalization of the definition in *stationary bandits* (SB). A notable feature of PSB models is that the context changes as time evolves, and hence the arm's average return across time also changes, in general. Hence, we aim to identify an arm whose *average return across contexts* is high, and benefits the agent for interacting with the environment *in the long run* after the arm identification task. We are thus inspired to introduce the distribution of contexts under the PSLB model, define the expected return $\mu_x$ for each arm, and use this ensemble (non-time varying) statistic as the benchmark for what we seek to learn. The BAI task using this statistic is meaningful but challenging, as the agent needs to reliably estimate the context vectors, changepoints, and context distribution.

**Secondly,** we propose PIECEWISE-STATIONARY $\varepsilon$-BAI$^+$ (PS$\varepsilon$BAI$^+$), an algorithm designed to tackle the BAI problem in our PSLB model. We prove that it is $(\varepsilon, \delta)$-PAC and bound its sample complexity. PS$\varepsilon$BAI$^+$ samples arms according to a suitably defined G-optimal allocation, and runs two algorithms: NAÏVE $\varepsilon$-BAI (N$\varepsilon$BAI) and PS$\varepsilon$BAI as subroutines in parallel to achieve efficiency and attain a bound on the sample complexity in expectation.

• Being a baseline but naïve algorithm, the complexity of N$\varepsilon$BAI grows linearly with the maximum length between two changepoints $L_{\max}$, motivating us to design a more efficient algorithm, PS$\varepsilon$BAI, to reduce the impact of $L_{\max}$.

• PS$\varepsilon$BAI is equipped with two delicately designed subroutines LINEAR-CHANGE DETECTION (LCD) and LINEAR-CONTEXT ALIGNMENT (LCA) to actively detect changepoints and align contexts with those observed in the previous time steps respectively. Concretely, **in terms of the design**, PS$\varepsilon$BAI determines whether samples from two intervals are under the identical context via a sliding window mechanism, and detects changepoints and aligns contexts accordingly; this facilitates the estimation of context vectors and their distribution. Combining these elements into the design of PS$\varepsilon$BAI and analyzing them requires some care. **On the theoretical side**, we prove PS$\varepsilon$BAI identifies an $\varepsilon$-optimal arm faster than N$\varepsilon$BAI with high probability. The success of PS$\varepsilon$BAI relies on the LCD and LCA subroutines, while a minor drawback is that they require a non-vanishing failure probability budget which does not allow us to bound the sample complexity of PS$\varepsilon$BAI in expectation. To achieve a complete theoretical understanding, we delicately design the PS$\varepsilon$BAI$^+$ algorithm whose efficiency is inherited via the LCD and LCA procedures in PS$\varepsilon$BAI as well as the effective utilization of running PS$\varepsilon$BAI and N$\varepsilon$BAI in parallel.

**Thirdly,** we derive a lower bound on the complexity of any $(\varepsilon, \delta)$-PAC algorithm in PSLB models. To derive this bound, we first lower bound the complexity of an algorithm when the contextual information (and changepoints) are available, and then quantify the number of arm samples (and realized contexts) required to reliably infer an $\varepsilon$-best arm. We compare the upper bound of PS$\varepsilon$BAI$^+$ and this generic lower bound in several instances. The matching (up to logarithmic terms) of bounds illustrate that our PS$\varepsilon$BAI$^+$ algorithm is almost asymptotically optimal.

**Lastly,** we demonstrate the efficiency of PS$\varepsilon$BAI$^+$ with numerical experiments. The first half of our experiment shows that PS$\varepsilon$BAI$^+$ is $(\varepsilon, \delta)$-PAC and with significantly lower sample complexity compared to N$\varepsilon$BAI, corroborating our theoretical findings. In the second half, we compare PS$\varepsilon$BAI$^+$ to N$\varepsilon$BAI, and two other benchmarks DISTRIBUTION $\varepsilon$-BAI (D$\varepsilon$BAI) and D$\varepsilon$BAI$_\beta$. While contexts and changepoints are not available to PS$\varepsilon$BAI$^+$ and N$\varepsilon$BAI, they are observed by D$\varepsilon$BAI and D$\varepsilon$BAI$_\beta$. Nevertheless, PS$\varepsilon$BAI$^+$ is still competitive compared to D$\varepsilon$BAI and D$\varepsilon$BAI$_\beta$ in our

empirical experiments. Hence, both experiments justify the necessity of the change detection and context alignment procedures for boosting the learning of contexts and their distributions, as well as the identification of the best arm. We also show empirically that misspecifications to the knowledge of $L_{\min}$ and $L_{\max}$ do not affect the performance of our algorithm significantly.

**Related work.** *Best arm identification* (BAI) and *regret minimization* (RM) are two fundamental problems in multi-armed bandits. In stationary linear bandits, [1, 2] focus on BAI and [3, 4] aim to solve the RM problem. An efficient algorithm can choose the G-optimal allocation or $\mathcal{XY}$-allocation rule to quickly identify a good arm [1, 5]. Besides, [6, 7] focus on $\varepsilon$-optimal arm identification, which is a generalization of the standard BAI problem.

The BAI and RM problems are also studied in thr *non-stationary bandits* (NSB), where in contrast to the SB model, the context varies with time [8–10]. NSB can be largely divided into two classes: the drifting bandit (DB) model, where the context changes at each step [8], and PSB, where the context changes less frequently [10]. [11] provides an extensive discussion on the definition of NSB models.

On one hand, the RM problem have been investigated extensively in DB models [12, 13]. On the other hand, concerning the BAI task in DB models, [14] investigated BAI with a *fixed-horizon*, where the best arm has the highest average return over this horizon; [15] assumes the best arm remains unchanged after certain time step and explores the *fixed-confidence* setting. Besides, when the contextual information in NSB models is available, NSB models are known as *contextual bandit* (CB) models [16–18]. [16] showed that the contextual information accelerates the best arm identification process. More discussions on DB and CB models are presented in Appendix C.

Moreover, the context can drift dramatically in a DB model while it remains unvarying in a SB model. Straddling between the DB and SB models, PSB models assume there is an interim stationary interval between each two consecutive changepoints, where the context remains unchanged. The context changes can be characterized in different ways and affect the performance of proposed algorithms in PSB models. For instance, a changepoint signals the return of at least one arm shift as in [10], and indicates the best arm changes as in [19].

In PSB models, a large body of works focus on RM. While [10, 20–22] equip their algorithms with changepoint detection techniques to handle the context changes, [23] actively checks the quality of each arm. However, there is no existing work on the fixed-confidence BAI problem in a PSB model. To fill this gap in the literature, we design PS$\varepsilon$BAI$^+$ for $\varepsilon$-optimal arm identification in our proposed PSLB model. We show PS$\varepsilon$BAI$^+$ is almost asymptotically optimal by comparing its complexity to a generic lower bound of all algorithms, and validate its efficiency through numerical experiments.

## 2 Problem Setup

For $m \in \mathbb{N}$, let $[m] := \{1, 2, \ldots, m\}$. For a finite set $S$, let $\mathbf{\Delta}_S$ denote the $|S|$-dim probability simplex on $S$. Let $A(\mathbf{q}) := \sum_{x \in \mathcal{X}} q_x x x^\top$ be the matrix induced by $\mathbf{q} \in \mathbf{\Delta}_{\mathcal{X}}$ with $\mathcal{X} \subset \mathbb{R}^d$. An instance of *piecewise stationary linear bandit* is a tuple $\Lambda = (\mathcal{X}, \Theta, P_\theta, \mathcal{C})$. Specifically, $x \in \mathbb{R}^d$ is an arm (vector) and the arm set $\mathcal{X} \subset \mathbb{R}^d$ is composed of $|\mathcal{X}| = K$ arms that spans $\mathbb{R}^d$. The latent vector matrix $\Theta = (\theta_1^*, \ldots, \theta_N^*) \in \mathbb{R}^{d \times N}$ contains $N$ latent column vectors where the $j^{\text{th}}$ column $\theta_j^*$ is associated with context $j \in [N]$. For the sake of normalization, we assume $|x^\top \theta_j^*| \leq 1$ for all $x \in \mathcal{X}, j \in [N]$. Let $P_\theta$ denote the distribution (probability mass function) of the latent vectors and $p_j = P_\theta[\theta_j^*]$. We represent the probabilities of latent vectors as $\mathbf{p} = (p_1, \ldots, p_N) \in \Delta_N$. The fixed but unknown sequence of changepoints $\mathcal{C} := (c_1, c_2, \ldots)$ is an sequence of increasing positive integers $1 = c_1 < c_2 < \ldots$, characterizing all the changepoints (time steps).

The *return* of arm $x$ under latent context $j$ is a random variable $Y = x^\top \theta_j^* + \eta$, where $\eta$ is a zero-mean random variable (noise) supported on $[-1, 1]$, and the *expected return* of arm $x$ is $\mu_x := \mathbb{E}_{\theta \sim P_\theta}[x^\top \theta] = \sum_{j=1}^N P_\theta[\theta_j^*] x^\top \theta_j^*$. The *best arm*, which we assume is unique, is denoted as $x^* := \arg\max_{x \in \mathcal{X}} \mu_x$ with mean $\mu^* := \mu_{x^*}$. Given a slackness parameter $\varepsilon > 0$, we define the set of $\varepsilon$-*best arms* $\mathcal{X}_\varepsilon := \{x \in \mathcal{X} : \mu_x \geq \mu^* - \varepsilon\}$. For each pair of arms $(x, \tilde{x}) \in \mathcal{X}^2$, define the *contextual mean gap* between $x$ and $\tilde{x}$ under latent context $j$ as $\Delta_j(x, \tilde{x}) := (x - \tilde{x})^\top \theta_j^*$ and the *mean gap* between $x$ and $\tilde{x}$ as $\Delta(x, \tilde{x}) := \mu_x - \mu_{\tilde{x}}$.

Given $l \in \mathbb{N}$, the interval $(c_l, \ldots, c_{l+1} - 1)$ is known as the $l^{\text{th}}$ *stationary segment* and its length is $L_l := c_{l+1} - c_l$. We assume $L_{\min} \leq L_l \leq L_{\max}$. Let $l_t := \max\{l : c_l \leq t\}$ denote the number of

stationary segments up to time step $t$. At time step $t \in [T]$ (see Dynamics 1),

**(i)** If $t \in \mathcal{C}$, the environment samples a latent vector $\theta_{j_t}^*$ according to $P_\theta$, that is, it generates a (latent) *context sample* with $P_\theta$; otherwise the latent vector remains unchanged, i.e., $\theta_{j_t}^* = \theta_{j_{t-1}}^*$. The contexts $\{\theta_{j_{c_l}}^*\}_{l \in \mathbb{N}}$ are sampled i.i.d. from $P_\theta$ at each changepoint $\{c_l\}_{l \in \mathbb{N}}$.

**(ii)** The agent pulls an arm $x_t$ and observes the stochastic return $Y_{t,x_t} = x_t^\top \theta_{j_t}^* + \eta_t$, where $\eta_t$ is drawn independently from a distribution supported on $[-1, 1]$.

The agent uses an *online* algorithm $\pi := \{(\pi_t, \tau_t, r_t)\}_{t \in \mathbb{N}}$ to actively interact with the instance $\Lambda$ and only has access to the arm set $\mathcal{X}$, number of latent vectors $N$, the bounds on the length of each segment $L_{\min}$ and $L_{\max}$, the slackness parameter $\varepsilon$, and the confidence parameter $\delta$.

• sampling rule $\pi_t : \mathcal{H}_t^\pi := \left((x_s^\pi, Y_{s,x_s^\pi})\right)_{s \in [t-1]} \to \mathcal{X}$ samples an arm $x_t^\pi$ based on the observation history $\mathcal{H}_t^\pi$ and observe the corresponding random return $Y_{t,x_t^\pi}$;

• stopping rule $\tau_t : \mathcal{H}_{t+1}^\pi \to \{\text{STOP}, \text{CONTINUE}\}$ decides whether to stop or continue to execute given the observation history $\mathcal{H}_{t+1}^\pi$. The stopping time under algorithm $\pi$ is denoted as $\tau^\pi$;

• recommendation rule $r_\tau : \mathcal{H}_{\tau+1}^\pi \to \mathcal{X}$ recommends an arm $\hat{x}^\pi$ based on $\mathcal{H}_{\tau+1}^\pi$ upon termination. The stopping time $\tau^\pi$ is the *sample complexity* of the algorithm $\pi$ under instance $\Lambda$. The *expected sample complexity* is $\mathbb{E}[\tau^\pi]$, where the expectation is taken w.r.t. the random returns, the realization of the contexts governed by the latent vector distribution $P_\theta$, and the randomness of the algorithm $\pi$.

An algorithm $\pi$ is $(\varepsilon, \delta)$-*PAC* (Probably Approximately-Correct) if
$$\mathbb{P}[\hat{x}^\pi \in \mathcal{X}_\varepsilon] \geq 1 - \delta.$$
Our overarching goal in this paper is to devise an $(\varepsilon, \delta)$-PAC algorithm that minimizes $\tau^\pi$ with high probability (w.h.p.) and in expectation.

## 3 Algorithms

### 3.1 A Naïve Baseline: N$\varepsilon$BAI

We first devise the N̲AÏVE $\varepsilon$-B̲EST A̲RM I̲DENTIFICATION (or N$\varepsilon$BAI) algorithm (presented in Algorithm 2). In the design of N$\varepsilon$BAI, only the choice of confidence radius $\tilde{\rho}_t$ takes the potential context changes into consideration. Even though N$\varepsilon$BAI does not attempt to detect potential changes in the context, it can identify an $\varepsilon$-optimal arm w.h.p. and is with a finite expected sample complexity.

**Proposition 3.1.** *Let* $\Delta_{\min} = \min_{x \neq x^*} \Delta(x^*, x)$,
$$T_V^N = \frac{d}{(\varepsilon + \Delta_{\min})^2} \ln \frac{1}{\delta} \qquad and \qquad T_D^N = \frac{L_{\max}}{(\varepsilon + \Delta_{\min})^2} \ln \frac{1}{\delta}.$$
*The N$\varepsilon$BAI algorithm is* $(\varepsilon, \delta)$-*PAC and its expected sample complexity is* $\tilde{O}(T_V^N + T_D^N)$.

The upper bound in Proposition 3.1 (also see Appendix F) consists of two main terms. (i) As N$\varepsilon$BAI samples arms according to the G-optimal allocation (see Appendix D), the amount of samples needed to estimate the average of latent vectors $\sum_{s=1}^t \theta_{j_s}^*/t$ contributes to $T_V^N$. (ii) $T_D^N$ quantifies how fast $\sum_{s=1}^t \theta_{j_s}^*/t$ converges to the expectation of context vectors $\sum_{j=1}^t p_j \theta_j^*$.

The sample complexity of N$\varepsilon$BAI grows linearly with $L_{\max}$, but we surmise that the sample complexity of a close-to-optimal algorithm should have a *reduced dependence on* $L_{\max}$.

### 3.2 P̲iecewise-S̲tationary $\varepsilon$-B̲est A̲rm I̲dentification

The algorithm P̲IECEWISE-S̲TATIONARY $\varepsilon$-B̲EST A̲RM I̲DENTIFICATION (or PS$\varepsilon$BAI) is presented in Algorithm 1. By using a sliding window mechanism, PS$\varepsilon$BAI actively detects the change-points and aligns the current latent context with contexts observed in the previous time steps via L̲INEAR-C̲HANGE D̲ETECTION (or LCD) and L̲INEAR-C̲ONTEXT A̲LIGNMENT (or LCA), which are presented in Algorithms 3 and 4 (see App. D.2.2), respectively. PS$\varepsilon$BAI consists of three phases:

(i) *Exploration phase* (E̲xp): Estimate latent vectors and their distribution $P_\theta$ (Lines 8 to 11 and 25);

(ii) *Change Detection phase* (C̲D): Detect changepoints (Lines 12 to 16);

(iii) *Context Alignment phase* (C̲A): Evaluate the current context and align it with the contexts observed in previous time steps (Lines 17 to 21).

At time $t$, we estimate $\Theta$ and $\mathbf{p}$ with $\hat{\Theta}_t = (\hat{\theta}_{t,1}, \ldots, \hat{\theta}_{t,N})$ and $\hat{\mathbf{p}}_t = (\hat{p}_{t,1}, \ldots, \hat{p}_{t,N})^\top$, respectively.[1] We denote the empirical mean gap between $x$ and $\tilde{x}$ under context $j$ as $\hat{\Delta}_{t,j}(x,\tilde{x}) := (x - \tilde{x})^\top \hat{\theta}_{t,j}$.

PS$\varepsilon$BAI first computes the G-optimal allocation [1] $\lambda^*$ on the arm set $\mathcal{X}$ and its maximum possible stopping time $\tau^*$ (Line 2). It initializes $\mathrm{CD}_{\mathrm{sample}}$ and $\mathrm{CA}_{\mathrm{id}}$. $\mathrm{CD}_{\mathrm{sample}}$ collects samples to detect changepoints and $\mathrm{CA}_{\mathrm{id}}$ maintains a dictionary of {latent context index : identification samples} pairs (Line 3);[2] $\mathrm{CA}_{\mathrm{id}}[j]$ is the sequence of CD samples used to *identify* latent context $j$. It also initializes $\mathcal{T}_{t,j}$, the collection of time indices in $[t]$ in the Exp phases under estimated context $j$. Define

$$\mathcal{T}_t = \bigcup_{j \in [N]} \mathcal{T}_{t,j}, \qquad T_t = |\mathcal{T}_t|, \qquad T_{t,j} = |\mathcal{T}_{t,j}|, \qquad \forall j \in [N]. \tag{3.1}$$

---

**Algorithm 1** Piecewise-Stationary $\varepsilon$-Best Arm Identification (PS$\varepsilon$BAI)

1: **Input:** arm set $\mathcal{X}$, size of the set of latent vectors $N$, bounds on the segment lengths $L_{\min}$ and $L_{\max}$, slackness parameter $\varepsilon$, confidence parameter $\delta$, sampling parameter $\gamma$ and window size $w$, threshold $b$.

2: **Initialize**: Compute the G-optimal allocation $\lambda^*$ and $\tau^* = \frac{38400 \ln(80) N L_{\max}}{\varepsilon^2} \ln \frac{N^2 K L_{\max}}{\delta \varepsilon^2}$.

3: Set $\mathrm{CD}_{\mathrm{sample}} = [\ ]$, $\mathrm{CA}_{\mathrm{id}} = \{\ \}$. Set $t_{\mathrm{CD}} = +\infty$.

4: Set $\mathcal{T}_{t,j} = \emptyset$ and initialize $\mathcal{T}_t$, $T_{t,j}$, $T_t$ with (3.1) for all $t \leq \tau^*$, $j \in [N]$.

5: Sample $\frac{w}{2}$ arms $\{x_s\}_{s=1}^{\frac{w}{2}} \sim \lambda^*$ and observe the associated returns $\{Y_{s,x_s}\}_{s=1}^{\frac{w}{2}}$, $t = \frac{w}{2}$, $t_{\mathrm{CA}} = \frac{w}{2}$.

6: $\mathrm{CA}_{\mathrm{id}} = \{1 : [(x_s, Y_{s,x_s})]_{s=1}^{\frac{w}{2}}\}$, $\hat{j}_t = 1$.

7: **while** $t \leq \tau^*$ **do**

8: $\quad t = t + 1$

9: $\quad$ Sample an arm $x_t \sim \lambda^*$ and observe return $Y_{t,x_t}$.

10: $\quad$ **if** $\mod(t - t_{\mathrm{CA}}, \gamma) \neq 0$ **then**

11: $\quad\quad$ Update $\hat{j}_t = \hat{j}_{t-1}$, $\mathcal{T}_{t,\hat{j}_t} = \mathcal{T}_{t-1,\hat{j}_t} \cup \{t\}$, $\mathcal{T}_{t,j} = \mathcal{T}_{t-1,j}$ for $j \neq \hat{j}_t$.

12: $\quad$ **else**

13: $\quad\quad \mathrm{CD}_{\mathrm{sample}} = \mathrm{CD}_{\mathrm{sample}} + [(x_t, Y_{t,x_t})]$.

14: $\quad\quad$ Update $\hat{j}_t = \hat{j}_{t-1}$, $\mathcal{T}_{t,j} = \mathcal{T}_{t-1,j}$ for all $j \in [N]$.

15: $\quad\quad$ **if** $|\mathrm{CD}_{\mathrm{sample}}| \geq w$ **then**

16: $\quad\quad\quad$ **if** $\mathrm{LCD}(\mathcal{X}, w, b, \mathrm{CD}_{\mathrm{sample}}[-w : ])$ **then**

17: $\quad\quad\quad\quad \mathrm{CD}_{\mathrm{sample}} = [\ ]$.

18: $\quad\quad\quad\quad t = t + \frac{w}{2}$, $t_{\mathrm{CA}} = t$, $t_{\mathrm{CD}} = +\infty$.

19: $\quad\quad\quad\quad \hat{j}_t, \mathrm{CA}_{\mathrm{id}} = \mathrm{LCA}(\mathcal{X}, w, b, \mathrm{CA}_{\mathrm{id}})$.

20: $\quad\quad\quad\quad$ **if** $\hat{j}_t = N + 1$ **then break**.

21: $\quad\quad\quad\quad$ Revert $\mathcal{T}_{t,j} = \mathcal{T}_{t - \frac{w(\gamma+1)}{2}, j}$ for all $j \in [N]$.

22: $\quad\quad\quad$ **end if**

23: $\quad\quad$ **end if**

24: $\quad$ **end if**

25: $\quad$ Update the estimates using (3.1), (3.2) and (3.3).

26: $\quad$ **if** Condition (3.4) is met and $t_{\mathrm{CD}} = +\infty$ **then**

27: $\quad\quad$ Record $\hat{x}_\varepsilon = \arg\max_{x \in \mathcal{X}} x^\top \hat{\Theta}_t \hat{\mathbf{p}}_t$.

28: $\quad\quad t_{\mathrm{CD}} = |\mathrm{CD}_{\mathrm{sample}}|$.

29: $\quad$ **else if** $t_{\mathrm{CD}} = |\mathrm{CD}_{\mathrm{sample}}| - \frac{w}{2}$ **then**

30: $\quad\quad$ Recommend arm $\hat{x}_\varepsilon$.

31: $\quad\quad$ **break**

32: $\quad$ **end if**

33: **end while**

---

It then collects $\frac{w}{2}$ samples and stores them in $\mathrm{CA}_{\mathrm{id}}$, which is then used to identify the first latent context (Lines 5 to 6).

In the Exp phase, PS$\varepsilon$BAI **firstly** samples an arm $x_t$ with $\lambda^*$ and observes the return $Y_{t,x_t} = x_t^\top \theta_{\hat{j}_t}^* + \eta_t$ (Line 9). It **then** updates the estimated context index and time collectors (Line 11). It also updates the estimates of value and probability of each context $j$ (Line 25) with

$$\hat{\theta}_{t,j} = \frac{1}{T_{t,j}} \sum_{s \in \mathcal{T}_{t,j}} A(\lambda^*)^{-1} x_s Y_{s,x_s} \qquad \text{and} \qquad \hat{p}_{t,j} = \frac{T_{t,j}}{T_t}, \tag{3.2}$$

---

[1] The empirical latent vector-probability pairs $[(\hat{\theta}_{t,j}, \hat{p}_{t,j})]_{j=1}^N$ can only approximate the unknown pairs $[(\theta_{\sigma(j)}^*, p_{\sigma(j)})]_{j=1}^N$ up to a permutation $\sigma : [N] \to [N]$, which is determined by the occurrence order of latent vectors. Thus we assume the latent vectors appear in order of increasing indices.

[2] A sample in $\mathrm{CD}_{\mathrm{sample}}$ is a CD sample; A dictionary has a pairing structure {key:value} and dictionary[key] = value. $[a_i]_{i=1}^n$ denotes a sequence of elements $a_1, \ldots, a_n$.

where $\hat{\theta}_{t,j} = \mathbf{0}$ if $T_{t,j} = 0$. We define $\alpha_t$, $\xi_t$, $\beta_{t,j}$, and $\hat{\Delta}_{t,j}^{\text{clip}_2}(x, \tilde{x})$ in Appendix D.2.1. For each pairs of arms $(x, \tilde{x})$, the confidence radius of $\Delta(x, \tilde{x})$ at time step $t$ is

$$\rho_t(x, \tilde{x}) := 2(\alpha_t + \xi_t) + \sum_{j=1}^{N} \beta_{t,j} |\hat{\Delta}_{t,j}^{\text{clip}_2}(x, \tilde{x}) + \zeta_t(x, \tilde{x})|; \tag{3.3}$$

PS$\varepsilon$BAI actively enters the CD phase every $\gamma$ time steps (Line 12). It **firstly** adds a CD sample to $\text{CD}_{\text{sample}}$ (Line 13). **Next**, if there are sufficient CD samples (Line 15), the LCD subroutine (presented in Algorithm 3) is called and utilizes the most recent $w$ CD samples to check whether a changepoint just occurred (Line 16). PS$\varepsilon$BAI steps into the CA phase if a changepoint is detected, and skips the CA phase otherwise, which is illustrated by Figures 1(b) and 1(a) respectively.

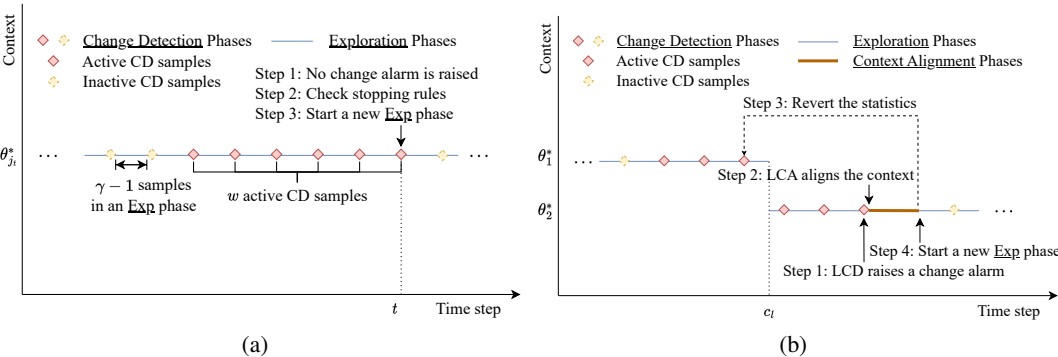

Figure 1: (a) No change alarm is raised during a stationary segment. The active CD samples are the input to the LCD subroutine at current time step $t$. (b) A changepoint is detected by LCD, followed by a CA phase and a statistics reversion step.

In the CA phase, PS$\varepsilon$BAI **starts** by resetting $\text{CD}_{\text{sample}}$ (Line 17), updating the count of time steps and recording the ending time of this CA phase (Line 18). **Thereafter**, the CA subroutine (presented in Algorithm 4) is invoked, which estimates the current latent context index $\hat{j}_t$ and updates $\text{CA}_{\text{id}}$ (Line 19). If $\hat{j}_t = N + 1$, i.e., PS$\varepsilon$BAI identifies $N + 1$ latent contexts, which is incorrect under instance $\Lambda$, it terminates and fails to identify an $\varepsilon$-optimal arm (Line 20). **Lastly**, all empirical statistics are reverted to those from $(w(\gamma + 1)/2)$ time steps ago, i.e., the most recent $(w\gamma/2)$ samples in the Exp phases are abandoned (Line 21).

The stopping rule is described in Lines 26 to 32. **(I)** If the following condition is satisfied (Line 26):

$$\min_{x:x \neq x_t^*} \hat{\Delta}_t(x_t^*, x) - \rho_t(x_t^*, x) \geq -\varepsilon \quad \text{and} \quad T_t \geq \frac{2L_{\max}}{9} \ln\left(\frac{2}{\delta_{d,T_t}}\right) \tag{3.4}$$

where the empirical mean gap $\hat{\Delta}_t(x_t^*, x) := (x_t^* - x)^\top \hat{\Theta}_t \hat{\mathbf{p}}_t$ and $x_t^* := \arg\max_{x \in \mathcal{X}} x^\top \hat{\Theta}_t \hat{\mathbf{p}}_t$, PS$\varepsilon$BAI records the arm with the highest empirical mean as $\hat{x}_\varepsilon$ and the number of CD samples $t_{\text{CD}}$ (Lines 27 and 28) but does not terminate immediately. Besides, a mild forced arm pull procedure is in the second line of (3.4), which is inspired by Lemma E.1 and to ensure the performance of PS$\varepsilon$BAI. **(II)** PS$\varepsilon$BAI will execute for another $(w\gamma/2)$ time steps in which $w/2$ CD samples are collected; if no changepoint is detected with these $w/2$ CD samples, the recorded arm $\hat{x}_\varepsilon$ is recommended and PS$\varepsilon$BAI terminates (Lines 29 to 30). Part **(II)** of the stopping rule assures PS$\varepsilon$BAI does not terminate when a changepoint has occurred but has not been detected, as PS$\varepsilon$BAI may fail to identify an $\varepsilon$-optimal arm otherwise.

We remark that even though PS$\varepsilon$BAI uses the knowledge of $L_{\max}$, our experiments show that the performance of PS$\varepsilon$BAI is robust to small misspecifications in $L_{\max}$ (see Appendix O.2). Furthermore, the computational complexity of PS$\varepsilon$BAI is computed in detail in Appendix D.4. The derived computational complexity indicates the proposed algorithm depends in a natural manner on the problem parameters such as $d, K, N$, and $\gamma$. Lastly, thanks to the LCD and LCA subroutines, a slightly modified variant of PS$\varepsilon$BAI can also solve the "$\varepsilon$-Best Arm Tuple identification problem", which aims to identify an $\varepsilon$-best arm *under each context*; see Appendix Q for details.

### 3.2.1 Theoretical guarantee of $\text{PS}\varepsilon\text{BAI}$

To facilitate the analysis of $\text{PS}\varepsilon\text{BAI}$, we propose the following assumptions. Note that our $\text{PS}\varepsilon\text{BAI}$ algorithm may still succeed to identify an $\varepsilon$-optimal arm w.h.p. when the assumptions do not hold.

**Assumption 1** (Distinguishability Condition). *The agent can choose $w$, $\gamma$ and $b$ such that (1) $2b \leq \Delta_c$ where $\Delta_c := \min_{\theta_j^* \neq \theta_{\tilde{j}}^*} \max_{x \in \mathcal{X}} |x^\top(\theta_j^* - \theta_{\tilde{j}}^*)|$ is the minimum gap between two contexts; and (2) $3w\gamma \leq L_{\min}$. A possible choice is*

$$b = \frac{8d}{3w}\ln\frac{2}{\delta_{\text{FAE}}} + \sqrt{\left(\frac{8d}{3w}\ln\frac{2}{\delta_{\text{FAE}}}\right)^2 + \frac{24d}{w}\ln\frac{2}{\delta_{\text{FAE}}}} \quad \text{where} \quad \delta_{\text{FAE}} = \frac{\gamma\delta}{4(\tau^*)^2 K}. \quad (3.5)$$

This assumption guarantees (i) $\text{PS}\varepsilon\text{BAI}$ will not abandon all samples during the reversion procedure (Line 21 of Algorithm 1); (ii) each two latent vectors can be distinguished if the window size $w$ is sufficiently large (e.g., $L_{\min}/6$). We clarify that this assumption is only for the rigor of theoretical guarantees and it holds provided that each stationary segment is sufficiently long; this is a feature of PSB models and similar assumptions are also present in existing works for their analyses [21, 10, 22]. We demonstrate the robustness of $\text{PS}\varepsilon\text{BAI}$ to these parameters using experiments in Section 6.

**Theorem 3.2.** *Define the context distribution estimation (DE) hardness parameter*

$$\text{H}_{\text{DE}}(x_\varepsilon, x) := \frac{L_{\max}}{(\Delta(x^*, x) + \varepsilon)^2}\bar{H}(x_\varepsilon, x)$$

*where $\bar{H}(x_\varepsilon, x) := \left(\sum_{j=1}^N \sqrt{\min\{16p_j, 1/4\}}|\Delta_j(x_\varepsilon, x) + \varepsilon|\right)^2$. Under Assumption 1, with probability at least $1 - \delta$, $\text{PS}\varepsilon\text{BAI}$ identifies an $\varepsilon$-optimal arm and its sample complexity is*

$$\tilde{O}\left(\max_{\substack{x_\varepsilon \in \mathcal{X}_\varepsilon \\ x \neq x_\varepsilon, x^*}} \underbrace{\frac{d}{(\Delta(x^*, x) + \varepsilon)^2}\ln\frac{1}{\delta}}_{T_\text{V}(x)} + \underbrace{\text{H}_{\text{DE}}(x_\varepsilon, x)\ln\frac{1}{\delta}}_{T_\text{D}(x_\varepsilon, x)} + \underbrace{\frac{NL_{\max}}{\Delta(x^*, x) + \varepsilon}\ln\frac{1}{\delta}}_{T_\text{R}(x)}\right). \quad (3.6)$$

The upper bound comprises three terms which serve distinct purposes:

(i) *Latent vector estimation* (VE): $\tilde{O}(T_\text{V}(x))$ quantifies the bulk of samples needed to obtain a good estimate of latent context vectors such that the returns of $x_\varepsilon$ and $x$ can be distinguished, where $x_\varepsilon$ is an $\varepsilon$-best arm and $x \notin \mathcal{X}_\varepsilon$ is a suboptimal arm. $T_\text{V}(x)$ recovers the sample complexity in the stationary linear bandits in [1], indicating that $\text{PS}\varepsilon\text{BAI}$ estimates latent vectors efficiently.

(ii) *Context distribution estimation* (DE): $\tilde{O}(T_\text{D}(x_\varepsilon, x))$ characterizes the bulk of samples needed to learn the distribution of latent context vectors.

(iii) *Residual estimation* (RE): $\tilde{O}(T_\text{R}(x))$ counts the remaining samples needed for VE and DE, in addition to $\tilde{O}(T_\text{V} + T_\text{D})$.

Besides, the max operator is applied to exclude all suboptimal arms. We also see that $T_\text{V}(x)$ and $T_\text{D}(x_\varepsilon, x)$ are similar to $T_\text{V}^\text{N}$ and $T_\text{D}^\text{N}$ in Proposition 3.1 respectively.

**Firstly,** the bound in (3.6) implies that, in an instance with smaller *relaxed mean gap* $\Delta(x^*, x) + \varepsilon$, $\text{PS}\varepsilon\text{BAI}$ terminates after a larger number of time steps; in other words, it is more difficult to identify an $\varepsilon$-optimal arm. In difficult instances with small $\Delta(x^*, x) + \varepsilon$, the different orders of this term in $T_\text{V}(x)$, $T_\text{D}(x_\varepsilon, x)$ and $T_\text{R}(x)$ indicate that, $T_\text{R}(x)$ is small compared to $T_\text{V}(x)$ and $T_\text{D}(x_\varepsilon, x)$.

**Secondly,** DE solely utilizes context samples generated with $P_\theta$ and they are generated *only* at changepoints in $\mathcal{C}$, while all the observations in Exp phases facilitate VE. From this perspective, there are less samples that can be used for DE than for VE as $\text{PS}\varepsilon\text{BAI}$ processes, and hence $T_\text{D}(x_\varepsilon, x)$ is supposed to be with larger order than $T_\text{V}(x)$.

**Moreover,** for the purpose of DE, $\text{PS}\varepsilon\text{BAI}$ needs to observe context samples at $\tilde{O}\left(\frac{\bar{H}(x_\varepsilon, x)}{(\Delta(x^*, x) + \varepsilon)^2}\ln\frac{1}{\delta}\right) = \tilde{O}\left(\frac{\text{H}_{\text{DE}}(x_\varepsilon, x)}{L_{\max}}\ln\frac{1}{\delta}\right)$ changepoints where $L_{\max}$ is the maximum length of a stationary segment, leading us to $T_\text{D}(x_\varepsilon, x)$. Close examination of the definition of $\bar{H}(x_\varepsilon, x)$ reveals that *both* the vectors and their probabilities influence the number of samples needed for DE. The comparison between $T_\text{D}(x_\varepsilon, x)$ and $T_\text{D}^\text{N}$ in Proposition 3.1 clearly indicates that $\text{PS}\varepsilon\text{BAI}$ mitigates the influence of $L_{\max}$ by detecting changepoints and aligning the detected context with observed ones, while $\text{N}\varepsilon\text{BAI}$ does not do so.

### 3.3 PS$\varepsilon$BAI$^+$ = PS$\varepsilon$BAI $\cup$ N$\varepsilon$BAI

We have provided a *high-probability* result for PS$\varepsilon$BAI in Theorem 3.2. The design of PS$\varepsilon$BAI (Line 7 of Algorithm 1) indicates that PS$\varepsilon$BAI will not recommend any arm if it does not terminate at time $\tau^*$. This result is nontrivial, as the high-probability result in Theorem 3.2 depends on the success of change detection (Algorithm 3) and context alignment (Algorithm 4), which requires a non-vanishing failure probability (e.g., $\delta/2$). Thus, we cannot derive an upper bound on the expected sample complexity of PS$\varepsilon$BAI. We devise a solution by designing the Piecewise-Stationary $\varepsilon$-Best Arm Identification$^+$ (PS$\varepsilon$BAI$^+$) algorithm with a simple but effective trick.

The PS$\varepsilon$BAI$^+$ algorithm samples *one* arm with the G-optimal allocation $\lambda^*$ at each time step, with which Algorithms 1 and 2 are executed in parallel (detailed in Algorithm 5). This is feasible since PS$\varepsilon$BAI and N$\varepsilon$BAI algorithms have the same sampling rule.

**Theorem 3.3.** *The* PS$\varepsilon$BAI$^+$ *algorithm is* $(\varepsilon, \delta)$-*PAC and its expected sample complexity is*

$$\tilde{O}\left( \min \left\{ \max_{x_\varepsilon \in \mathcal{X}_\varepsilon, x \neq x_\varepsilon, x^*} T_V(x) + T_D(x_\varepsilon, x) + T_R(x), \ T_V^N + T_D^N \right\} \right).$$

PS$\varepsilon$BAI$^+$ inherits the superiority of PS$\varepsilon$BAI to adapt to the piecewise stationary environment, and employs the stopping rule of N$\varepsilon$BAI to maintain a finite expected sample complexity. As a result, the expected complexity of PS$\varepsilon$BAI$^+$ in Theorem 3.3 is of the same order as the high-probability one of PS$\varepsilon$BAI in Theorem 3.2 and is not larger than the complexity of N$\varepsilon$BAI in Proposition 3.1. We show how our results particularize to the *stationary* linear bandits BAI problem, as well as additional discussions on the upper bound, in Appendix P.

## 4 Lower Bound on the Sample Complexity

Given $\Lambda = (\mathcal{X}, \Theta, P_\theta, \mathcal{C})$, define the alternative instance $\Lambda' = (\mathcal{X}, \Theta', P_{\theta'}, \mathcal{C})$ w.r.t. $\Lambda$, where $\Theta' = (\theta_1', \ldots, \theta_n') \in \mathbb{R}^{d \times N}$, $P_{\theta'}[\theta_j'] = P_\theta[\theta_j^*]$, and there exists $x \in \mathcal{X} \setminus \mathcal{X}_\varepsilon$, such that $x_\varepsilon^\top \mathbb{E}_{\theta' \sim P_{\theta'}}[\theta'] < x^\top \mathbb{E}_{\theta' \sim P_{\theta'}}[\theta'] - \epsilon$ for all $x_\varepsilon \in \mathcal{X}_\varepsilon$. Let $\mathrm{Alt}_\Theta(\Lambda)$ be set of all alternative instances (w.r.t. $\Lambda$).

**Theorem 4.1.** *For all* $(\varepsilon, \delta)$-*PAC algorithm* $\pi$, *there exists an instance* $\Lambda = (\mathcal{X}, \Theta, P_\theta, \mathcal{C})$ *such that*

$$\mathbb{E}[\tau_\pi] \geq \max \left\{ T_\varepsilon(\Lambda) \ln \frac{1}{2.4\delta}, \ c_{N_\mathcal{C}} \right\},$$

*where*

$$T_\varepsilon(\Lambda)^{-1} := \max_{\{v_j\}_{j=1}^N} \min_{\Lambda' \in \mathrm{Alt}_\Theta(\Lambda)} \sum_{j,x} p_j v_{j,x} \frac{(x^\top(\theta_j^* - \theta_j'))^2}{2}, \quad and$$

$$N_\mathcal{C} := \max_{x \neq x^*} \frac{\sum_j p_j (\Delta_j(x^*, x) + \varepsilon)^2}{(\Delta(x^*, x) + \varepsilon)^2} \ln \frac{1}{4\delta}.$$

Recall that $c_{N_\mathcal{C}}$ is the $N_\mathcal{C}$-th changepoint in the changepoint sequence $\mathcal{C}$, which is lower bounded by $N_\mathcal{C} L_{\min}$ and is $N_\mathcal{C} L_{\max}$ in the worst case. To derive the lower bound in Theorem 4.1, we investigate two environments different from the one defined in Section 2 (and as in Dynamics 1):

• Dynamics 2: the agent observes the index of current context $j_t$ (i.e., contextual linear bandits);

• Dynamics 3: the agent observes the changepoints in $\mathcal{C}$ and context vector $\theta_{j_t}^*$'s, and hence she solely needs to estimate the distribution of contexts.

We bound the sample complexity of an $(\varepsilon, \delta)$-BAI algorithm in Dynamics 2 and 3 respectively, which when combined, yield the lower bound in Theorem 4.1; this is detailed in Appendix M.

Note that $T_\varepsilon(\Lambda)^{-1}$ in the lower bound generalizes [16] to the setting of linear bandits. In addition, Theorem 4.1 can be reduced to a bound in stationary linear bandits with one latent vector [24] (see the discussion leading to (M.15)).

## 5 On the Asymptotic Optimality of PS$\varepsilon$BAI$^+$

To illustrate the efficiency of our PS$\varepsilon$BAI$^+$ algorithm, we compare the upper bound on its expected sample complexity in Theorem 3.3 and the generic lower bound in Theorem 4.1 under specific instances below and in Appendix N. We also gain further insight into our PS$\varepsilon$BAI$^+$ algorithm.

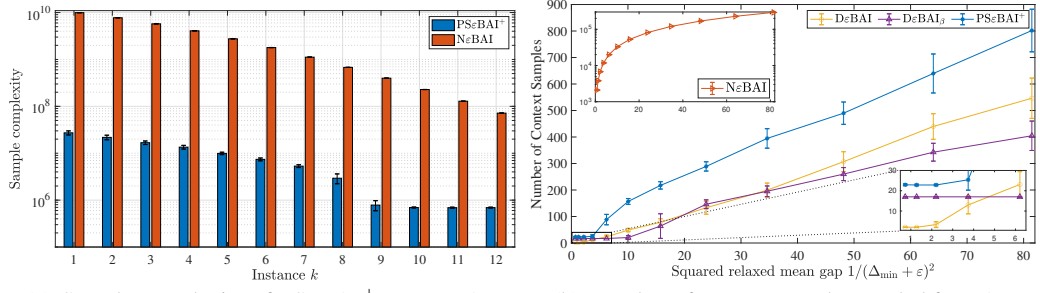

Figure 2: Experimental results

**Example 1.** *Instance $\Lambda = (\mathcal{X}, \Theta, P_\theta, \mathcal{C})$ is with (i) $2d-1$ arms: $x_{(1)} = \mathbf{e}_1, x_{(i)} = \mathbf{e}_i, x_{(d+i-1)} = \mathbf{e}_1 \cos\phi + \mathbf{e}_i \sin\phi$ for all $i \in \{2, \ldots, d\}$ where $\phi \in [0, \pi/4)$, (ii) $2d-2$ contexts: $\theta_{j\pm}^* = \mathbf{e}_1 \cos\phi \pm \mathbf{e}_{j+1} \sin\phi$ for all $j \in [d-1]$, (iii) Context distribution: $p_j = 1/N$ for all $j \in [N]$.*

Under the instance defined in Example 1, $x_{(i)}$ for all $i \neq 1$ is inferior to $x_{(1)}$ under all contexts and $x_{(i+d)}$ for all $i \in [d-1]$ is marginally better than $x_{(1)}$ by $1 - \cos\phi$ only under context $\theta_{i+}^*$ and $\Delta(x_{(1)}, x_{(i+d)}) = \cos\phi - \cos^2\phi$. We expect PS$\varepsilon$BAI$^+$ to discover this feature of the instances and quickly identify an $\varepsilon$-optimal arm with a course estimation of the context distribution.

**Corollary 5.1.** *For the instance defined in Example 1, we have $\mathrm{H}_{\mathrm{DE}}(x_\varepsilon, x) = \tilde{O}(N L_{\max})$ for all $(x_\varepsilon, x) \in \mathcal{X}_\varepsilon \times (\mathcal{X} \setminus \mathcal{X}_\varepsilon)$. In addition, if $\varepsilon < (\cos\phi)(1 - \cos\phi)$, we have*

$$\frac{\mathbb{E}[\tau]^*}{\ln(1/\delta)} \in \tilde{\Theta}\left( (1 + f(\phi)) \cdot \frac{d}{(\Delta_{x_{(1)}, x_{(d+1)}} + \varepsilon)^2} \right), \tag{5.1}$$

*where $\mathbb{E}[\tau]^*$ is the minimal expected sample complexity over all $(\varepsilon, \delta)$-PAC algorithms and $f : \mathbb{R} \to \mathbb{R}$ satisfies $f(\phi) \to 0$ as $\phi \to 0^+$. The upper bound in (5.1) is achieved by PS$\varepsilon$BAI$^+$.*

The order of $\mathrm{H}_{\mathrm{DE}}$ in Corollary 5.1 indicates that $T_{\mathrm{D}}(x_{(1)}, x_{(d+1)}) = \tilde{O}(N L_{\max} \ln(1/\delta))$ is not dominating the sample complexity of PS$\varepsilon$BAI$^+$, suggesting that a coarse estimation of the context distribution is sufficient when $\phi$ is small. In other words, PS$\varepsilon$BAI$^+$ exploits the feature of instances and utilize samples mostly for estimate context vectors, which is again expected.

Corollary 5.1 implies that under such instances, the upper and lower bounds on the sample complexity of PS$\varepsilon$BAI$^+$ match up to logarithmic factors, that is, the performance PS$\varepsilon$BAI$^+$ is near optimal. Besides, the bound of N$\varepsilon$BAI in Proposition 3.1 is with an extra additive term $L_{\max}$ compared to the lower bound in Corollary 5.1, illustrating that N$\varepsilon$BAI is suboptimal and again emphasizing the significance of detecting changes and aligning contexts for PS$\varepsilon$BAI$^+$ to reduce the impact of $L_{\max}$.

## 6  Numerical Experiments

We now evaluate the empirical performance of PS$\varepsilon$BAI$^+$. We utilize the instance defined in Example 1 with $d = 2$, $\phi = \pi/8$, We generate a changepoint sequence $\mathcal{C}$ such that $c_{l+1} = c_l + L_l$ with $L_{\min} = 3 \times 10^4$, $L_{\max} = 5 \times 10^4$, $\mathbb{P}[L_l = L_{\min}] = 0.8$, $\mathbb{P}[L_l = L_{\max}] = 0.2$, and fix it throughout the whole set of experiments. We set the confidence parameter $\delta = 0.05$ and vary the slackness parameter $\varepsilon$ from 0.04 to 0.6 (i.e., $\varepsilon = 0.03 \times 1.35^k$ for $k \in [12]$). We set $\gamma = 6$, the window size $w = L_{\min}/(3\gamma)$ and compute $b$ via (3.5) in Assumption 1.[3] For each choice of algorithm and instance, we run 20 independent trials. All the code to reproduce our experiments can be found at https://github.com/Y-Hou/BAI-in-PSLB.git.

We **first** compare PS$\varepsilon$BAI$^+$ and N$\varepsilon$BAI. Both algorithms succeed to identify an $\varepsilon$-optimal arm, while empirically, the complexity of PS$\varepsilon$BAI$^+$ is $\leq 1\%$ of that of N$\varepsilon$BAI. The empirical averages and standard deviations of the sample complexities of both algorithms are presented in Figure 2(a).

---

[3]We clarify that (3.5) in Assumption 1 is only for our theoretical guarantees. In practice, our algorithm has shown robustness w.r.t. the parameters and we can safely neglect the constants in the formula.

Figure 2(a) illustrates that empirically, the termination and arm recommendation of PS$\varepsilon$BAI$^+$ are determined by the execution of PS$\varepsilon$BAI as a subroutine, suggesting that in Theorem 3.3, the first term resulting from PS$\varepsilon$BAI actually determines the complexity of PS$\varepsilon$BAI$^+$.

**Next,** we test the efficacy of PS$\varepsilon$BAI$^+$ to learn and exploit the latent vectors and the distribution of contexts. Specifically, we run PS$\varepsilon$BAI$^+$ and N$\varepsilon$BAI under Dynamics 1 where neither the index nor the vector of current context is visible to the agent. We also run benchmark algorithms: D$\varepsilon$BAI and its variant D$\varepsilon$BAI$_\beta$ under Dynamics 3 where context vectors and changepoints are all observed; these two algorithms are detailed and analyzed in Appendix O.4.

As the changepoint sequence $\mathcal{C}$ is fixed in a given instance and $t/L_{\max} \le l_t \le t/L_{\min}$ for all $t \in \mathbb{N}$, we regard the number of context samples $l_\tau$ as a proxy of the sample complexity $\tau$. We present the number of context samples need by PS$\varepsilon$BAI$^+$, N$\varepsilon$BAI, D$\varepsilon$BAI and D$\varepsilon$BAI$_\beta$ for arm identification w.r.t. $1/(\Delta_{\min} + \varepsilon)^2$ in Figure 2(b).

Figure 2(b) contains three messages. First, the complexity of PS$\varepsilon$BAI$^+$ scales as $1/(\Delta_{\min} + \varepsilon)^2$, corroborating Theorem 3.3. Second, although PS$\varepsilon$BAI$^+$ has access to neither context vectors nor changepoints, it needs roughly the same number of context samples as D$\varepsilon$BAI and D$\varepsilon$BAI$_\beta$, suggesting that it is competitive compared to these algorithms that have oracle information about the environment. Third, D$\varepsilon$BAI$_\beta$ uses the confidence radius in (3.3) and terminates with fewer context samples compared to D$\varepsilon$BAI, implying that the confidence radius is well-designed.

**Furthermore,** when $\varepsilon$ decreases from $0.03 \times 1.35^{12}$ to $0.03 \times 1.35^9$, the complexity of PS$\varepsilon$BAI almost remains unchanged while that of N$\varepsilon$BAI increases rapidly as presented in instances 9 to 12 in Figure 2(a). Meanwhile, the number of context samples need by two algorithms are shown to be with the same pattern in Figure 2(b). This contrast indicates that the cost of distribution estimation ($T_{\mathrm{D}}$ in (3.6)) for PS$\varepsilon$BAI$^+$ has been significantly minimized compared to N$\varepsilon$BAI.

**To summarize,** we emphasize that the empirical superiority of PS$\varepsilon$BAI$^+$ over N$\varepsilon$BAI implies that *the efficacy of* PS$\varepsilon$BAI$^+$ *is inherited from* PS$\varepsilon$BAI. Our experiments show that actively exploiting the context information, via changepoint detection and context alignment (as in PS$\varepsilon$BAI$^+$ and PS$\varepsilon$BAI) facilitates identifying the $\varepsilon$-optimal arm efficiently.

Similar to many existing algorithms in piecewise-stationary bandits [21, 10, 22], our algorithm requires Assumption 1 and the knowledge of $L_{\max}$. These may not be available in practice. Thus, we conduct more experiments in Appendix O.2 to exhibit the robustness of PS$\varepsilon$BAI$^+$. Specifically, in Appendix O.2, we conduct experiments for the case in which $L_{\max}$ is *misspecified*. In Appendix O.3, we alter the change detection frequency $\gamma$ so that $w$ and $b$ change accordingly. In both sets of experiments, the overall sample complexity of PS$\varepsilon$BAI$^+$ does not vary significantly and retains its superiority over N$\varepsilon$BAI. We conclude that PS$\varepsilon$BAI$^+$ is robust to slight misspecifications in these parameters, as long as Assumption 1 is not severely violated. Please refer to Appendix O for further details and experiments.

# 7 Conclusion and Future Work

We proposed a novel PSLB model and designed the PS$\varepsilon$BAI$^+$ algorithm to identify an $\varepsilon$-optimal arm with probability $\ge 1 - \delta$. The efficacy of PS$\varepsilon$BAI$^+$ has been demonstrated both empirically and theoretically. We argued that this is due to the embedded change detection and context alignment procedures. There are several directions for further exploration.

**Firstly,** our PS$\varepsilon$BAI$^+$ algorithm provides a fairly general *framework* for algorithm design. For instance, in addition to utilization the G-optimal allocation to sample arms as in PS$\varepsilon$BAI$^+$, the $\mathcal{X}\mathcal{Y}$-allocation and adaptive $\mathcal{X}\mathcal{Y}$-allocation [1] can also be considered. In other words, our PS$\varepsilon$BAI$^+$ algorithm can be generalized to form an entire class of algorithms for BAI in PSLB models. In addition, deriving instance-dependent guarantees is also of great interest.

**Secondly,** most of the literature on piecewise-stationary bandits [21, 10, 22] make assumptions to provide theoretical guarantees. It would be interesting to remove or reduce these assumptions under our $\varepsilon$-BAI problem setup, and yet still be able to provide similar theoretical guarantees.

**Finally,** we believe that it is possible to adapt our PS$\varepsilon$BAI$^+$ algorithm to the fixed-budget setting, i.e., to identify an $\varepsilon$-optimal arm with high probability in a fixed time horizon in PSLB models.

**Acknowledgements:** This work is funded by the Singapore Ministry of Education AcRF Tier 2 grant (A-8000423-00-00) and Tier 1 grants (A-8000189-01-00 and A-8000980-00-00).

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

# Appendices

The contents of the appendices are organized as follows:

- In Appendix A, we provide further motivating examples for our problem.
- In Appendix B, we discuss the limitations of our method.
- In Appendix C, we review more related works on drifting and contextual bandits.
- In Appendix D, we provide more details about our algorithms
  - Appendix D.1: pseudo-code of N$\varepsilon$BAI in Algorithm 2.
  - Appendix D.2: more details of PS$\varepsilon$BAI including the precise definition of the confidence radius $\rho_t$ and details of LCD and LCA subroutines.
  - Appendix D.3: pseudo-code of PS$\varepsilon$BAI$^+$ in Algorithm 5.
  - Appendix D.4: the computational complexity of PS$\varepsilon$BAI in Algorithm 1.
- In Appendix E: we provide a useful lemma for estimating the expected return of any arm in linear bandits when the sampling rule is according to the G-optimal allocation.
- In Appendix F: we proof the upper bound on the expected sample complexity of N$\varepsilon$BAI.
- In Appendices G to K: we detail the analysis of PS$\varepsilon$BAI, i.e., we provide the proof of the upper bound on its complexity in Theorem 3.2.
  - Appendix G: outline of the proof.
  - Appendix H: analysis of the Change Detection (CD) and Context Alignment (CA) procedures.
  - Appendix I: analysis of the estimation error by decomposing it into three terms: Vector-Estimation Error (VE), Distribution-Estimation Error (DE), and Residual Estimation Error (RE).
  - Appendix J: proof of Theorem 3.2 based on the analysis above.
  - Appendix K: proof of technical lemmas that are utilized in the analysis of PS$\varepsilon$BAI.
- In Appendix L: we prove the upper bound on the expected sample complexity of PS$\varepsilon$BAI$^+$ based on the analysis of N$\varepsilon$BAI and PS$\varepsilon$BAI.
- In Appendix M: we derive the lower bound on the expected sample complexity of any algorithm, i.e., proof Theorem 4.1.
- In Appendix N: we provide more examples to compare the derived upper and lower bounds on the expected sample complexity, and illustrate the efficacy of our PS$\varepsilon$BAI$^+$ algorithm.
- In Appendix O: we provide more details of numerical experiments.
- In Appendix P: we provide more discussions on
  - the related methods for BAI in nonstationary bandits,
  - the instance-dependent upper bound,
  - the connection between the piecewise-stationary linear bandits model to the stationary linear bandits model,
  - the special case where $N = 1$.
- In Appendix Q: we provide analytical results on the "Best Arm Tuple Identification Problem".

## A    Further Motivating Examples

We elaborate on the some concrete real-life examples that motivate our problem setup of identifying the ensemble best arm in piecewise-stationary linear bandits.

In scenarios such as investment option selection and portfolio management also mentioned by [10, 20], there is a multitude of options for fund managers to choose from and typically, they want to find, in the initial pure exploration process, a small subset of candidate portfolios (or even the "best" portfolio) based on various economic indicators and the market performance of individual stocks before further exploitation. In a bearish market, more portfolios tend to incur losses; while in a bullish market, more portfolios tend to generate gains. The transition between these two contexts can be effected by stochastic factors, e.g., the weather, or the outbreak of a pandemic, making the market conditions (contexts) stochastic. In the face of these uncertainties (in the contexts and rewards), we wish to design and analyze algorithms that selected portfolio to yield the best long-term option under such a piecewise-stationary environment.

Crop rotation is another example. Since crop yields can be influenced by various factors, such as weather conditions (analogous to our stochastically generated contexts), selecting the most suitable crop to grow and harvest from is crucial. Given several candidate crops, crops of similar types (e.g., potatoes and sweet potatoes) are correlated as they tend to favor similar conditions, thus, they can be

modelled by bandits with a linear reward structure. Contextual factors, like weather conditions, are well-modelled as being stochastic. A fixed weather condition will last a period of time and it will not change suddenly. Domain knowledge from historical data/records provides us with prior knowledge on $L_{\min}$ and $L_{\max}$. These observations dovetail with our model. Our objective is to choose the crop that offers the near-highest yield potential over a long time period (an ensemble $\varepsilon$-best arm) and is adaptable to local environment factors, such as weather patterns.

## B  Limitations

Similar to existing works on piecewise stationary bandits [21, 10, 22], we introduced some assumptions to provide theoretical guarantees for PS$\varepsilon$BAI$^+$ although PS$\varepsilon$BAI$^+$ has shown to be robust even in the absence of these assumptions. Since an algorithm may not need to differentiate two contexts with close latent vectors for identifying an $\varepsilon$-optimal arm, we surmise it is possible to weaken these assumptions. For this purpose, we will consider clustering the contexts into a few classes based on the distances between their latent vectors and design an algorithm that only aims to detect the change of context class, instead of the change of context.

## C  More Related Work

In **drifting bandits** (DB), the regrets of algorithms are affected by the level of nonstationarity of the environment, which can be measured by various quantities, such as the total variation of the context sequence and the number of time steps when the return of at least one arm changes [23, 25].

In **contextual bandits** (CB), where the contextual information is visible to the agent, [16, 26, 17, 27] aim to identify the best arm with the assumption that the context changes at every time step according to a fixed distribution, and the return of an arm is averaged across all contexts. We see that the context distribution is involved to measure the quality of arms. However, while the agent in CB models can observe the context information, the agent has no access to the contexts but still aims to identify an $\varepsilon$-optimal arm in our piecewise stationary linear bandit (PSLB) model.

**In adversarial bandits**, existing works pertaining to the BAI problem only explored the fixed-horizon setting [28]. Due to the difference between the fixed-confidence and fixed-horizon setting and the difference between adversarial and piecewise stationary bandits, their results cannot be trivially extended to solve the fixed-confidence BAI problem in piecewise stationary bandits.

## D  More Details of Algorithms N$\varepsilon$BAI, PS$\varepsilon$BAI and PS$\varepsilon$BAI$^+$

All proposed algorithms make use of the well known G-optimal allocation (design) [1, 29], which is widely used in the linear bandits literature. The G-optimal allocation minimizes the maximal mean-squared prediction error in all directions [1]. Given an arm set $\mathcal{X}$, the G-optimal allocation $\lambda^*$ is a distribution over the arm set, which is the minimizer of $g(\lambda) = \max_{x \in \mathcal{X}} \|x\|^2_{A(\lambda)^{-1}}$, where $A(\lambda) = \sum_{x \in \mathcal{X}} \lambda(x) x x^\top$ and $\lambda \in \Delta_{\mathcal{X}}$. Interested readers may refer to Chapter 21 in [29].

### D.1  The NAÏVE $\varepsilon$-BEST ARM IDENTIFICATION (N$\varepsilon$BAI) Algorithm

As presented in Algorithm 2, N$\varepsilon$BAI samples an arm with the G-optimal allocation at each time steps; its stopping rule is grounded on the property of G-optimal allocation (see Lemma E.1) and affected by the maximum length of a single stationary segment $L_{\max}$.

**Algorithm 2** NAÏVE $\varepsilon$-BEST ARM IDENTIFICATION (N$\varepsilon$BAI)

---

1: **Input:** the arm set $\mathcal{X}$, the phase length bounds $L_{\min}, L_{\max}$, the slackness parameter $\varepsilon$, the confidence parameter $\delta$.
2: **Initialize**: Compute the G-optimal allocation $\lambda^*$.
3: Compute $C_3 = \sum_{n=1}^{\infty} n^{-3}$ and $t^* = 3d\ln(6dKC_3/\delta)$.
4: Sample $t^*$ arms $\{x_s\}_{s=1}^{t^*} \sim \lambda^*$ and observe the associated returns $\{Y_{s,x_s}\}_{s=1}^{t^*}$. Let $t = t^*$.
5: Compute

$$
\tilde{\theta}_t = \frac{1}{t}\sum_{s=1}^{t} A(\lambda^*)^{-1} x_s Y_{s,x_s}, \quad \dot{x}_t = \arg\max_{x \in \mathcal{X}} x^\top \tilde{\theta}_t,
$$
$$
\tilde{\rho}_t = \sqrt{\frac{8L_{\max}}{t}\ln\frac{4KC_3 t^3}{\delta}} + 5\sqrt{\frac{d}{t}\ln\frac{4KC_3 t^3}{\delta}}. \tag{D.1}
$$

6: **while** $\dot{x}_t^\top \tilde{\theta}_t - \tilde{\rho}_t + \varepsilon < \max_{x \neq \dot{x}_t} x^\top \tilde{\theta}_t + \tilde{\rho}_t$ **do**
7:     Sample an arm $x_t \sim \lambda^*$ and observe return $Y_{t,x_t}$ and let $t = t+1$.
8:     Update $\tilde{\theta}_t, \dot{x}_t$ and $\tilde{\rho}_t$ with (D.1).
9: **end while**
10: Recommend arm $\dot{x}_\varepsilon = \dot{x}_t$.

---

### D.2 Details about PS$\varepsilon$BAI

#### D.2.1 Confidence radius utilized in PS$\varepsilon$BAI

For each pair of arms $(x, \tilde{x})$, the confidence radius of $\Delta(x, \tilde{x})$ at time step $t$ is

$$
\rho_t(x, \tilde{x}) := 2(\alpha_t + \xi_t) + \sum_{j=1}^{N} \beta_{t,j} |\hat{\Delta}_{t,j}^{\mathrm{clip}_2}(x, \tilde{x}) + \zeta_t(x, \tilde{x})|,
$$

where $a^{\mathrm{clip}_2} := \min\{\max\{a, -2\}, 2\}$ denotes the value of $a$ that is clipped to the interval $[-2, 2]$ and

$$
\alpha_t := 5\sqrt{\frac{d}{T_t}\ln\frac{2}{\delta_{v,T_t}}}, \ \xi_t := 25\sqrt{2}\frac{N L_{\max}}{T_t}\ln\frac{2}{\delta_{m,T_t}},
$$
$$
\beta_{t,j} := \min\left\{\frac{5}{2}\sqrt{\frac{2\phi_{t,j} L_{\max}}{T_t}\ln\frac{2}{\delta_{d,T_t}}}, 1\right\},
$$
$$
\phi_{t,j} := \min\left\{4\max\left\{\hat{p}_{t,j}, \frac{25}{4}\frac{L_{\max}}{T_t}\ln\frac{2}{\delta_{d,T_t}}\right\}, \frac{1}{4}\right\},
$$
$$
\delta_{v,T_t} = \frac{\delta}{15KT_t^3}, \ \delta_{m,T_t} = \frac{\delta}{15KNT_t^3}, \ \delta_{d,T_t} = \frac{\delta}{15NT_t^3}.
$$

$\zeta_t(x, \tilde{x})$ minimizes the last summation. In the final theoretical upper bound on the sample complexity, we take $\zeta_t(x, \tilde{x}) = \varepsilon$ for simplicity. In the experiment, we utilize

$$
\zeta_t(x, \tilde{x}) = \arg\min_{\zeta_t(x,\tilde{x}) \in \mathbb{R}} \sum_{j=1}^{N} \beta_{t,j}(\hat{\Delta}_{t,j}(x, \tilde{x}) + \zeta_t(x, \tilde{x}))^2 = -\frac{\sum_{j=1}^{N} \beta_{t,j}\hat{\Delta}_{t,j}(x, \tilde{x})}{\sum_{j=1}^{N} \beta_{t,j}}
$$

for a simple and effective analytic expression in the experiment.

#### D.2.2 Subroutines of PS$\varepsilon$BAI

In the LCD subroutine (Algorithm 3), two estimates $\tilde{\theta}_1$ and $\tilde{\theta}_2$ of the latent vectors are independently computed from the first and second halves of the input CD samples (Line 2). LCD only raises an alarm of changepoint (Line 4) when the difference between $\tilde{\theta}_1$ and $\tilde{\theta}_2$ is sufficiently large (Line 3) and indicates a changepoint occurs w.h.p.

The LCA subroutine (Algorithm 4) estimates the latent vector of current context and updates the dictionary $\mathrm{CA_{id}}$. Specifically, it **firstly** samples $w/2$ samples with $\lambda^*$ (Line 2). It **then** checks

---

**Algorithm 3** LINEAR-CHANGE DETECTION (LCD)

---

1: **Input:** arm set $\mathcal{X}$, detection window $w$, threshold $b$ and $w$ arms with the associated observations $[(\tilde{x}_1, Y_{s,\tilde{x}_1}), \ldots, (\tilde{x}_w, Y_{w,\tilde{x}_w})]$.

2: Compute $\tilde{\theta}_1 = \frac{2}{w} \sum_{s=1}^{\frac{w}{2}} A(\lambda^*)^{-1} \tilde{x}_s Y_{s,\tilde{x}_s}$ and $\tilde{\theta}_2 = \frac{2}{w} \sum_{s=\frac{w}{2}+1}^{w} A(\lambda^*)^{-1} \tilde{x}_s Y_{s,\tilde{x}_s}$.

3: **if** $\exists x \in \mathcal{X}$, $s.t. |x^\top (\tilde{\theta}_2 - \tilde{\theta}_1)| > b$ **then**

4:     **Return** True

5: **else**

6:     **Return** False

7: **end if**

---

---

**Algorithm 4** LINEAR-CONTEXT ALIGNMENT (LCA)

---

1: **Input:** arm set $\mathcal{X}$, detection window $w$, threshold $b$ and context ids $\mathrm{CA}_{\mathrm{id}}$.

2: Sample $\frac{w}{2}$ arms $\{\tilde{x}_s\}_{s=1}^{\frac{w}{2}} \sim \lambda^*$ and observe the associated returns $\{Y_{s,\tilde{x}_s}\}_{s=1}^{\frac{w}{2}}$ , where

$$\{(x_s, Y_{s,x_s})\}_{s=t-\frac{w}{2}+1}^{t} = \{\tilde{x}_s, Y_{s,\tilde{x}_s}\}_{s=1}^{\frac{w}{2}}.$$

3: **for** $j \in \mathrm{CA}_{\mathrm{id}}$   **do**

4:     **if** not $\mathrm{LCD}(\mathcal{X}, w, b, [(\tilde{x}_s, Y_{s,\tilde{x}_s})]_{s=1}^{\frac{w}{2}} + \mathrm{CA}_{\mathrm{id}}[j])$ **then**

5:         $\mathrm{CA}_{\mathrm{id}}[j] = [(\tilde{x}_s, Y_{s,\tilde{x}_s})]_{s=1}^{\frac{w}{2}}$.

6:         **Return** $j, \mathrm{CA}_{\mathrm{id}}$.

7:     **end if**

8: **end for**

9: index $= \mathrm{len}(\mathrm{CA}_{\mathrm{id}}) + 1$.

10: $\mathrm{CA}_{\mathrm{id}}[\mathrm{index}] = [(\tilde{x}_s, Y_{s,\tilde{x}_s})]_{s=1}^{\frac{w}{2}}$.

11: **Return** index, $\mathrm{CA}_{\mathrm{id}}$.

---

whether the current latent context has been visited in previous time steps by scanning through $\mathrm{CA}_{\mathrm{id}}$ (Line 3). In order to learn if the current context can be aligned with context $j$, the LCD subroutine is called with the $w/2$ recent samples and $\mathrm{CA}_{\mathrm{id}}[j]$ as input:

**(i)** If the current context is aligned with $\mathrm{CA}_{\mathrm{id}}[j]$ (Line 4), the current latent context is $j$ w.h.p. Thus the LCA subroutine updates $\mathrm{CA}_{\mathrm{id}}[j]$ and returns the index $j$ (Lines 5 to 6), which would be $\hat{j}_t$ in Line 19 of Algorithm 1. Note that in Lines 10 and 11 of Algorithm 1, all the collected samples will be assigned to context $\hat{j}_t$ (until the next changing alarm) and will be used to estimate $\theta^*_{\hat{j}_t}$ and all $p_j$'s. Therefore, *aligning* contexts allows PS$\varepsilon$BAI to make good use of observation history.

**(ii)** If the current context is not aligned with any observed context, i.e., it has not been visited, $\mathrm{CA}_{\mathrm{id}}$ gets extended with a new index-samples pair and is returned along with the new index (Lines 9 to 11).

## D.3 The PIECEWISE-STATIONARY $\varepsilon$-BEST ARM IDENTIFICATION$^+$ (PS$\varepsilon$BAI$^+$) Algorithm

To help understand PS$\varepsilon$BAI$^+$ algorithm, we highlight the differences between PS$\varepsilon$BAI and PS$\varepsilon$BAI$^+$, and the differences between subroutines LINEAR-CONTEXT ALIGNMENT (LCA) and LINEAR-CONTEXT ALIGNMENT$^+$ (LCA$^+$).

---

**Algorithm 5** PIECEWISE-STATIONARY $\varepsilon$-BEST ARM IDENTIFICATION$^+$ (PS$\varepsilon$BAI$^+$)

---

1: **Input:** arm set $\mathcal{X}$, number of latent vectors $N$, bounds on the segment lengths $L_{\min}$ and $L_{\max}$, slackness parameter $\varepsilon$, confidence parameter $\delta$, sampling parameter $\gamma$, window size $w$ and threshold $b$.

2: **Initialize**: Compute the G-optimal allocation $\lambda^*$ and $\tau^* = \frac{38400\ln(80)NL_{\max}}{\varepsilon^2}\ln\frac{N^2 KL_{\max}}{\delta\varepsilon^2}$.

3: Set $\mathrm{CD}_{\mathrm{sample}} = [\,]$, $\mathrm{CA}_{\mathrm{id}} = \{\,\}$, $t_{\mathrm{CD}} = +\infty$. Set $\mathcal{T}_{t,j} = \emptyset$ and initialize $\mathcal{T}_t, T_{t,j}, T_t$ with (3.2) for all $t \leq \tau^*$, $j \in [N]$.

4: Compute $C_3 = \sum_{n=1}^{\infty} n^{-3}$ and $t^* = 3d\ln(6dKC_3/\delta)$.

5: Sample $w/2$ arms $\{x_s\}_{s=1}^{w/2} \sim \lambda^*$ and observe the associated returns $\{Y_{s,x_s}\}_{s=1}^{w/2}$.

6: Set $t = \frac{w}{2}$, $t_{\mathrm{CA}} = w/2$, $\mathrm{CA}_{\mathrm{id}} = \{1 : [(x_s, Y_{s,x_s})]_{s=1}^{w/2}\}$, $\hat{j}_t = 1$.

7: Set $\dot{x}_t = 0$, $\tilde{\rho}_t = 2\varepsilon$ and $\dot{x}_\varepsilon = -\infty$. Compute $\tilde{\theta}_t$ with (D.1).

8: **while** True **do**

9:     $t = t + 1$

10:     Sample an arm $x_t \sim \lambda^*$ and observe return $Y_{t,x_t}$.

11:     Compute $\tilde{\theta}_t$, $\dot{x}_t$, $\tilde{\rho}_t = \mathrm{EU}(t, \tilde{\theta}_{t-1}, t^*, C_3)$.

12:     **if** $\dot{x}_t^\top \tilde{\theta}_t - \tilde{\rho}_t + \varepsilon \geq \max_{x \neq \dot{x}_t} x^\top \tilde{\theta}_t + \tilde{\rho}_t$ **then**

13:         $\dot{x}_\varepsilon = \dot{x}_t$.

14:         **Break**

15:     **end if**

16:     **if** $t > \tau^*$ **then**

17:         **Continue**

18:     **end if**

19:     **if** $\mathrm{mod}\,(t - t_{\mathrm{CA}}, \gamma) \neq 0$ **then**

20:         Update $\hat{j}_t = \hat{j}_{t-1}$, $\mathcal{T}_{t,\hat{j}_t} = \mathcal{T}_{t-1,\hat{j}_t} \cup \{t\}$, $\mathcal{T}_{t,j} = \mathcal{T}_{t-1,j}$ for $j \neq \hat{j}_t$.

21:     **else**

22:         $\mathrm{CD}_{\mathrm{sample}} = \mathrm{CD}_{\mathrm{sample}} + [(x_t, Y_{t,x_t})]$.

23:         **if** $|\mathrm{CD}_{\mathrm{sample}}| \geq w$ **then**

24:             **if** $\mathrm{LCD}(\mathcal{X}, w, b, \mathrm{CD}_{\mathrm{sample}}[-w:])$ **then**

25:                 $\mathrm{CD}_{\mathrm{sample}} = [\,]$.

26:                 $t = t + \frac{w}{2}$, $t_{\mathrm{CA}} = t$, $t_{\mathrm{CD}} = +\infty$.

27:                 $\hat{j}_t, \mathrm{CA}_{\mathrm{id}} = \mathrm{LCA}^+(\mathcal{X}, w, b, \mathrm{CA}_{\mathrm{id}})$

28:                 **if** $\hat{j}_t = N + 1$ **then break**.

29:                 Revert $\mathcal{T}_{t,j} = \mathcal{T}_{t - \frac{w(\gamma+1)}{2}, j}$ for all $j \in [N]$.

30:             **end if**

31:         **end if**

32:     **end if**

33:     **if** $\dot{x}_\varepsilon \neq -\infty$ **then**

34:         **Break**

35:     **end if**

36:     Update the empirical estimates by (3.2) and (3.3).

37:     **if** condition (3.4) is met and $t_{\mathrm{CD}} = +\infty$ **then**

38:         Record $\hat{x}_\varepsilon = \arg\max_{x \in \mathcal{X}} x^\top \hat{\Theta}_t \hat{\mathbf{p}}_t$.

39:         $t_{\mathrm{CD}} = |\mathrm{CD}_{\mathrm{sample}}|$.

40:     **else if** $t_{\mathrm{CD}} = |\mathrm{CD}_{\mathrm{sample}}| - \frac{w}{2}$ **then**

41:         $\dot{x}_\varepsilon = \hat{x}_\varepsilon$

42:         **Break**

43:     **end if**

44: **end while**

45: Recommend $\mathring{x}_\varepsilon = \dot{x}_\varepsilon$.

---

---

**Algorithm 6** ESTIMATE UPDATE (EU)

---

1: **Input:** time step $t$, vector $\tilde{\theta}_{t-1}$, threshold $t^*$ and constant $C_3$.

2: Compute $\tilde{\theta}_t = \frac{1}{t}\left[(t-1)\tilde{\theta}_{t-1} + A(\lambda^*)^{-1}x_t Y_{s,x_t}\right],\ \dot{x}_t = 0,\ \tilde{\rho}_t = 2\varepsilon$.

3: **if** $t \geq t^*$ **then**

4:     Compute $\dot{x}_t = \underset{x \in \mathcal{X}}{\arg\max}\, x^\top \tilde{\theta}_t,\ \tilde{\rho}_t = \sqrt{\dfrac{8L_{\max}}{t}\ln\dfrac{4KC_3 t^3}{\delta}} + 5\sqrt{\dfrac{d}{t}\ln\dfrac{4KC_3 t^3}{\delta}}$.

5: **end if**

6: **Return** $\tilde{\theta}_t,\ \dot{x}_t,\ \tilde{\rho}_t$.

---

---

**Algorithm 7** LINEAR-CONTEXT ALIGNMENT$^+$ (LCA$^+$)

---

1: **Input:** arm set $\mathcal{X}$, detection window $w$, threshold $b$, context ids $\mathrm{CA_{id}}$, time step $t$, vector $\tilde{\theta}_t$, threshold $t^*$ and constant $C_3$.

2: Set $s = 0, \dot{x}_\varepsilon = -\infty$.

3: **for** $s \leq w/2$ **do**

4:     $s = s+1, t = t+1$.

5:     Sample an arm $\tilde{x}_s \sim \lambda^*$ and observe return $Y_{s,\tilde{x}_s}$.

6:     Set $(x_t, Y_{t,x_t}) = (\tilde{x}_s, Y_{s,\tilde{x}_s})$.

7:     Compute $\tilde{\theta}_t, \dot{x}_t, \tilde{\rho}_t = \mathrm{EU}(t, \tilde{\theta}_{t-1}, t^*, C_3)$.

8:     **if** $\dot{x}_t^\top \tilde{\theta}_t - \tilde{\rho}_t + \varepsilon \geq \max_{x \neq \dot{x}_t} x^\top \tilde{\theta}_t + \tilde{\rho}_t$ **then**

9:         $\dot{x}_\varepsilon = \dot{x}_t, \mathrm{index} = \mathrm{len}(\mathrm{CA_{id}}) + 1$.

10:         **Return** index, $\mathrm{CA_{id}}, \dot{x}_\varepsilon, \tilde{\theta}_t$.

11:     **end if**

12: **end for**

13: **for** $j \in \mathrm{CA_{id}}$ **do**

14:     **if** not $\mathrm{LCD}(w, b, [(\tilde{x}_s, Y_{s,\tilde{x}_s})]_{s=1}^{\frac{w}{2}}, \mathrm{CA_{id}}[j])$ **then**

15:         $\mathrm{CA_{id}}[j] = [(\tilde{x}_s, Y_{s,\tilde{x}_s})]_{s=1}^{\frac{w}{2}}$.

16:         **Return** $j, \mathrm{CA_{id}}$.

17:     **end if**

18: **end for**

19: $\mathrm{index} = \mathrm{len}(\mathrm{CA_{id}}) + 1$.

20: $\mathrm{CA_{id}}[\mathrm{index}] = [(\tilde{x}_s, Y_{s,\tilde{x}_s})]_{s=1}^{\frac{w}{2}}$.

21: **Return** index, $\mathrm{CA_{id}}, \dot{x}_\varepsilon, \tilde{\theta}_t$.

---

### D.4 Computational Complexity of PS$\varepsilon$BAI

We provide analysis for the computational complexity of PS$\varepsilon$BAI in this subsection.

We consider the number of these operations: arithmetic (addition and multiplication) operations, logic operations and comparison operations and we also regard $\ln(\cdot)$, $\sqrt{\cdot}$ and sampling from a distribution as one step of operation.

We decompose the main loop of Algorithm 1 as follows, where the lines with $O(1)$ operations are omitted:

- Exploration phase (Lines 8 to 11): $O(1)$.
- Change Detection phase (Lines 12 to 16):
  - The $LCD$ subroutine in Line 16 needs $O(wd^2 + Kd)$ operations.
- Context Alignment phase (Lines 17 to 21):
  - The $LCA$ subroutine in Line 19 needs $O((wd^2 + Kd)N)$ operations, as the $LCD$ subroutine will be invoked $N$ times in the worst case in Line 4 of $LCA$.
  - The reversion procedure in Line 21 needs $O(\gamma wd)$ operations.
- The updating procedure and the stopping rule checking (Lines 25 to 32):
  - Updating $\hat{\theta}_{t,j}$ needs $O(d^2)$ operations, as we need to incorporate the latest sample into the estimate.

- Updating $\hat{p}_{t,j}, j \in [N]$ requires $N$ operations.
- Updating the confidence radius and the empirical best arm need $O(KN)$ operations.
- The stopping condition in Equation (3.4) requires $O(K)$ operations.

We remark that some intermediate results can be stored to avoid repeated computations and we believe the algorithm is efficient overall. In particular, since the reward structure is linear, there are $K$ arms, and $N$ contexts, $O(d^2)$, $O(K)$, and $O(N)$ operations probably cannot be be avoided.

From the above analysis, in the decreasing order of the number of operations:

- The Change Detection phase requires the most budget as it will be invoked every $\gamma$ time steps.
- The Context Alignment phase requires the second many operations. While it requires many operations when it is implemented, it will not be called during a stationary segment.
- The updating procedure uses small portion of the operations, as $w$ is usually much greater than $K$ and $N$.
- The exploration phase demands constant operations in each loop.

# E   Auxiliary results

**Lemma E.1.** *Let $X_s$ be the arm drawn with the G-optimal allocation $\lambda^*$ at time step $s$ and $Y_{s,x_s} = x_s^\top \theta_{j_s}^* + \eta_s$ be the corresponding return with $\mathbb{E}\eta_s = 0$, $|x_s^\top \theta_{j_s}^*| \leq 1$ and $|\eta_s| \leq 1$. Then we have*

$$
\mathbb{P}\left[ \frac{1}{n} \left| \sum_{s=1}^{n} x^\top A(\lambda^*)^{-1} x_s Y_{s,x_s} - x^\top \theta_{j_s}^* \right| \geq \epsilon \right] \leq \delta
$$

*for all $\epsilon \geq \frac{d}{n} \ln \frac{2}{\delta} + \sqrt{\left( \frac{d}{n} \ln \frac{2}{\delta} \right)^2 + \frac{4d}{n} \ln \frac{2}{\delta}}$. In addition, if $n \geq \frac{d}{4} \ln \frac{2}{\delta}$, we have*

$$
\mathbb{P}\left[ \frac{1}{n} \left| \sum_{s=1}^{n} x^\top A(\lambda^*)^{-1} x_s Y_{s,x_s} - x^\top \theta_{j_s}^* \right| \geq 5\sqrt{\frac{d}{n} \ln \frac{2}{\delta}} \right] \leq \delta
$$

*Proof.* Note that the arms are selected according to the G-optimal design, and thus $\mathbb{E}_{x \sim \lambda^*}[xx^\top] = A(\lambda^*)$. Therefore, for all $s \in [n]$,

$$
\mathbb{E}\left[ x^\top A(\lambda^*)^{-1} x_s Y_{s,x_s} - x^\top \theta_{j_s}^* \right] = 0 \tag{E.1}
$$

and

$$
\begin{aligned}
&\left| x^\top A(\lambda^*)^{-1} x_s Y_{s,x_s} - x^\top \theta_{j_s}^* \right| \\
&\leq \left| x^\top A(\lambda^*)^{-1} x_s x_s^\top \theta_{j_s}^* \right| + \left| x^\top A(\lambda^*)^{-1} x_s \eta_s \right| + 1 \\
&\leq \left| x^\top A(\lambda^*)^{-1} x_s \right| \left| x_s^\top \theta_{j_s}^* \right| + \left| x^\top A(\lambda^*)^{-1} x_s \right| \left| \eta_s \right| + 1 \\
&\leq 2\left| x^\top A(\lambda^*)^{-1} x_s \right| + 1 \\
&\leq 2\|x\|_{A(\lambda^*)^{-1}} \|x_s\|_{A(\lambda^*)^{-1}} + 1 \\
&\leq 3d
\end{aligned} \tag{E.2}
$$

where we make use of the property of the G-optimal allocation in the last inequality

$$
\|x\|_{A(\lambda^*)^{-1}}^2 \leq d, \ \forall x \in \mathcal{X}. \tag{E.3}
$$

Additionally,

$$
\begin{aligned}
&\mathbb{E}\left[ \left( x^\top A(\lambda^*)^{-1} x_s Y_{s,x_s} \right)^2 \right] \\
&= \mathbb{E}\left[ \left( x^\top A(\lambda^*)^{-1} x_s (x_s^\top \theta_{j_s}^* + \eta_s) \right)^2 \right] \\
&= \mathbb{E}\left[ \left( x^\top A(\lambda^*)^{-1} x_s x_s^\top \theta_{j_s}^* \right)^2 \right] + \mathbb{E}\left[ \left( x^\top A(\lambda^*)^{-1} x_s \eta_s \right)^2 \right] \\
&\quad + 2\mathbb{E}\left[ x^\top A(\lambda^*)^{-1} x_s x_s^\top \theta_{j_s}^* \cdot x^\top A(\lambda^*)^{-1} x_s \eta_s \right] \\
&\stackrel{(a)}{=} \mathbb{E}\left[ \left( x^\top A(\lambda^*)^{-1} x_s \right)^2 \left( x_s^\top \theta_{j_s}^* \right)^2 \right] + \mathbb{E}\left[ \eta_s^2 \left( x^\top A(\lambda^*)^{-1} x_s \right)^2 \right]
\end{aligned} \tag{E.4}
$$

$$\overset{(b)}{\leq} 2\mathbb{E}\left[\left(x^\top A(\lambda^*)^{-1}x_s\right)^2\right]$$

$$\overset{(c)}{=} 2\,x^\top A(\lambda^*)^{-1}x$$

$$\overset{(d)}{\leq} 2d$$

where we make use of the fact that $\eta_t$ is zero-mean and is independent of other random variables in $(a)$; $|x_s^\top \theta_{j_s}^*| \leq 1$ and $\mathbb{P}\left[\eta_s^2 \leq 1\right] = 1$ in $(b)$; $x_s \sim \lambda^*$ in $(c)$; and the property of the G-optimal allocation (E.3) in $(d)$.

According to the Bernstein's inequality with (E.1), (E.2) and (E.4),

$$\mathbb{P}\left[\frac{1}{n}\Big|\sum_{s=1}^n x^\top A(\lambda^*)^{-1}x_s Y_{s,x_s} - x^\top\theta_{j_s}^*\Big| \geq \epsilon\right]$$

$$\leq 2\exp\left(-\frac{\frac{1}{2}(n\epsilon)^2}{n\cdot 2d + dn\epsilon}\right)$$

$$= 2\exp\left(-\frac{n\epsilon^2}{4d + 2d\epsilon}\right).$$

In order to upper bound the error probability by $\delta$, we need

$$2\exp\left(-\frac{n\epsilon^2}{4d + 2d\epsilon}\right) \leq \delta$$

$$\Rightarrow \quad \epsilon \geq \frac{d}{n}\ln\frac{2}{\delta} + \sqrt{\left(\frac{d}{n}\ln\frac{2}{\delta}\right)^2 + \frac{4d}{n}\ln\frac{2}{\delta}}.$$

In addition, if $n \geq \frac{d}{4}\ln\frac{2}{\delta}$, we have

$$5\sqrt{\frac{d}{n}\ln\frac{2}{\delta}} = \frac{5}{2}\max\left\{\frac{d}{n}\ln\frac{2}{\delta}, \sqrt{\frac{4d}{n}\ln\frac{2}{\delta}}\right\} \geq \frac{d}{n}\ln\frac{2}{\delta} + \sqrt{\left(\frac{d}{n}\ln\frac{2}{\delta}\right)^2 + \frac{4d}{n}\ln\frac{2}{\delta}}.$$

This finishes the proof. $\qquad\square$

# F   Analysis of N$\varepsilon$BAI

To analyze the theoretical performance of N$\varepsilon$BAI, we first show that it can identify an $\varepsilon$-optimal arm with probability $1-\delta$ and derive a high-probability upper bound on its stopping time in Lemma F.1.

**Lemma F.1** (High-probability upper bound of N$\varepsilon$BAI). *With probability $1-\delta$, the N$\varepsilon$BAI algorithm identifies an $\varepsilon$-optimal arm after at most*

$$\tilde{O}\left(\frac{L_{\max} + d}{(\varepsilon + \Delta_{\min})^2}\ln\frac{1}{\delta}\right)$$

*time steps, where $\Delta_{\min} = \min_{i \neq i^*}\Delta(x^* - x_i)$.*

Next, we prove that after a sufficiently large number of time steps, the probability that N$\varepsilon$BAI does not terminate is small in Lemma F.2.

**Lemma F.2.** *Let*

$$T_0 = \frac{768(8L_{\max} + 25d)}{(\varepsilon + \Delta_{\min})^2}\ln\frac{768KC_3(8L_{\max} + 25d)}{(\varepsilon + \Delta_{\min})^2\delta}$$

*with $C_3 = \sum_{n=1}^\infty n^{-3}$. For $t \geq T_0$, the probability that N$\varepsilon$BAI does not terminate before $t$ time steps is $\frac{\delta}{(\alpha-1)C_3(t/2)^2}$.*

Lastly, we apply Lemmas F.1 and F.2 to prove Proposition 3.1.

**Proposition 3.1.** *Let* $\Delta_{\min} = \min_{x \neq x^*} \Delta(x^*, x)$,

$$T_{\mathrm{V}}^{\mathrm{N}} = \frac{d}{(\varepsilon + \Delta_{\min})^2} \ln \frac{1}{\delta} \quad and \quad T_{\mathrm{D}}^{\mathrm{N}} = \frac{L_{\max}}{(\varepsilon + \Delta_{\min})^2} \ln \frac{1}{\delta}.$$

*The* N$\varepsilon$BAI *algorithm is* $(\varepsilon, \delta)$-*PAC and its expected sample complexity is* $\tilde{O}(T_{\mathrm{V}}^{\mathrm{N}} + T_{\mathrm{D}}^{\mathrm{N}})$.

The detailed analysis is presented as below.

### F.1 Detailed analysis of N$\varepsilon$BAI

*Proof of Lemma F.1.* The result is to be proven with the following procedures.

**First step**: Prove $\left| x^\top \mathbb{E}_{\theta \sim P_\theta} \theta - x^\top \tilde{\theta}_t \right|$ can be bounded with high probability.

Let $\bar{\theta}_t = \frac{1}{t} \sum_{s=1}^t \theta_{j_s}^*$. (i) For all $\varepsilon > 0$, we have,

$$\mathbb{P}\left[ \left| x^\top \mathbb{E}_{\theta \sim P_\theta} \theta_t - x^\top \bar{\theta} \right| > \varepsilon \right] = \mathbb{P}\left[ \left| t x^\top \mathbb{E}_{\theta \sim P_\theta} \theta - x^\top \left( \sum_{l=1}^{N_t} L_l \theta^* j_{c_l} \right) \right| > t\varepsilon \right]$$

$$= \mathbb{P}\left[ \left| \sum_{l=1}^{N_t} L_l \left( x^\top \mathbb{E}_{\theta \sim P_\theta} \theta - L_l x^\top \theta^* j_{c_l} \right) \right| > t\varepsilon \right] \overset{(a)}{\leq} 2 \exp\left( -\frac{t^2 \varepsilon^2}{2 \sum_{l=1}^{N_t} (2L_l)^2} \right)$$

$$\overset{(b)}{\leq} 2 \exp\left( -\frac{t\varepsilon^2}{8 L_{\max}} \right).$$

Since $|x^\top \mathbb{E}_{\theta \sim P_\theta} \theta| \leq 1$ and $x^\top \theta^* j_{c_l} | \leq 1$ for all $l$, we obtain (a) by applying Hoeffding's inequality. We obtain (b) from the fact that $\sum_{l=1}^{N_t} L_l = t$ and $L_l \leq L_{\max}$ for all $l$. In other words, for all $\delta > 0$,

$$\mathbb{P}\left[ \left| x^\top \mathbb{E}_{\theta \sim P_\theta} \theta - x^\top \bar{\theta} \right| > \sqrt{\frac{8 L_{\max}}{t} \ln \frac{2}{\delta}} \right] \leq \delta.$$

(ii) Besides, according to Lemma E.1, if $t \geq \frac{d}{4} \ln \frac{2}{\delta}$, we have

$$\mathbb{P}\left[ \left| x^\top \tilde{\theta}_t - x^\top \bar{\theta}_t \right| \geq 5\sqrt{\frac{d}{t} \ln \frac{2}{\delta}} \right] = \mathbb{P}\left[ \frac{1}{t} \left| \sum_{s=1}^t x^\top A(\lambda^*)^{-1} x_s Y_{s, x_s} - x^\top \theta_{j_s}^* \right| \geq 5\sqrt{\frac{d}{t} \ln \frac{2}{\delta}} \right] \leq \delta.$$

(iii) For all $t > 0$, all $x \in \mathcal{X}$, since

$$\left| x^\top \mathbb{E}_{\theta \sim P_\theta} \theta - x^\top \tilde{\theta}_t \right| \leq \left| x^\top \mathbb{E}_{\theta \sim P_\theta} \theta - x^\top \bar{\theta} \right| + \left| x^\top \tilde{\theta}_t - x^\top \bar{\theta}_t \right|,$$

we have

$$\mathbb{P}\left[ \left| x^\top \mathbb{E}_{\theta \sim P_\theta} \theta - x^\top \tilde{\theta}_t \right| > \tilde{\rho}_t(\alpha) \right] \leq \frac{\delta}{KC_\alpha t^\alpha} \quad \text{when} \quad t \geq \frac{d}{4} \ln \frac{4KC_\alpha t^\alpha}{\delta},$$

where

$$\tilde{\rho}_t(\alpha) = \sqrt{\frac{8 L_{\max}}{t} \ln \frac{4KC_\alpha t^\alpha}{\delta}} + 5\sqrt{\frac{d}{t} \ln \frac{4KC_\alpha t^\alpha}{\delta}}, \quad C_\alpha = \sum_{n=1}^\infty n^{-\alpha}, \quad \alpha > 2.$$

For simplicity, we set $\alpha = 3$ and write $\tilde{\rho}_t(3)$ as $\tilde{\rho}_t$ in the following analysis. We now show that $t \geq \frac{d}{4} \ln \frac{4KC_\alpha t^\alpha}{\delta}$ holds when $t \geq 3d \ln \frac{6dKC_3}{\delta}$ with the following lemma (the proof of which is presented by the end of Appendix F.1).

**Lemma F.3.** *Let* $a, b, c > 0$ *and* $ac \geq 1/2$, *then*

$$t \geq \max\{2a \ln b, 4ac \ln(2ac)\} \implies t \geq a \ln(bt^c).$$

With

$$a = \frac{d}{4}, \quad b = \frac{4KC_\alpha}{\delta} = \frac{4KC_3}{\delta}, \quad c = \alpha = 3,$$

Lemma F.3 implies that

$$t \geq 4ac\ln(2abc) = 3d\ln\frac{6dKC_3}{\delta} \ \Rightarrow\ t \geq \max\{2a\ln b,\, 4ac\ln(2ac)\} \ \Rightarrow\ t \geq a\ln(bt^c).$$

Define event $\tilde{E}_t = \bigcap_{x\in\mathcal{X}} \left\{ \left| x^\top \mathbb{E}_{\theta\sim P_\theta}\theta - x^\top\tilde\theta_t \right| \leq \tilde\rho_t(\alpha) \right\}$ for $t \geq t^*$ and $\tilde{E} = \bigcap_{t\geq t^*}\tilde{E}_t$. We have

$$\mathbb{P}\left[\tilde{E}_t^\mathsf{c}\right] \leq \sum_{x\in\mathcal{X}}\frac{\delta}{KC_\alpha t^\alpha} = \frac{\delta}{C_\alpha t^\alpha}, \quad \mathbb{P}\left[\tilde{E}^\mathsf{c}\right] \leq \sum_{t=t^*}^\infty \mathbb{P}\left[\tilde{E}_t^\mathsf{c}\right] \leq \sum_{t=1}^\infty \frac{\delta}{C_\alpha t^\alpha} = \delta.$$

We assume $\tilde{E}$ holds in the second and third steps in this proof.

**Second step**: Prove N$\varepsilon$BAI recommends an $\varepsilon$-optimal arm when it stops.

If the algorithm terminates and returns $\dot{x}_\varepsilon \neq x^*$, we have

$$\dot{x}_\varepsilon^\top\tilde\theta_t - \tilde\rho_t + \varepsilon \geq \max_{x\neq\dot{x}_\varepsilon} x^\top\tilde\theta_t + \tilde\rho_t.$$

Since

$$\dot{x}_\varepsilon^\top\mathbb{E}_{\theta\sim P_\theta}\theta + \varepsilon \geq \dot{x}_\varepsilon^\top\tilde\theta_t - \tilde\rho_t + \varepsilon \quad\text{and}\quad \max_{x\neq\dot{x}_\varepsilon} x^\top\tilde\theta_t + \tilde\rho_t \geq \max_{x\neq\dot{x}_\varepsilon} x^\top\mathbb{E}_{\theta\sim P_\theta}\theta \geq x^{*\top}\mathbb{E}_{\theta\sim P_\theta}\theta,$$

we have

$$\dot{x}_\varepsilon^\top\mathbb{E}_{\theta\sim P_\theta}\theta + \varepsilon \geq x^{*\top}\mathbb{E}_{\theta\sim P_\theta}\theta,$$

implying that $\dot{x}_\varepsilon \in \mathcal{X}_\varepsilon$. Hence, we always have $\dot{x}_\varepsilon \in \mathcal{X}_\varepsilon$.

**Third step**: Derive the stopping time of N$\varepsilon$BAI.

(i) Let $x_{(2)} = \arg\max_{x\neq x^*} x^\top\mathbb{E}_{\theta\sim P_\theta}\theta$ and $\Delta_{\min} = \min_{x\neq x^*}\Delta(x^*-x) = (x^*-x_{(2)})^\top\mathbb{E}_{\theta\sim P_\theta}\theta$. We apply the following lemma to show that N$\varepsilon$BAI stops when $\tilde\rho_t$ is sufficiently small.

**Lemma F.4** ([30, Lemma 3]). *Denote by $\hat{i}$ the index of the item with empirical mean is $i$-th largest: i.e., $\hat{w}(\hat{1}) \geq \ldots \geq \hat{w}(\hat{L})$. Assume that the empirical means of the arms are controlled by $\varepsilon$ : i.e., $\forall i, |\hat{w}(i) - w(i)| < \varepsilon$. Then,*

$$\forall i, w(i) - \varepsilon \leq \hat{w}(\hat{i}) \leq w(i) + \varepsilon.$$

Lemma F.4 implies that

$$\dot{x}_t^\top\tilde\theta_t - \tilde\rho_t + \varepsilon \geq x^{*\top}\mathbb{E}_{\theta\sim P_\theta}\theta - 2\tilde\rho_t + \varepsilon = x_{(2)}^\top\mathbb{E}_{\theta\sim P_\theta}\theta - 2\tilde\rho_t + \varepsilon + \Delta_{\min} \quad\text{and}$$

$$\max_{x\neq\dot{x}_t} x^\top\tilde\theta_t + \tilde\rho_t \leq x_{(2)}^\top\mathbb{E}_{\theta\sim P_\theta}\theta + 2\tilde\rho_t.$$

When $\tilde\rho_t \leq (\varepsilon + \Delta_{\min})/4$, we have

$$\dot{x}_t^\top\tilde\theta_t - \tilde\rho_t + \varepsilon \geq x_{(2)}^\top\tilde\theta_t + \frac{\varepsilon+\Delta_{\min}}{2} \geq x_{(2)}^\top\tilde\theta_t + 2\tilde\rho_t \geq \max_{x\neq\dot{x}_t} x^\top\tilde\theta_t + \tilde\rho_t,$$

which will lead to the termination of N$\varepsilon$BAI.

(ii) According to the definition of $\tilde\rho_t$, we have

$$\tilde\rho_t(\alpha) \leq (\varepsilon + \Delta_{\min})/4$$

$$\Leftrightarrow \sqrt{\frac{8L_{\max}}{t}\ln\frac{4KC_\alpha t^\alpha}{\delta}} + 5\sqrt{\frac{d}{t}\ln\frac{4KC_\alpha t^\alpha}{\delta}} \leq (\varepsilon+\Delta_{\min})/4$$

$$\Leftrightarrow \sqrt{\frac{1}{t}\ln\frac{4KC_\alpha t^\alpha}{\delta}}\left(\sqrt{8L_{\max}} + 5\sqrt{d}\right) \leq (\varepsilon+\Delta_{\min})/4$$

$$\Leftrightarrow t \geq \frac{\left(\sqrt{8L_{\max}} + 5\sqrt{d}\right)^2}{(\varepsilon+\Delta_{\min})^2/16}\ln\frac{4KC_\alpha t^\alpha}{\delta}$$

$$\Leftarrow t \geq \frac{32(8L_{\max} + 25d)}{(\varepsilon+\Delta_{\min})^2}\ln\frac{4KC_\alpha t^\alpha}{\delta}$$

$$\Leftarrow \quad t \geq \frac{192(8L_{\max} + 25d)}{(\varepsilon + \Delta_{\min})^2} \ln \frac{768 K C_3 (8L_{\max} + 25d)}{(\varepsilon + \Delta_{\min})^2 \delta}.$$

We apply Lemma F.3 to invert the last line above. With

$$a = \frac{32(8L_{\max} + 25d)}{(\varepsilon + \Delta_{\min})^2}, \quad b = \frac{4KC_\alpha}{\delta}, \quad c = \alpha = 3,$$

Lemma F.3 indicates that

$$t \geq 4ac \ln(2abc) = \frac{384(8L_{\max} + 25d)}{(\varepsilon + \Delta_{\min})^2} \ln \frac{768 K C_3 (8L_{\max} + 25d)}{(\varepsilon + \Delta_{\min})^2 \delta}$$
$$\Rightarrow \quad t \geq \max\{2a \ln b, \, 4ac \ln(2ac)\} \quad \Rightarrow \quad t \geq a \ln(bt^c).$$

**Altogether**, we show that with probability $1 - \delta$, N$\varepsilon$BAI identifies an $\varepsilon$-optimal arm after at most

$$\frac{384(8L_{\max} + 25d)}{(\varepsilon + \Delta_{\min})^2} \ln \frac{768 K C_3 (8L_{\max} + 25d)}{(\varepsilon + \Delta_{\min})^2 \delta}$$

time steps.

$\square$

*Proof of Lemma F.2.* To begin with, let

$$T_0 = \frac{768(8L_{\max} + 25d)}{(\varepsilon + \Delta_{\min})^2} \ln \frac{768 K C_3 (8L_{\max} + 25d)}{(\varepsilon + \Delta_{\min})^2 \delta}$$

For any $T \geq T_0$, let $\bar{T} = \lceil T/2 \rceil$.

**(I)** If N$\varepsilon$BAI terminates within $\bar{T}$ time steps, there is nothing to prove.

**(II)** Assume N$\varepsilon$BAI does not terminates within $\bar{T}$ time steps. According to the analysis in the proof of Lemma F.1, if N$\varepsilon$BAI does not terminate before $T$ time steps, i.e., the stopping time $\tau$ satisfies that $\tau \geq \bar{T}$, then event $\bigcap_{t=\bar{T}+1}^{T-1} \tilde{E}_t$ does not hold. This indicates

$$\mathbb{P}\left[\tau \geq T | \tau \geq \bar{T}\right] \leq \mathbb{P}\left[\bigcup_{t=\bar{T}+1}^{T-1} \tilde{E}_t^{\mathsf{c}}\right] \leq \sum_{t=\bar{T}}^{T-1} \mathbb{P}\left[\tilde{E}_t^{\mathsf{c}}\right] \leq \sum_{t=\bar{T}}^{T-1} \frac{\delta}{C_\alpha t^\alpha} \leq \frac{\delta}{C_\alpha} \int_{\bar{T}}^{T} t^{-\alpha} \, \mathrm{d}t$$
$$\leq \frac{\delta}{(\alpha - 1)C_\alpha} \left(\frac{1}{(\bar{T})^{\alpha-1}} - \frac{1}{T^\alpha}\right) \overset{(a)}{\leq} \frac{\delta}{(\alpha - 1)C_\alpha (T/2)^{\alpha-1}}.$$

(a) results from the fact that $\bar{T} \geq T/2$.

**Altogether**, for $T \geq T_0$, the probability N$\varepsilon$BAI does not terminate before $T$ time steps is $\frac{\delta}{(\alpha-1)C_\alpha (T/2)^{\alpha-1}}$.

$\square$

*Proof of Proposition 3.1.* First, Tonelli's Theorem implies that

$$\mathbb{E}[\tau] = \mathbb{E}\left[\int_0^\tau 1 \, \mathrm{d}x\right] = \mathbb{E}\left[\int_0^{+\infty} \mathbb{I}(\tau > x) \, \mathrm{d}x\right] = \int_0^{+\infty} \mathbb{E}\left[\mathbb{I}(\tau > x)\right] \, \mathrm{d}x = \int_0^{+\infty} \mathbb{P}(\tau > x) \, \mathrm{d}x.$$

Next, we apply Lemma F.2 to show

$$\mathbb{E}\tau \leq T_0 + \mathbb{E}[\tau | \tau \geq T_0 + 1] \cdot \mathbb{P}[\tau \geq T_0 + 1] \leq T_0 + \int_{T_0}^{+\infty} \mathbb{P}(\tau \geq x) \, \mathrm{d}x$$

$$\leq T_0 + \int_{T_0}^{+\infty} \frac{\delta}{(\alpha - 1)C_\alpha (x/2)^{\alpha-1}} \, \mathrm{d}x = T_0 + \frac{\delta}{(\alpha - 1)(\alpha - 2)(T_0/2)^{\alpha-2}}. \quad \text{(F.1)}$$

Besides, Lemma F.1 indicates that N$\varepsilon$BAI can identify an $\varepsilon$-optimal arm with probability $1 - \delta$. $\square$

*Proof of Lemma F.3.* For $x > 0$, let

$$f(x) = x - a\ln(bx^c) = x - a\ln b - ac\ln x = x\left(1 - \frac{a\ln b}{x} - \frac{ac\ln x}{x}\right).$$

Since

$$f(x) \geq 0 \impliedby \begin{cases} \dfrac{a\ln b}{x} \leq \dfrac{1}{2} \\[2mm] \dfrac{ac\ln x}{x} \leq \dfrac{1}{2} \end{cases} \iff \begin{cases} x \geq 2a\ln b \\[2mm] x \geq 2ac\ln x \end{cases}.$$

Let $d = 2ac$, $x = d\ln(dy)$, then

$$x \geq 2ac\ln x = d\ln x \iff d\ln(dy) \geq d\ln(d\ln(dy)) \iff \ln(dy) \geq \ln(d\ln(dy)) \iff y \geq \ln(dy).$$

Since $z^{0.4} \geq \ln z$ for all $z \geq 1$, we have $\ln(dy) \leq (dy)^{0.4} \leq \sqrt{dy}$ and

$$y \geq \ln(dy) \impliedby y \geq \sqrt{dy} \impliedby y \geq d$$

when $yd \geq 1$. Hence, when $ac \geq 1/2$, $y \geq d = 2ac \geq 1$, we have $y \geq \ln(dy)$. Furthermore, for $x$ such that $x \geq \max\{2a\ln b,\ 4ac\ln(2ac)\}$, we have $f(x) \geq 0$.

$\square$

# G Proof Outline of Theorem 3.2

A proof outline of Theorem 3.2 is provided in this section. It consists of three steps:

**Step 1:** PS$\varepsilon$BAI (Algorithm 1) depends on the success of change detection and context alignment (Algorithm 3 and 4). We firstly upper bound the failure probability of these two subroutines via Lemma G.1, Lemma G.2, Lemma G.3 and summarized in Lemma G.4. More details about these two subroutines are provided in Appendix D.2.2 and the proof of the Lemmas are postponed to Appendix H.

**Step 2:** Subsequently, conditioned on their success, we provide a theoretical guarantee for the choice of the confidence radius $\rho_t(x, \tilde{x})$ for $\Delta(x, \tilde{x})$ at time step $t$ in Lemma G.5. The proof is detailed in Appendix I.

**Step 3:** Lastly, utilizing the above elements, we provide a sufficient condition for the stopping rule, and upper bound the number of time steps in Exp phases $T_\tau$ via Lemma G.6 whose proof is presented in Appendix J. As the total number of time steps $\tau$ is upper bounded by a constant multiple of $T_\tau$, the high-probability upper bound on the stopping time $\tau$ is obtained. This finishes the proof of Theorem 3.2 (please refer to Appendix J).

**Step 1:** We borrow the terminology in hypothesis testing to define the errors. Let the null hypothesis be: *the algorithm has undergone a changepoint within the last $w$ CD samples*. For Algorithm 3, we will characterize the type I error: *the algorithm has experienced a changepoint within the last $w$ CD samples but it fails to raise a changing alarm*, and the type II error: *a changepoint has not occurred but Algorithm 3 raises a changing alarm*. We refer to the event Failed Alarm (FA) and the event False Alarm Error (FAE) as that a type I error occurs and that a type II error occurs, respectively:

$$\text{FA}_l := \{\text{a type I error occurs at changepoint } c_l\}, l \in \mathbb{N},$$

$$\text{FAE}_t := \{\text{a type II error occurs time step } t\}, t \in \mathcal{T}_{\text{FAE}},$$

$$\text{FA} := \bigcup_{\{l:c_l \leq \tau\}} \text{FA}_l \quad \text{and} \quad \text{FAE} := \bigcup_{t \in \mathcal{T}_{\text{FAE}}} \text{FAE}_t,$$

where $\mathcal{T}_{\text{FAE}} := \{t : t \text{ in CD phase}, t \leq \tau, [t - w\gamma, t] \cap \mathcal{C} = \emptyset\}$ and $\tau$ is the stopping time. In term of Algorithm 4, define the event $\text{MI}_l := \{\text{Algorithm 4 misidentifies } \theta^*_{j_t}\}$, $l \in \mathbb{N}$ and $\text{MI} := \bigcup_{\{l:c_l \leq \tau\}} \text{MI}_l$. Define the good event

$$\text{Good} := \{\text{FA}^c \cap \text{FAE}^c \cap \text{MI}^c\}. \tag{G.1}$$

We upper bound the failure probability of Goodin Lemma G.1, Lemma G.2, Lemma G.3 and conclude the results in Lemma G.4.

**Lemma G.1.** *For any $\delta_{\text{FAE}} \in (0,1)$, with $b \geq \frac{8d}{3w} \ln \frac{2}{\delta_{\text{FAE}}} + \sqrt{\left(\frac{8d}{3w} \ln \frac{2}{\delta_{\text{FAE}}}\right)^2 + \frac{24}{w} d \ln \frac{2}{\delta_{\text{FAE}}}}$, LCD makes no false alarm before the stopping time $\tau$ with probability at least*

$$\mathbb{P}[\text{FAE}^c] \geq 1 - \frac{\tau}{\gamma} K \delta_{\text{FAE}}.$$

Lemma G.1 can also be stated as, fix $b$,

$$\mathbb{P}\left[\text{FAE}^c\right] \geq 1 - \frac{\tau}{\gamma} 2K \exp\left(-\min\left\{\frac{3wb}{20d}, \frac{wb^2}{150d}\right\}\right),$$

which indicates we can always decrease the error probability by enlarging the window size $w$.

Assume $c_l$ is observable, i.e., $\theta^*_{j_{c_l}} \neq \theta^*_{j_{c_{l+1}}}$, and denote $\hat{c}_l$ as the alarm time of $c_l$ from Algorithm 3.

**Lemma G.2.** *Conditional on $\text{FAE}^c$, for any $\delta_{\text{FA}} \in (0,1)$, if $\frac{\Delta_c - b}{2} \geq \frac{d}{w} \ln \frac{2}{\delta_{\text{FA}}} + \sqrt{\left(\frac{d}{w} \ln \frac{2}{\delta_{\text{FA}}}\right)^2 + \frac{4d}{w} \ln \frac{2}{\delta_{\text{FA}}}}$,*

$$\mathbb{P}\left[c_l \leq \hat{c}_l \leq c_l + \frac{w\gamma}{2} \Big| \hat{c}_l \geq c_l\right] \geq 1 - \delta_{\text{FA}}.$$

*In addition, $\mathbb{P}\left[\text{FA}\big|\text{FAE}^c\right] \leq l_\tau \delta_{\text{FA}}$.*

Lemma G.2 guarantees a prompt change alarm will be raised within $\frac{w\gamma}{2}$ time steps after a changepoint occurs. This also explains why $\frac{w\gamma}{2}$ samples are abandoned at line 16 of Algorithm 1 so that $\mathcal{T}_{t,j}$ only keeps samples from context $j$.

**Lemma G.3.** *Conditional on $\text{FAE}^c$ and $\text{FA}^c$, based on the conditions in Lemma G.1 and Lemma G.2,*

$$\mathbb{P}\left[\text{MI}|\text{FAE}^c, \text{FA}^c\right] \leq l_\tau \left[(N-1)\delta_{\text{FA}} + K\delta_{\text{FAE}}\right].$$

The intuition of Lemma G.3 is simple: change alarm should not be raised when the samples are from the same context and, change alarms should be raised when the samples are from the other $(N-1)$ contexts. The error made by Algorithm 3 and 4 can be concluded as follows

**Lemma G.4.** *Assume the instance satisfies Assumption 1,*

$$\mathbb{P}\left[\text{Good}\right] \geq 1 - \frac{\delta}{2\tau^*} \geq 1 - \frac{\delta}{2}.$$

**Step 2:** To give an upper bound on $T_\tau$, we firstly prove that the estimate of $\Delta(x, \tilde{x})$ at time $t$ is within distance $\rho_t(x, \tilde{x})$ from the ground truth w.h.p. Define the event CI as the estimates of all the mean gaps locate in the high-probability Confidence Interval

$$\text{CI}_t := \{|\hat{\Delta}_t(x, \tilde{x}) - \Delta(x, \tilde{x})| \leq \rho_t(x, \tilde{x}), \forall x, \tilde{x} \in \mathcal{X}\}$$

and $\text{CI} := \bigcap_t \text{CI}_t$, where $\rho_t$ is defined in (3.3).

**Lemma G.5.** *If $T_t \geq \frac{2L_{\max}}{9} \ln \frac{2}{\delta_{d,T_t}}$,*

$$\mathbb{P}\left[\text{CI}_t\big|\text{Good}\right] \geq 1 - \left(K\delta_{v,T_t} + N\delta_{d,T_t} + KN\delta_{m,T_t}\right).$$

*In addition,*

$$\mathbb{P}\left[\text{CI}\big|\text{Good}\right] \geq 1 - \frac{\delta}{2}.$$

In addition to an estimation over the latent vector which is sufficient under the stationary case ($\alpha_t$ in $\rho_t$), we also need to control the derivation between the empirical distribution $\hat{\mathbf{p}}_t$ and the ground truth $\mathbf{p}$ ($\beta_t$ in $\rho_t$), as well as the interactions between the latent vectors and the distribution ($\xi_t$ in $\rho_t$). This is reflected by $K\delta_{v,T_t}, N\delta_{d,T_t}$ and $KN\delta_{m,T_t}$ which bound the the failure probability of Vector Estimation, Distribution Estimation and Residual Estimation, respectively.

**Step 3:**

**Lemma G.6.** *Conditional on Good and CI, the recommended arm $\hat{x}_\varepsilon \in \mathcal{X}_\varepsilon$ and when Algorithm 1 terminates, the order of $T_t$ is upper bounded by (3.6).*

The detailed proof is presented in Appendix J.

# H Analysis of PS$\varepsilon$BAI: Change Detection and Context Alignment

As the output of Algorithms 3 and 4 depends on random samples, the output may not meet our expectation with some probabilities. We will characterize the probabilities of the three "bad" events: FAE in Appendix H.1, FA in Appendix H.2 and MI in Appendix H.3; and finally upper bound the failure probability of the good event Good in this section.

**Lemma G.4.** *Assume the instance satisfies Assumption 1,*

$$\mathbb{P}\left[\text{Good}\right] \geq 1 - \frac{\delta}{2\tau^*} \geq 1 - \frac{\delta}{2}.$$

Some notations are introduced first

- $\theta_{1:\infty} := (\theta^*_{j_t})_{t=1}^\infty$ is the latent vector sequence.
- $\hat{c}_l$ indicates the time step when we raise alarm for the $l^{\text{th}}$ changepoint.

According to the dynamics (see Dynamics 1) of the problem, at a changepoint $c_l \in \mathcal{C}$, the next latent vector will be sampled from $P_\theta$. Therefore, it may happen that $\theta^*_{j_{c_l}} = \theta^*_{j_{c_l-1}}$, i.e, this changepoint $c_l$ is hidden and these two consecutive stationary segments are observed as one stationary segment. In order to upper bound the errors made by Algorithm 3, we make the following assumption, which yields the worst-case scenario in terms of the error probability.

**Assumption 2.** *Given $\theta_{1:\infty}$ and the changepoint sequence $\mathcal{C}$, $\theta^*_{j_{c_l}} \neq \theta^*_{j_{c_{l+1}}}, \forall l \in \mathbb{N}$.*

In this section, we do not consider the randomness of $\theta_{1:\infty}$, i.e., we consider a realization of the sequence.[4] The results do not involve any parameter depending on the specific sequence and thus can be applied to any latent vector sequence.

## H.1 False Alarm Error

**Lemma G.1.** *For any $\delta_{\text{FAE}} \in (0,1)$, with $b \geq \frac{8d}{3w} \ln \frac{2}{\delta_{\text{FAE}}} + \sqrt{\left(\frac{8d}{3w} \ln \frac{2}{\delta_{\text{FAE}}}\right)^2 + \frac{24}{w} d \ln \frac{2}{\delta_{\text{FAE}}}}$, LCD makes no false alarm before the stopping time $\tau$ with probability at least*

$$\mathbb{P}[\text{FAE}^c] \geq 1 - \frac{\tau}{\gamma} K \delta_{\text{FAE}}.$$

*Proof of Lemma G.1.* Note that within a stationary segment of length $t$, the observations are i.i.d., and thus the time segments between two consecutive false alarm time $\{\hat{c}_l - \hat{c}_{l-1}\}_{l=1}^t$ (where $\hat{c}_0 = 0$) are i.i.d. i.e., it is a renewal process. Thus, it is sufficient to bound the error probability under the stationary case, i.e. $\mathcal{C} = \emptyset$ and $\theta_{1:\infty} = (\theta^*_{j_t} = \theta^*_{j_1})_{t=1}^\infty$.

Assume that we are under a stationary segment of length $t$. We wish to upper bound the error $\mathbb{P}[\text{FAE}] = \mathbb{P}[\hat{c}_1 \leq t]$. This is given by Lemma H.1.

**Lemma H.1.** *Given $w$ CD samples from the same context $[(\tilde{x}_s, Y_{s,\tilde{x}_s})]_{s=1}^w$ in a CD phase at time step $t$, Algorithm 3 makes a false alarm error with probability upper bounded by $K\delta_{\text{FAE}}$, i.e.*

$$\mathbb{P}\left[\text{FAE}_t\right] \leq K\delta_{\text{FAE}}$$

*if $b \geq \frac{8d}{3w} \ln \frac{2}{\delta_{\text{FAE}}} + \sqrt{\left(\frac{8d}{3w} \ln \frac{2}{\delta_{\text{FAE}}}\right)^2 + \frac{24}{w} d \ln \frac{2}{\delta_{\text{FAE}}}}$.*

Assume $\gamma$ divides $t$. In the case of a stationary segment, the input samples to Algorithm 3 are $\left\{[(x_{s\gamma}, Y_{s\gamma,x_{s\gamma}})]_{s=k}^{k+w-1}\right\}_{k=1}^{\frac{t}{\gamma}}$ if no changing alarm is raised (We ignore the initialization at Line 5 in Algorithm 1 for simplicity). Denote $\Gamma_i(t) := \{k\gamma : k \equiv i \mod w, k\gamma \leq t\}, i = 0, 1, \ldots, w-1$. Given any $i = 0, \ldots, w-1$, for any $k_1 \neq k_2 \in \Gamma_i(t)$, the two list of samples, $[(x_{s\gamma}, Y_{s\gamma,x_{s\gamma}})]_{s=k_1}^{k_1+w-1}$ and $[(x_{s\gamma}, Y_{s\gamma,x_{s\gamma}})]_{s=k_2}^{k_2+w-1}$, do not overlap (thus, they are independent). Therefore, $\mathbb{P}[k] := \mathbb{P}[\hat{c}_1 =$

---

[4]This can also be taken as, we are conditioning on $\theta_{1:\infty}$, as the realization of the sequence is independent from the behavior of any agent/algorithm.

$k\gamma + i, \hat{c}_1 \in \Gamma_i(t)]$ is a geometric distribution with parameter upper bounded by $K\delta_{FAE}$. We have

$$\mathbb{P}\left[\hat{c}_1 \in \Gamma_i(t)\right] \leq 1 - (1 - K\delta_{\mathrm{FAE}})^{|\Gamma_i(t)|}.$$

Hence, the cumulative false alarm error is

$$
\begin{aligned}
\mathbb{P}[\mathrm{FAE}] = \mathbb{P}\left[\hat{c}_1 \leq t\right] \\
= \sum_{i=0}^{w-1}\left[1 - (1 - K\delta_{\mathrm{FAE}})^{|\Gamma_i(t)|}\right] \\
\leq \sum_{i=0}^{w-1} |\Gamma_i(t)| K\delta_{\mathrm{FAE}} \\
\leq \frac{t}{\gamma} K\delta_{\mathrm{FAE}}.
\end{aligned}
$$

$\square$

*Proof of Lemma H.1.* According to Algorithm 3, given $w$ CD samples from the same context $[(\tilde{x}_s, Y_{s,\tilde{x}_s})]_{s=1}^w$ and $x \in \mathcal{X}$, we need to bound

$$\mathbb{P}\left[\left|\sum_{s=1}^{\frac{w}{2}} x^\top A(\lambda^*)^{-1} \tilde{x}_s Y_{s,\tilde{x}_s} - \sum_{s=\frac{w}{2}+1}^{w} x^\top A(\lambda^*)^{-1} \tilde{x}_s Y_{s,\tilde{x}_s}\right| \geq \frac{w}{2} b\right].$$

The proof is similar to Lemma E.1. For any $s \in [\frac{w}{2}]$ and $\tilde{s} = s + \frac{w}{2}$

$$\mathbb{E}\left[x^\top A(\lambda^*)^{-1} \tilde{x}_s Y_{s,\tilde{x}_s} - x^\top A(\lambda^*)^{-1} \tilde{x}_{\tilde{s}} Y_{\tilde{s},\tilde{x}_{\tilde{s}}}\right] = 0$$

and

$$
\begin{aligned}
&\left|x^\top A(\lambda^*)^{-1} \tilde{x}_s Y_{s,\tilde{x}_s} - x^\top A(\lambda^*)^{-1} \tilde{x}_{\tilde{s}} Y_{\tilde{s},\tilde{x}_{\tilde{s}}}\right| \\
&\leq \left|x^\top A(\lambda^*)^{-1} \tilde{x}_s \tilde{x}_s^\top \theta_{j_s}^*\right| + \left|x^\top A(\lambda^*)^{-1} \tilde{x}_s \eta_s\right| + \left|x^\top A(\lambda^*)^{-1} \tilde{x}_{\tilde{s}} \tilde{x}_{\tilde{s}}^\top \theta_{j_{\tilde{s}}}^*\right| + \left|x^\top A(\lambda^*)^{-1} \tilde{x}_{\tilde{s}} \eta_{\tilde{s}}\right| \\
&\leq 4d.
\end{aligned}
$$

By making use of (E.4),

$$
\begin{aligned}
&\mathbb{E}\left[\left(x^\top A(\lambda^*)^{-1} \tilde{x}_s Y_{s,\tilde{x}_s} - x^\top A(\lambda^*)^{-1} \tilde{x}_{\tilde{s}} Y_{\tilde{s},\tilde{x}_{\tilde{s}}}\right)^2\right] \\
&= \mathbb{E}\left[\left(x^\top A(\lambda^*)^{-1} \tilde{x}_s Y_{s,\tilde{x}_s}\right)^2\right] + \mathbb{E}\left[\left(x^\top A(\lambda^*)^{-1} \tilde{x}_{\tilde{s}} Y_{\tilde{s},\tilde{x}_{\tilde{s}}}\right)^2\right] \\
&\quad - 2\mathbb{E}\left[x^\top A(\lambda^*)^{-1} \tilde{x}_s Y_{s,\tilde{x}_s} \cdot x^\top A(\lambda^*)^{-1} \tilde{x}_{\tilde{s}} Y_{\tilde{s},\tilde{x}_{\tilde{s}}}\right] \\
&\leq 4d + 2\mathbb{E}\left[\left|x^\top \theta_{j_s}^* \cdot x^\top \theta_{j_{\tilde{s}}}^*\right|\right] \\
&\leq 6d.
\end{aligned}
$$

According to the Bernstein's inequality,

$$\mathbb{P}\left[\left|\sum_{s=1}^{\frac{w}{2}} \left(x^\top A(\lambda^*)^{-1} \tilde{x}_s Y_{s,\tilde{x}_s} - x^\top A(\lambda^*)^{-1} \tilde{x}_{s+\frac{w}{2}} Y_{s+\frac{w}{2},\tilde{x}_{s+\frac{w}{2}}}\right)\right| \geq \frac{w}{2}\epsilon\right]$$

$$\leq 2\exp\left(-\frac{\frac{1}{2}\left(\frac{w}{2}\epsilon\right)^2}{\frac{w}{2} \cdot 6d + \frac{4d}{3}\frac{w}{2}\epsilon}\right)$$

$$= 2\exp\left(-\frac{\frac{w}{2}\epsilon^2}{12d + \frac{8d}{3}\epsilon}\right).$$

In order to upper bound the error probability by $\delta_{\mathrm{FAE}}$, we need

$$2\exp\left(-\frac{\frac{w}{2}\epsilon^2}{12d + \frac{8d}{3}\epsilon}\right) \leq \delta_{FAE}$$

$$\Rightarrow \quad \epsilon \geq \frac{4d}{3 \cdot \frac{w}{2}} \ln \frac{2}{\delta_{\text{FAE}}} + \sqrt{\left(\frac{4d}{3 \cdot \frac{w}{2}} \ln \frac{2}{\delta_{\text{FAE}}}\right)^2 + 12 \cdot \frac{2}{w} d \ln \frac{2}{\delta_{\text{FAE}}}}.$$

By the choice of $b \geq \frac{8d}{3w} \ln \frac{2}{\delta_{\text{FAE}}} + \sqrt{\left(\frac{8d}{3w} \ln \frac{2}{\delta_{\text{FAE}}}\right)^2 + \frac{24}{w} d \ln \frac{2}{\delta_{\text{FAE}}}}$, we have

$$\mathbb{P}\left[\left|\frac{2}{w} \sum_{s=1}^{\frac{w}{2}} x^\top A(\lambda^*)^{-1} \tilde{x}_s Y_{s,\tilde{x}_s} - \frac{2}{w} \sum_{s=\frac{w}{2}+1}^{w} x^\top A(\lambda^*)^{-1} \tilde{x}_s Y_{s,\tilde{x}_s}\right| \geq b\right] \leq \delta_{\text{FAE}}.$$

A union bound over the $K$ arms yields the final result. $\qquad \square$

## H.2 Failed Alarm

**Lemma G.2.** *Conditional on* $\text{FAE}^c$*, for any* $\delta_{\text{FA}} \in (0,1)$*, if* $\frac{\Delta_c - b}{2} \geq \frac{d}{w} \ln \frac{2}{\delta_{\text{FA}}} + \sqrt{\left(\frac{d}{w} \ln \frac{2}{\delta_{\text{FA}}}\right)^2 + \frac{4d}{w} \ln \frac{2}{\delta_{\text{FA}}}}$,

$$\mathbb{P}\left[c_l \leq \hat{c}_l \leq c_l + \frac{w\gamma}{2}\Big|\hat{c}_l \geq c_l\right] \geq 1 - \delta_{\text{FA}}.$$

*In addition,* $\mathbb{P}\left[\text{FA}\big|\text{FAE}^c\right] \leq l_\tau \delta_{\text{FA}}$.

*Proof of Lemma G.2.* Conditioned on $\text{FAE}^c$, the detection at the changepoints is independent. Thus we can assume there is only one changepoint $c_1$ within a certain number of consecutive time steps.

Algorithm 3 is given $w$ CD samples which are collected under two different context. Without loss of generality, we assume the sample selected at time $c_1$ is among the CD samples (otherwise, we can regard the time step of the first sample from the second context as $c_1$). We wish to bound

$$\mathbb{P}\left[c_1 \leq \hat{c}_1 \leq c_1 + \frac{\gamma w}{2}\Big|\hat{c}_1 \geq c_1\right]$$

which is the probability of the event that, after a changepoint occurs, a changing alarm will be raised within $\frac{w\gamma}{2}$ time steps (or $\frac{w}{2}$ CD samples). Here, $\hat{c}_1 \geq c_1$ can be guaranteed as we are conditioning on that there is no false alarm error ($\text{FAE}^c$).

The event $c_1 \leq \hat{c}_1 \leq c_1 + \frac{\gamma w}{2}$ indicates, at least one of the CD sample list in

$$\left\{\left[\left(x_{c_1+(s-k)\gamma}, Y_{c_1+(s-k)\gamma, x_{c_1+(s-k)\gamma}}\right)\right]_{s=1}^{w}\right\}_{k=\frac{w}{2}+1}^{w}$$

will trigger the changing alarm in Algorithm 3. In particular, in Lemma H.2, we consider the failed arm probability when the CD samples are composed by exactly half samples from each contexts,

$$\left[\left(x_{c_1+(s-1-\frac{w}{2})\gamma}, Y_{c_1+(s-1-\frac{w}{2})\gamma, x_{c_1+(s-1-\frac{w}{2})\gamma}}\right)\right]_{s=1}^{w}.$$

**Lemma H.2.** *Given $w$ CD samples from the two different contexts $[(\tilde{x}_s, Y_{s,\tilde{x}_s})]_{s=1}^{w}$ in a CD phase, where $[(\tilde{x}_s, Y_{s,\tilde{x}_s})]_{s=1}^{\frac{w}{2}}$ is from latent vector $\theta_j^*$ and $[(\tilde{x}_s, Y_{s,\tilde{x}_s})]_{s=\frac{w}{2}+1}^{w}$ is from latent vector $\theta_{\tilde{j}}^*$, Algorithm 3 raises a changing alarm with probability lower bounded by*

$$1 - \delta_{\text{FA}}$$

*if* $\frac{\Delta_c - b}{2} \geq \frac{d}{w} \ln \frac{2}{\delta_{\text{FA}}} + \sqrt{\left(\frac{d}{w} \ln \frac{2}{\delta_{\text{FA}}}\right)^2 + \frac{4d}{w} \ln \frac{2}{\delta_{\text{FA}}}}$ *where* $\Delta_c := \max_{x \in \mathcal{X}} |x^\top(\theta_j^* - \theta_{\tilde{j}}^*)|$ *and it is assumed to be greater than $b$.*

By making use of Lemma H.2,

$$\mathbb{P}\left[c_1 \leq \hat{c}_1 \leq c_1 + \frac{w\gamma}{2}\Big|\hat{c}_1 \geq c_1\right]$$
$$= \mathbb{P}\left[\exists k \in \{\frac{w}{2}+1, \ldots, w\},\right.$$
$$\left.\text{LCD}\left(w, b, [(x_{c_1+(s-k)\gamma}, Y_{c_1+(s-k)\gamma, x_{c_1+(s-k)\gamma}})]_{s=1}^{w}\right) = \text{True}\Big|\hat{c}_1 \geq c_1\right]$$

$$\geq \mathbb{P}\left[\text{LCD}\left(w, b, [(x_{c_1+(s-1-\frac{w}{2})\gamma}, Y_{c_1+(s-1-\frac{w}{2})\gamma}, x_{c_1+(s-1-\frac{w}{2})\gamma})]_{s=1}^w\right) = \text{True}\Big|\hat{c}_1 \geq c_1\right]$$

$$\geq 1 - \delta_{\text{FA}}$$

where $\Delta_c = \max_{x \in \mathcal{X}} |x^\top(\theta_j^* - \theta_{\tilde{j}}^*)|$ and $\theta_j^*, \theta_{\tilde{j}}^*$ are the latent vectors. Hence,

$$\mathbb{P}\left[\text{FA}\big|\text{FAE}^c\right] = 1 - \mathbb{P}\left[\text{FA}^c\big|\text{FAE}^c\right] \leq 1 - (1 - \delta_{\text{FA}})^{l_\tau} \leq l_\tau \delta_{\text{FA}}$$

$\square$

*Proof of Lemma H.2.* According to the design of LCD, there exists an $x \in \mathcal{X}$,

$$\mathbb{P}\left[\left|\frac{2}{w}\sum_{s=1}^{\frac{w}{2}} x^\top A(\lambda^*)^{-1}\tilde{x}_s Y_{s,\tilde{x}_s} - \frac{2}{w}\sum_{s=\frac{w}{2}+1}^{w} x^\top A(\lambda^*)^{-1}\tilde{x}_s Y_{s,\tilde{x}_s}\right| \geq b\right]$$

$$\geq 1 - \mathbb{P}\left[\left|\sum_{s=1}^{\frac{w}{2}} x^\top A(\lambda^*)^{-1}\tilde{x}_s Y_{s,\tilde{x}_s} - \sum_{s=\frac{w}{2}+1}^{w} x^\top A(\lambda^*)^{-1}\tilde{x}_s Y_{s,\tilde{x}_s} - \frac{w}{2}x^\top(\theta_j^* - \theta_{\tilde{j}}^*)\right|\right.$$

$$\left.\geq \left|\frac{w}{2}b - \frac{w}{2}x^\top(\theta_j^* - \theta_{\tilde{j}}^*)\right|\right]$$

$$= 1 - \mathbb{P}\left[\left|\sum_{s=1}^{\frac{w}{2}} \left(x^\top A(\lambda^*)^{-1}\tilde{x}_s Y_{s,\tilde{x}_s} - x^\top\theta_j^*\right) - \sum_{s=\frac{w}{2}+1}^{w} \left(x^\top A(\lambda^*)^{-1}\tilde{x}_s Y_{s,\tilde{x}_s} - x^\top\theta_{\tilde{j}}^*\right)\right|\right.$$

$$\left.\geq \left|\frac{w}{2}b - \frac{w}{2}x^\top(\theta_j^* - \theta_{\tilde{j}}^*)\right|\right]$$

$$\geq 1 - \delta_{\text{FA}}$$

where the last inequality holds as $\frac{\Delta_c - b}{2} \geq \frac{d}{w}\ln\frac{2}{\delta_{\text{FA}}} + \sqrt{\left(\frac{d}{w}\ln\frac{2}{\delta_{\text{FA}}}\right)^2 + \frac{4d}{w}\ln\frac{2}{\delta_{\text{FA}}}}.$ $\square$

## H.3 Context Alignment

**Lemma G.3.** *Conditional on* $\text{FAE}^c$ *and* $\text{FA}^c$*, based on the conditions in Lemma G.1 and Lemma G.2,*
$$\mathbb{P}\left[\text{MI}|\text{FAE}^c, \text{FA}^c\right] \leq l_\tau\left[(N-1)\delta_{\text{FA}} + K\delta_{\text{FAE}}\right].$$

*Proof of Lemma G.3.* The error of the context alignment procedure can be derived from the FAE and FA analyses. Conditioned on $\text{FAE}^c$, $\text{FA}^c$ and that the previous $l-1$ contexts are correctly identified, i.e., $\cap_{k=1}^{l-1}\text{MI}_k^c$, we have the following statements.

Firstly, according to Lemma H.1, the change alarm at Line 4 in Algorithm 4 will not be triggered with probability at least $(1 - K\delta_{\text{FAE}})$ if the current context is context $j$. This error has been taken into account in the FAE (see Remark H.3).

Secondly, if the current context is not $j$ (which will occur at most $N-1$ times), a change alarm will be raised with probability at least $1 - \delta_{\text{FA}}$ by Lemma G.2.

Therefore, the error probability at $c_l$ is upper bounded by
$$\mathbb{P}\left[\text{MI}_l|\text{FAE}^c, \text{FA}^c, \cap_{k=1}^{l-1}\text{MI}_k^c\right] \leq (N-1)\delta_{\text{FA}} + K\delta_{\text{FAE}}$$
and by a union bound, the cumulative error probability bounded by
$$\mathbb{P}\left[\text{MI}|\text{FAE}^c, \text{FA}^c\right] = 1 - \mathbb{P}\left[\text{MI}^c|\text{FAE}^c, \text{FA}^c\right] \tag{H.1}$$
$$\leq 1 - (1 - (N-1)\delta_{\text{FA}} - K\delta_{\text{FAE}})^{l_\tau}$$
$$\leq l_\tau\left((N-1)\delta_{\text{FA}} + K\delta_{\text{FAE}}\right).$$

$\square$

**Remark H.3.** *Note that (1) we will only bound the* FAE *when the CD samples are from the same context; (2) in the analysis of* FAE*, we bound the error probability on the whole horizon up to time* $t$

*as we assume there is no changepoint. In particular, there will be unused and redundant $\frac{w}{2}K\delta_{\text{FAE}}$ errors budget for FAE at each changepoint, which accumulates to $l_\tau \frac{w}{2}K\delta_{\text{FAE}}$ before time step $\tau$. Therefore, the second error term in (H.1), $l_\tau K\delta_{\text{FAE}}$, can be covered by the unused $l_\tau \frac{w}{2}K\delta_{\text{FAE}}$ error budget from FAE, and thus it can be neglected in (H.1).*

*Proof of Lemma G.4.* According to Lemma G.1, G.2, G.3, Assumption 1 and Remark H.3, given a time $\tau$, the total failure probability is upper bounded by

$$\mathbb{P}\left[\text{Good}^c\right] = \mathbb{P}\left[\text{FAE} \cup \text{FA} \cup \text{MI}\right] \tag{H.2}$$
$$\leq \frac{\tau}{\gamma}K\delta_{\text{FAE}} + l_\tau \delta_{\text{FA}} + l_\tau \cdot (N-1)\delta_{\text{FA}}$$
$$= \frac{\tau}{\gamma}K\delta_{\text{FAE}} + l_\tau N\delta_{\text{FA}}.$$

In particular, when $\tau$ is upper bounded by $\tau^*$ in Line 2 of Algorithm 1

$$\tau \leq \tau^* = c_0 \frac{NL_{\max}}{\varepsilon^2} \ln \frac{N^2 K L_{\max}/\varepsilon^2}{\delta}$$

where $c_0 = 3 \cdot 6400 \ln 6400$. The number of changepoints till $\tau$ is upper bounded by $l_{\tau^*} \leq \frac{\tau^*}{L_{\min}}$.

By Assumption 1, when $b = \frac{8d}{3w} \ln \frac{2}{\delta_{\text{FAE}}} + \sqrt{\left(\frac{8d}{3w} \ln \frac{2}{\delta_{\text{FAE}}}\right)^2 + \frac{24}{w}d\ln \frac{2}{\delta_{\text{FAE}}}}$ and $\delta_{\text{FAE}} = \frac{\gamma\delta}{4(\tau^*)^2 K}$, the conditions of Lemma G.1 are met. And we can upper bound $\frac{\tau}{\gamma}K\delta_{\text{FAE}}$ by $\frac{\delta}{4\tau^*} \leq \frac{\delta}{4}$.

By making use of Assumption 1 and setting $\delta_{\text{FA}} = \frac{\delta}{4Nl_{\tau^*}\tau^*}$, we have $\delta_{\text{FA}} > \delta_{\text{FAE}}$ and

$$\frac{\Delta_c - b}{2} \geq \frac{b}{2} \geq \frac{4d}{3w} \ln \frac{2}{\delta_{\text{FAE}}} + \sqrt{\left(\frac{4d}{3w} \ln \frac{2}{\delta_{\text{FAE}}}\right)^2 + \frac{6}{w}d\ln \frac{2}{\delta_{\text{FAE}}}}$$
$$\geq \frac{d}{w} \ln \frac{2}{\delta_{\text{FA}}} + \sqrt{\left(\frac{d}{w} \ln \frac{2}{\delta_{\text{FA}}}\right)^2 + \frac{4d}{w} \ln \frac{2}{\delta_{\text{FA}}}}.$$

Thus, Lemma G.2 can be applied. $l_{\tau^*} N\delta_{\text{FA}}$ can be upper bounded by $\frac{\delta}{4\tau^*} \leq \frac{\delta}{4}$.

Therefore, according to (H.2), $\mathbb{P}[\text{Good}] \geq 1 - \frac{\delta}{2\tau^*} \geq 1 - \frac{\delta}{2}$.

$\square$

# I   Analysis of PS$\varepsilon$BAI: Estimation Error

Lemma G.5 is proved in this section. We will upper bound these three error terms (see (I.1)): VE error in App I.1, DE error in App I.2, RE error in App I.3, and finally we will prove Lemma G.5 which upper bounds the failure probability of CI at the end of this section.

**Lemma G.5.** *If* $T_t \geq \frac{2L_{\max}}{9} \ln \frac{2}{\delta_{d,T_t}}$,
$$\mathbb{P}\left[\text{CI}_t\big|\text{Good}\right] \geq 1 - \left(K\delta_{v,T_t} + N\delta_{d,T_t} + KN\delta_{m,T_t}\right).$$
*In addition,*
$$\mathbb{P}\left[\text{CI}\big|\text{Good}\right] \geq 1 - \frac{\delta}{2}.$$

Given any two arms $x, \tilde{x} \in \mathcal{X}$, by the triangular inequality, the deviation between $\hat{\Delta}_t(x, \tilde{x})$ and $\Delta(x, \tilde{x})$ can be upper bounded by three terms: the Vector-Estimation Error (VE) term, the Distribution-Estimation Error (DE) term, and the Residual Estimation Error (RE) term:

$$|\hat{\Delta}_t(x, \tilde{x}) - \Delta(x, \tilde{x})| \tag{I.1}$$
$$= |(x - \tilde{x})^\top \hat{\Theta}_t \hat{\mathbf{p}}_t - (x - \tilde{x})^\top \Theta \mathbf{p}|$$

$$\leq \underbrace{\Big| \sum_{j=1}^{N} \left( \hat{\Delta}_{t,j}(x,\tilde{x}) - \Delta_j(x,\tilde{x}) \right) \hat{p}_{t,j} \Big|}_{\text{VE term}} + \underbrace{\Big| \sum_{j=1}^{N} \hat{\Delta}_{t,j}^{\text{clip}_2}(x,\tilde{x})(\hat{p}_{t,j} - p_j) \Big|}_{\text{DE term}}$$

$$+ \underbrace{\Big| \sum_{j=1}^{N} \left( \Delta_j(x,\tilde{x}) - \hat{\Delta}_{t,j}^{\text{clip}_2}(x,\tilde{x}) \right) (\hat{p}_{t,j} - p_j) \Big|}_{\text{RE term}},$$

where $a^{\text{clip}_2} := \text{clip}_2(a) := \min\{\max\{a, -2\}, 2\}$ is a shorthand notation for the value of $a$ that is clipped to the interval $[-2, 2]$. The reason the value $\hat{\Delta}_{t,j}(x, \tilde{x})$ is clipped is the ground truth $|\Delta_{t,j}(x, \tilde{x})| \leq 2$.

Recall the event CI

$$\text{CI}_t := \{|\hat{\Delta}_t(x,\tilde{x}) - \Delta(x,\tilde{x})| \leq \rho_t(x,\tilde{x}), \forall x, \tilde{x} \in \mathcal{X}\},$$

$$\text{CI} := \bigcap_t \text{CI}_t,$$

and the confidence radius

$$\rho_t(x,\tilde{x}) := 2(\alpha_t + \xi_t) + \sum_{j=1}^{N} \beta_{t,j} |\hat{\Delta}_{t,j}^{\text{clip}_2}(x,\tilde{x}) + \zeta_t(x,\tilde{x})|, \tag{I.2}$$

where $\quad \alpha_t := 5\sqrt{\dfrac{d}{T_t} \ln \dfrac{2}{\delta_{v,T_t}}}, \quad \beta_{t,j} := \dfrac{5}{2} \sqrt{\dfrac{2\phi_{t,j} L_{\max}}{T_t} \ln \dfrac{2}{\delta_{d,T_t}}},$

$$\xi_t := 25\sqrt{2} \dfrac{N L_{\max}}{T_t} \ln \dfrac{2}{\delta_{m,T_t}}, \quad \phi_{t,j} := \min\left\{ 4 \max\left\{ \hat{p}_{t,j}, \dfrac{25}{4} \dfrac{L_{\max}}{T_t} \ln \dfrac{2}{\delta_{d,T_t}} \right\}, \dfrac{1}{4} \right\},$$

and $\zeta_t(x,\tilde{x}) \in \mathbb{R}$ can be any value. In particular, it can be the value that minimizes $\sum_{j=1}^{N} \beta_{t,j} |\hat{\Delta}_{t,j}(x,\tilde{x}) + \zeta_t(x,\tilde{x})|$ or it can be taken as $\varepsilon$. For simplicity, we will take $\zeta_t(x,\tilde{x}) = \arg\min_{\zeta_t(x,\tilde{x}) \in \mathbb{R}} \sum_{j=1}^{N} \beta_{t,j} (\hat{\Delta}_{t,j}(x,\tilde{x}) + \zeta_t(x,\tilde{x}))^2 = -\dfrac{\sum_{j=1}^{N} \beta_{t,j} \hat{\Delta}_{t,j}(x,\tilde{x})}{\sum_{j=1}^{N} \beta_{t,j}}$ for a simple and effective analytic expression in the algorithm.

### I.1 Vector-Estimation Error

For the VE term $\Big| \sum_{j=1}^{N} \left( \hat{\Delta}_{t,j}(x,\tilde{x}) - \Delta_j(x,\tilde{x}) \right) \hat{p}_{t,j} \Big|$, note that

$$\Big| \sum_{j=1}^{N} \left( \hat{\Delta}_{t,j}(x,\tilde{x}) - \Delta_j(x,\tilde{x}) \right) \hat{p}_{t,j} \Big| \leq |x^\top (\hat{\Theta}_t - \Theta)\hat{p}_t| + |\tilde{x}^\top (\hat{\Theta}_t - \Theta)\hat{p}_t|$$

**Lemma I.1.** *Given $x \in \mathcal{X}$ and $T_t \geq \frac{d}{4} \ln \frac{2}{\delta_{v,T_t}}$,*

$$\mathbb{P}\left[ |x^\top (\hat{\Theta}_t - \Theta)\hat{\mathbf{p}}_t| \geq \alpha_t \big| \text{Good} \right] \leq \delta_{v,T_t}$$

*Proof of Lemma I.1.* By the definition of the estimators in (3.2),

$$x^\top (\hat{\Theta}_t - \Theta)\hat{\mathbf{p}}_t = \sum_{j=1}^{N} x^\top (\hat{\theta}_{t,j} - \theta_j^*)\hat{\mathbf{p}}_{t,j}$$

$$= \sum_{j=1}^{N} x^\top \left( \dfrac{1}{T_{t,j}} \sum_{s \in \mathcal{T}_{t,j}} A(\lambda^*)^{-1} x_s Y_{s,x_s} - \theta_j^* \right) \dfrac{T_{t,j}}{T_t}$$

$$= \dfrac{1}{T_t} \sum_{j=1}^{N} \sum_{s \in \mathcal{T}_{t,j}} x^\top A(\lambda^*)^{-1} x_s Y_{s,x_s} - x^\top \theta_j^*$$

By Lemma E.1,
$$\mathbb{P}\left[|x^\top(\hat{\Theta}_t - \Theta)\hat{\mathbf{p}}_t| \geq \alpha_t \big| \theta_{1:\infty}, \text{Good}\right] \leq \delta_{v,T_t}$$
By the property of conditional probability, we have the desired result. $\qquad\square$

Therefore, conditional on Good, with probability at least $1 - K\delta_{v,T_t}$, VE $\leq 2\alpha_t$ for any $x, \tilde{x} \in \mathcal{X}$.

## I.2  Distribution-Estimation Error

**Lemma I.2.** *Given* Good *and* $T_t \geq \frac{2L_{\max}}{9} \ln \frac{2}{\delta_{d,T_t}}$, *for any* $j \in [N]$,
$$\mathbb{P}\left[|\hat{p}_{t,j} - p_j| \geq \beta_{t,j} \big| \text{Good}\right] \leq \delta_{d,T_t}.$$
*Additionally, with probability* $1 - N\delta_{d,T_t}$,
$$\left|\sum_{j=1}^N \hat{\Delta}_{t,j}^{\text{clip}_2}(x, \tilde{x})(\hat{p}_{t,j} - p_j)\right| \leq \sum_{j=1}^N \beta_{t,j}|\hat{\Delta}_{t,j}^{\text{clip}_2}(x, \tilde{x}) + \zeta_t(x, \tilde{x})|$$
*where* $\zeta_t(x, \tilde{x})$ *can be any value.*

*Proof of Lemma I.2.* Given any $j \in [N]$, and the stationary segment $l$, we denote $X_{j,l} := \hat{L}_l\mathbb{1}\{\theta_{j_l}^* = \theta_j^*\} - p_j\hat{L}_l$, where $\hat{L}_l$ is the total length of the Exp phases in the $l$th stationary segment. Note that $\mathbb{E}[X_{j,l}|\{\hat{L}_l\}_{l=1}^{l_t}, \text{Good}] = 0, |X_{j,l}| \leq L_{\max}$ *a.s.* and $\text{Var}[X_{j,l}|\{\hat{L}_l\}_{l=1}^{l_t}, \text{Good}] = p_j(1 - p_j)\hat{L}_l^2 \leq p_j(1 - p_j)\hat{L}_l L_{\max}$. By Bernstein's inequality,
$$\mathbb{P}\left[|\sum_{l=1}^{l_t} X_{j,l}| \geq \epsilon \big| \{\hat{L}_l\}_{l=1}^{l_t}, \text{Good}\right] \leq 2\exp\left(-\frac{\epsilon^2}{2\sum_{l=1}^{l_t}\text{Var}[X_{j,l}|\text{Good}] + \frac{2}{3}L_{\max}\epsilon}\right)$$
$$\Rightarrow \quad \mathbb{P}\left[|\frac{1}{T_t}\sum_{l=1}^{l_t} X_{j,l}| \geq \epsilon \big| \{\hat{L}_l\}_{l=1}^{l_t}, \text{Good}\right] \leq 2\exp\left(-\frac{T_t^2\epsilon^2}{2\sum_{l=1}^{l_t}\text{Var}[X_{j,l}|\text{Good}] + \frac{2}{3}L_{\max}T_t\epsilon}\right)$$

As $T_{t,j} = \sum_{l=1}^{l_t}\hat{L}_l\mathbb{1}\{\theta_{j_l}^* = \theta_j^*\}, T_t = \sum_{l=1}^{l_t}\hat{L}_l$ and $\sum_{l=1}^{l_t}\hat{L}_l = T_t$, we have
$$\mathbb{P}\left[|\hat{p}_{t,j} - p_j| \geq \epsilon \big| \{\hat{L}_l\}_{l=1}^{l_t}, \text{Good}\right] \leq 2\exp\left(-\frac{T_t^2\epsilon^2}{2\sum_{l=1}^{l_t}\text{Var}[X_{j,l}|\{\hat{L}_l\}_{l=1}^{l_t}, \text{Good}] + \frac{2}{3}L_{\max}T_t\epsilon}\right)$$
$$\leq 2\exp\left(-\frac{T_t^2\epsilon^2}{2\sum_{l=1}^{l_t}p_j(1 - p_j)\hat{L}_l L_{\max} + \frac{2}{3}L_{\max}T_t\epsilon}\right)$$
$$\leq 2\exp\left(-\frac{T_t\epsilon^2}{2p_j(1 - p_j)L_{\max} + \frac{2}{3}L_{\max}\epsilon}\right)$$
As the last bound is independent of $\{\hat{L}_l\}_{l=1}^{l_t}$, we have
$$\mathbb{P}\left[|\hat{p}_{t,j} - p_j| \geq \epsilon \big| \text{Good}\right] \leq 2\exp\left(-\frac{T_t\epsilon^2}{2p_j(1 - p_j)L_{\max} + \frac{2}{3}L_{\max}\epsilon}\right)$$
If we want to upper bound the above by $\delta_{d,T_t} \in (0, 1)$, i.e.,
$$2\exp\left(-\frac{T_t\epsilon^2}{2p_j(1 - p_j)L_{\max} + \frac{2}{3}L_{\max}\epsilon}\right) \leq \delta_{d,T_t}$$
we require
$$\epsilon \geq \frac{\frac{1}{3}L_{\max}\ln\frac{2}{\delta_{d,T_t}} + \sqrt{\left(\frac{1}{3}L_{\max}\ln\frac{2}{\delta_{d,T_t}}\right)^2 + 2T_tp_j(1 - p_j)L_{\max}\ln\frac{2}{\delta_{d,T_t}}}}{T_t} \qquad\text{(I.3)}$$

In particular,

$$\epsilon = \tilde{\beta}_{t,j} := \frac{5}{2}\max\left\{\frac{L_{\max}}{3T_t}\ln\frac{2}{\delta_{d,T_t}}, \sqrt{\frac{2p_j(1-p_j)L_{\max}}{T_t}\ln\frac{2}{\delta_{d,T_t}}}\right\}$$

satisfies the condition, we have

$$\mathbb{P}\left[|\hat{p}_{t,j} - p_j| \geq \tilde{\beta}_{t,j}\big|\text{Good}\right] \leq \delta_{d,T_t}. \tag{I.4}$$

A problem here is we do not have access to $p_j$ during the dynamics, therefore, we adopt Lemma K.1 to further upper bound $\tilde{\beta}_{t,j}$ by $\beta_{t,j}$.

**Lemma K.1.** *Given any $j \in [N]$ and $T_t \geq \frac{2L_{\max}}{9}\ln\frac{2}{\delta_{d,T_t}}$, $\tilde{\beta}_{t,j} \leq \beta_{t,j}$.*

Therefore,

$$\mathbb{P}\left[|\hat{p}_{t,j} - p_j| \geq \beta_{t,j}\big|\text{Good}\right] \leq \delta_{d,T_t}.$$

In addition, with probability at least $1 - N\delta_{d,T_t}$, we can upper bound DE term as:

$$\left|\sum_{j=1}^N \hat{\Delta}_{t,j}^{\text{clip}_2}(x,\tilde{x})(\hat{p}_{t,j} - p_j)\right| = \left|\sum_{j=1}^N \left(\hat{\Delta}_{t,j}^{\text{clip}_2}(x,\tilde{x}) + \zeta_t(x,\tilde{x})\right)(\hat{p}_{t,j} - p_j)\right| \tag{I.5}$$

$$\leq \sum_{j=1}^N \beta_{t,j}\left|\hat{\Delta}_{t,j}^{\text{clip}_2}(x,\tilde{x}) + \zeta_t(x,\tilde{x})\right|$$

where $\zeta_t(x,\tilde{x}) \in \mathbb{R}$ can be any value. $\qquad\square$

## I.3 Residual-Estimation Error

RE is composed by the product of two deviations, i.e., $\left(\Delta_j(x,\tilde{x}) - \hat{\Delta}_{t,j}^{\text{clip}_2}(x,\tilde{x})\right)$ and $(\hat{p}_{t,j} - p_j)$, as the time step $t$ becomes large, we expect it will converge to zero fast. Thus, it is sufficient to have a coarse estimation of it.

**Lemma I.3.** *For any $x \in \mathcal{X}$, conditional on that $|\hat{p}_{t,j} - p_j| \leq \beta_{t,j}, \forall j \in [N]$,*

$$\mathbb{P}\left[\left|\sum_{j=1}^N \left(\Delta_j(x,\tilde{x}) - \hat{\Delta}_{t,j}^{\text{clip}_2}(x,\tilde{x})\right)(\hat{p}_{t,j} - p_j)\right| \geq \xi_t\bigg|\text{Good}\right] \leq N\delta_{m,T_t}$$

*where $\xi_t := 25\sqrt{2}\frac{NL_{\max}}{T_t}\ln\frac{2}{\delta_{m,T_t}}$.*

*Proof of Lemma I.3.* According to Lemma E.1, for any $x \in \mathcal{X}, j \in [N]$ we have

$$\mathbb{P}\left[|x^\top(\theta_j^* - \hat{\theta}_{t,j})| \geq 5\sqrt{\frac{d}{T_{t,j}}\ln\frac{2}{\delta_{m,T_t}}}\bigg|\text{Good}\right] \leq \delta_{m,T_t} \tag{I.6}$$

if $T_{t,j} \geq \frac{d}{4}\ln\frac{2}{\delta_{m,T_t}}$. Note the fact that $\Delta_j(x,\tilde{x}) \in [-2,2]$, hence,

$$\left|\Delta_j(x,\tilde{x}) - \hat{\Delta}_{t,j}^{\text{clip}_2}(x,\tilde{x})\right| \leq \left|\Delta_j(x,\tilde{x}) - \hat{\Delta}_{t,j}(x,\tilde{x})\right| \leq |x^\top(\theta_j^* - \hat{\theta}_{t,j})| + |\tilde{x}^\top(\theta_j^* - \hat{\theta}_{t,j})|. \tag{I.7}$$

And $|x^\top(\theta_j^* - \hat{\theta}_{t,j})| \leq 3d$ with probability 1. Denote

$$\psi_{t,j,1} := \max\left\{\hat{p}_{t,j}, \frac{d}{4T_t}\ln\frac{2}{\delta_{m,T_t}}\right\}, \quad \psi_{t,j,2} := \max\left\{\hat{p}_{t,j}, \frac{25}{4}\frac{L_{\max}}{T_t}\ln\frac{2}{\delta_{d,T_t}}\right\}. \tag{I.8}$$

We have[5]

$$\left| \sum_{j=1}^{N} \left( \Delta_j(x, \tilde{x}) - \hat{\Delta}_{t,j}^{\mathrm{clip}_2}(x, \tilde{x}) \right) (\hat{p}_{t,j} - p_j) \right|$$

$$\leq \sum_{j: \psi_{t,j,1} > \hat{p}_{t,j}} \left| \left( \Delta_j(x, \tilde{x}) - \hat{\Delta}_{t,j}^{\mathrm{clip}_2}(x, \tilde{x}) \right) \beta_{t,j} \right| + \sum_{\substack{j: \psi_{t,j,1} = \hat{p}_{t,j}, \\ \psi_{t,j,2} > \hat{p}_{t,j}}} \left| \left( \Delta_j(x, \tilde{x}) - \hat{\Delta}_{t,j,2}^{\mathrm{clip}_2}(x, \tilde{x}) \right) \beta_{t,j} \right|$$

$$+ \sum_{j: \psi_{t,j,2} = \hat{p}_{t,j}} \left| \left( \Delta_j(x, \tilde{x}) - \hat{\Delta}_{t,j}^{\mathrm{clip}_2}(x, \tilde{x}) \right) \beta_{t,j} \right|$$

$$\overset{(a)}{\leq} \sum_{j: \psi_{t,j,1} > \hat{p}_{t,j}} \left| \left( \Delta_j(x, \tilde{x}) - \hat{\Delta}_{t,j}^{\mathrm{clip}_2}(x, \tilde{x}) \right) \beta_{t,j} \right| + \sum_{\substack{j: \psi_{t,j,1} = \hat{p}_{t,j}, \\ \psi_{t,j,2} > \hat{p}_{t,j}}} \left| \left( \Delta_j(x, \tilde{x}) - \hat{\Delta}_{t,j,2}^{\mathrm{clip}_2}(x, \tilde{x}) \right) \beta_{t,j} \right|$$

$$+ \sum_{j: \psi_{t,j,2} = \hat{p}_{t,j}} |x^\top (\theta_j^* - \hat{\theta}_{t,j}) \beta_{t,j}| + |\tilde{x}^\top (\theta_j^* - \hat{\theta}_{t,j}) \beta_{t,j}| \tag{I.9}$$

$$\overset{(b)}{\leq} \sum_{j: \psi_{t,j,1} > \hat{p}_{t,j}} 4 \cdot \frac{5}{2} \cdot \frac{5\sqrt{2}}{2} \frac{L_{\max}}{T_t} \ln \frac{2}{\delta_{d,T_t}}$$

$$+ \sum_{\substack{j: \psi_{t,j,1} = \hat{p}_{t,j}, \\ \psi_{t,j,2} > \hat{p}_{t,j}}} \min \left\{ 4, 2 \cdot 5 \sqrt{\frac{d}{T_{t,j}} \ln \frac{2}{\delta_{m,T_t}}} \right\} \cdot \frac{5}{2} \cdot \frac{5\sqrt{2}}{2} \frac{L_{\max}}{T_t} \ln \frac{2}{\delta_{d,T_t}}$$

$$+ \sum_{j: \psi_{t,j,3} = \hat{p}_{t,j}} 2 \cdot 5 \sqrt{\frac{d}{T_{t,j}} \ln \frac{2}{\delta_{m,T_t}}} \cdot \frac{5}{2} \sqrt{\frac{8 \hat{p}_{t,j} L_{\max}}{T_t} \ln \frac{2}{\delta_{d,T_t}}}$$

$$\leq \sum_{j: \psi_{t,j,1} > \hat{p}_{t,j}} 25 \sqrt{2} \frac{L_{\max}}{T_t} \ln \frac{2}{\delta_{d,T_t}}$$

$$+ \sum_{\substack{j: \psi_{t,j,1} = \hat{p}_{t,j}, \\ \psi_{t,j,2} > \hat{p}_{t,j}}} \min \left\{ 4, 10 \sqrt{\frac{d}{T_{t,j}} \ln \frac{2}{\delta_{m,T_t}}} \right\} \cdot \frac{25\sqrt{2}}{4} \frac{L_{\max}}{T_t} \ln \frac{2}{\delta_{d,T_t}}$$

$$+ \sum_{j: \psi_{t,j,2} = \hat{p}_{t,j}} 50 \sqrt{2} \frac{\sqrt{d L_{\max}}}{T_t} \ln \frac{2}{\delta_{m,T_t}} \tag{I.10}$$

$$\leq 50 \sqrt{2} \frac{N L_{\max}}{T_t} \ln \frac{2}{\delta_{m,T_t}} = 2 \xi_t$$

where $(a)$ adopts (I.7) and $(b)$ is obtained by the following derivations:

(1) $\psi_{t,j,1} > \hat{p}_{t,j}$ indicates $\hat{p}_{t,j} < \frac{d}{4 T_t} \ln \frac{2}{\delta_{m,T_t}} < \frac{25}{4} \frac{L_{\max}}{T_t} \ln \frac{2}{\delta_{d,T_t}}$, and thus $\beta_{t,j} \leq \frac{5}{2} \cdot \frac{5\sqrt{2}}{2} \frac{L_{\max}}{T_t} \ln \frac{2}{\delta_{d,T_t}}$. In addition,[6]

$$\left| \Delta_j(x, \tilde{x}) - \hat{\Delta}_{t,j}^{\mathrm{clip}_2}(x, \tilde{x}) \right| \leq 4.$$

(2) $\psi_{t,j,1} = \hat{p}_{t,j}$ indicates $\hat{p}_{t,j} \leq \frac{d}{4 T_t} \ln \frac{2}{\delta_{m,T_t}}$ and $\psi_{t,j,2} > \hat{p}_{t,j}$ implies $\hat{p}_{t,j} < \frac{25}{4} \frac{L_{\max}}{T_t} \ln \frac{2}{\delta_{d,T_t}}$, thus (I.6) can be applied and $\beta_{t,j} \leq \frac{5}{2} \cdot \frac{5\sqrt{2}}{2} \frac{L_{\max}}{T_t} \ln \frac{2}{\delta_{d,T_t}}$.

---

[5]The last inequality can be loose. When each context has been visited at least once, the first summation in (I.10) can be ignored. For empirical performance, (I.10) should be used, while $\xi$ is adopted for theoretical guarantees.

[6]This demonstrates the usefulness of the clipping technique. Without the clipping technique, we can only get an upper bound of $6d$ by utilizing (E.2) which introduces another $d$ factor in $\xi_t$.

(3) $\psi_{t,j,2} = \hat{p}_{t,j}$ indicates $\hat{p}_{t,j} \geq \frac{25}{16} \frac{L_{\max}}{T_t} \ln \frac{2}{\delta_{d,T_t}} \geq \frac{d}{4T_t} \ln \frac{2}{\delta_{m,T_t}}$, thus (I.6) can be applied and $\beta_{t,j} \leq \frac{5}{2} \sqrt{\frac{8\hat{p}_{t,j}L_{\max}}{T_t} \ln \frac{2}{\delta}}$.

Therefore, conditional on Good, with probability at least $1 - KN\delta_{m,T_t}$, $\mathrm{RE} \leq 2\xi_t$. $\qquad\square$

*Proof of Lemma G.5.* By Lemma I.1, I.2 and I.3, conditional on Good, with probability at least $1 - (K\delta_{v,T_t} + N\delta_{d,T_t} + KN\delta_{m,T_t})$, $\left| \hat{\Delta}_t(x,\tilde{x}) - \Delta(x,\tilde{x}) \right| \leq \rho_t(x,\tilde{x})$. This finishes the proof of the first result in Lemma G.5.

We then accumulates all the error probabilities. By the choices of $\delta_{v,T_t} = \frac{\delta}{15KT_t^3}, \delta_{d,T_t} = \frac{\delta}{15NT_t^3}, \delta_{m,T_t} = \frac{\delta}{15KNT_t^3}$,

$$
\begin{aligned}
\sum_{T_t=1}^{\infty} K\delta_{v,T_t} + N\delta_{d,T_t} + KN\delta_{m,T_t} &= \sum_{T_t=1}^{\infty} K \cdot \frac{\delta}{15KT_t^3} + N \cdot \frac{\delta}{15NT_t^3} + KN \cdot \frac{\delta}{15KNT_t^3} \\
&= \sum_{T_t=1}^{\infty} \frac{3\delta}{15T_t^3} \\
&\leq \frac{\delta}{4}
\end{aligned}
$$

Note that there is an reversion step at Line 18 Algorithm 1, for each $T \in \mathbb{N}$, there are at most two time steps $t_1 < t_2$ with $T_{t_1} = T_{t_2}$. Therefore,

$$
\sum_{t=1}^{\infty} K\delta_{v,T_t} + N\delta_{d,T_t} + KN\delta_{m,T_t} \leq 2 \sum_{T_t=1}^{\infty} K\delta_{v,T_t} + N\delta_{d,T_t} + KN\delta_{m,T_t} \leq \frac{\delta}{2}
$$

This proves the second statement. $\qquad\square$

## J Upper Bound of PS$\varepsilon$BAI: Proof of Theorem 3.2

Theorem 3.2 is proved in this section by three steps.

- Firstly, we show that the recommended arm $\hat{x}_\varepsilon$ is an $\varepsilon$-best arm upon the termination of the algorithm in Lemma J.1.
- Secondly, we present a sufficient condition for the termination of the algorithm in terms of $T_t$ in Lemma G.6.
- Lastly, we show that $T_\tau$ is bounded by a constant fraction of $\tau$ and thus, obtain an upper bound on $\tau$.

**Step 1:**

According to Lemma G.5, $\mathrm{CI}_t$ holds with probability $1 - (K\delta_{v,T_t} + N\delta_{d,T_t} + KN\delta_{m,T_t})$. Conditional on the good event Good in (G.1) and the event $\mathrm{CI}_t$ (the mean gaps are well approximated at time step $t$), we expect the recommended arm will be an $\varepsilon$-optimal arm. This is formalized in the following lemma.

**Lemma J.1.** *Conditional on* Good *and* $\mathrm{CI}_t$, *if the algorithm stops at time step $t$, the recommended arm $\hat{x}_\varepsilon = x_t^*$ is an $\varepsilon$-best arm.*

*Proof of Lemma J.1.* If $\hat{x}_\varepsilon = x^*$, the lemma holds.

If $\hat{x}_\varepsilon \neq x^*$, according to the termination condition (3.4)

$$
\min_{x:x\neq\hat{x}_\varepsilon} \hat{\Delta}_t(\hat{x}_\varepsilon, x) - \rho_t(\hat{x}_\varepsilon, x) > -\varepsilon
$$

we have

$$
\Delta(\hat{x}_\varepsilon, x^*) \geq \hat{\Delta}_t(\hat{x}_\varepsilon, x^*) - \rho_t(\hat{x}_\varepsilon, x^*) \geq \min_{x:x\neq\hat{x}_\varepsilon} \hat{\Delta}_t(\hat{x}_\varepsilon, x) - \rho_t(\hat{x}_\varepsilon, x) > -\varepsilon
$$

where the first inequality holds due to $\mathrm{CI}_t$. This indicates $\Delta(x^*, \hat{x}_\varepsilon) \leq \varepsilon$ and thus the recommended arm $\hat{x}_\varepsilon = x_t^*$ is an $\varepsilon$-best arm. $\qquad\square$

## Step 2:

**Lemma G.6.** *Conditional on* Good *and* CI*, the recommended arm $\hat{x}_\varepsilon \in \mathcal{X}_\varepsilon$ and when Algorithm 1 terminates, the order of $T_t$ is upper bounded by (3.6).*

*Proof of Lemma G.6.* By Lemma J.1, $\hat{x}_\varepsilon \in \mathcal{X}_\varepsilon$. Note that for any $x \neq \hat{x}_\varepsilon$, when

$$\begin{cases} 2\rho_t(x, \hat{x}_\varepsilon) \leq \Delta(x, \hat{x}_\varepsilon) + \varepsilon, & \hat{x}_\varepsilon \neq x^* \text{ and } x = x^*, \\ \rho_t(\hat{x}_\varepsilon, x) + \rho_t(x^*, x) \leq \Delta(x^*, x) + \varepsilon, & \text{otherwise.} \end{cases} \tag{J.1}$$

By utilizing $\text{CI}_t$, we have

$$\begin{cases} \hat{\Delta}_t(\hat{x}_\varepsilon, x^*) - \rho_t(\hat{x}_\varepsilon, x^*) \geq \hat{\Delta}_t(x^*, \hat{x}_\varepsilon) - \rho_t(\hat{x}_\varepsilon, x^*) \geq \Delta_t(x^*, \hat{x}_\varepsilon) - 2\rho_t(x^*, \hat{x}_\varepsilon) \geq -\varepsilon, \\ \qquad\qquad\qquad\qquad\qquad\qquad\qquad\qquad\qquad\qquad \hat{x}_\varepsilon \neq x^* \text{ and } x = x^*, \\ \hat{\Delta}_t(\hat{x}_\varepsilon, x) - \rho_t(\hat{x}_\varepsilon, x) \geq \hat{\Delta}_t(x^*, x) - \rho_t(\hat{x}_\varepsilon, x) \geq \Delta(x^*, x) - \rho_t(x^*, x) - \rho_t(\hat{x}_\varepsilon, x) \geq -\varepsilon, \\ \qquad\qquad\qquad\qquad\qquad\qquad\qquad\qquad\qquad\qquad\qquad\qquad \text{otherwise.} \end{cases}$$

Therefore, if (J.1) hold for all $x \neq \hat{x}_\varepsilon$, the algorithm must have stopped, i.e., it is sufficient to have the following for any $x \neq \hat{x}_\varepsilon$:

$$\begin{cases} \rho_t(x^*, \hat{x}_\varepsilon) \leq \dfrac{\Delta(x^*, \hat{x}_\varepsilon) + \varepsilon}{2}, & \hat{x}_\varepsilon \neq x^* \text{ and } x = x^*, \\ \rho_t(\hat{x}_\varepsilon, x) \leq \dfrac{\Delta(x^*, x) + \varepsilon}{2}, \rho_t(x^*, x) \leq \dfrac{\Delta(x^*, x) + \varepsilon}{2}, & \text{otherwise.} \end{cases}$$

Lemma J.2 gives an upper bound on the total number of time steps in Exp phases at which (J.1) must hold for any two arms $x_\varepsilon \in \mathcal{X}_\varepsilon, x \in \mathcal{X} \setminus \{x_\varepsilon\}$.

**Lemma J.2.** *For any two arms $(x_\varepsilon, x)$, where $x \neq x_\varepsilon, x^*$, when*

$$T_t = \frac{6400 \ln 6400 \cdot d}{(\Delta(x^*, x) + \varepsilon)^2} \ln \frac{Kd/(\Delta(x^*, x) + \varepsilon)^2}{\delta}$$

$$+ 6400 \ln 6400 \cdot \mathrm{H}_{\mathrm{DE}}(x_\varepsilon, x) \ln \frac{N\mathrm{H}_{\mathrm{DE}}(x_\varepsilon, x)}{\delta}$$

$$+ \frac{3200\sqrt{2}\ln 3200\sqrt{2} \cdot NL_{\max}}{\Delta(x^*, x) + \varepsilon} \ln \frac{\frac{KNL_{\max}}{\Delta(x^*, x) + \varepsilon}}{\delta} \tag{J.2}$$

$$= \tilde{O}\left(\frac{d}{(\Delta(x^*, x) + \varepsilon)^2} \ln \frac{1}{\delta} + \mathrm{H}_{\mathrm{DE}}(x_\varepsilon, x) \ln \frac{1}{\delta} + \frac{NL_{\max}}{\Delta(x^*, x) + \varepsilon} \ln \frac{1}{\delta}\right)$$

*where*

$$\mathrm{H}_{\mathrm{DE}}(x_\varepsilon, x) := \frac{L_{\max}}{(\Delta(x^*, x) + \varepsilon)^2} \left(\sum_{j=1}^{N} \sqrt{\min\left\{16p_j, \frac{1}{4}\right\}} |\Delta_j(x_\varepsilon, x) + \varepsilon|\right)^2,$$

*we must have*

$$\rho_t(x_\varepsilon, x) \leq \frac{\Delta(x^*, x) + \varepsilon}{2}$$

By taking the maximum of (J.2) over all $\varepsilon$-best arms and the suboptimal arms, we get

$$\tilde{T} := \max_{x_\varepsilon \in \mathcal{X}_\varepsilon, x \neq x_\varepsilon, x^*} \frac{6400 \ln 6400 \cdot d}{(\Delta(x^*, x) + \varepsilon)^2} \ln \frac{Kd/(\Delta(x^*, x) + \varepsilon)^2}{\delta}$$

$$+ 6400 \ln 6400 \cdot \mathrm{H}_{\mathrm{DE}}(x_\varepsilon, x) \ln \frac{N\mathrm{H}_{\mathrm{DE}}(x_\varepsilon, x)}{\delta}$$

$$+ \frac{3200\sqrt{2}\ln 3200\sqrt{2} \cdot NL_{\max}}{\Delta(x^*, x) + \varepsilon} \ln \frac{\frac{KNL_{\max}}{\Delta(x^*, x) + \varepsilon}}{\delta} \tag{J.3}$$

$$= \tilde{O}\left(\max_{x_\varepsilon \in \mathcal{X}_\varepsilon, x \neq x_\varepsilon, x^*} \frac{d}{(\Delta(x^*, x) + \varepsilon)^2} \ln \frac{1}{\delta} + \mathrm{H}_{\mathrm{DE}}(x_\varepsilon, x) \ln \frac{1}{\delta} + \frac{NL_{\max}}{\Delta(x^*, x) + \varepsilon} \ln \frac{1}{\delta}\right).$$

$\square$

*Proof of Lemma J.2.* Let $c_v, c_d, c_m \in (0,1)$ be constants, satisfying $c_v + c_d + 2c_m = 1$. By the definition of $\rho_t(x_\varepsilon, x)$ in (3.3), it is sufficient to have

$$
\begin{cases}
2\alpha_t \leq c_v \dfrac{\Delta(x^*, x) + \varepsilon}{2} \\[2mm]
\displaystyle\sum_{j=1}^{N} \beta_{t,j} |\hat{\Delta}_{t,j}^{\mathrm{clip}_2}(x_\varepsilon, x) + \zeta_t(x_\varepsilon, x)| \leq (c_d + c_m)\dfrac{\Delta(x^*, x) + \varepsilon}{2} \\[2mm]
2\xi_t \leq c_m \dfrac{\Delta(x^*, x) + \varepsilon}{2}
\end{cases}
$$

where we take $\zeta_t(x_\varepsilon, x) = \varepsilon$ for theoretical simplicity, but we adopt $\zeta_t(x, \tilde{x}) = \arg\min_{\zeta_t(x,\tilde{x}) \in \mathbb{R}} \sum_{j=1}^{N} \beta_{t,j} (\hat{\Delta}_{t,j}(x, \tilde{x}) + \zeta_t(x, \tilde{x}))^2 = -\frac{\sum_{j=1}^{N} \beta_{t,j} \hat{\Delta}_{t,j}(x,\tilde{x})}{\sum_{j=1}^{N} \beta_{t,j}}$ in the algorithm, which still enjoys the theoretical guarantee.

**VE Term** According to the definition of $\alpha_t$, we need to bound:

$$\alpha_t \leq c_v \frac{\Delta(x^*, x) + \varepsilon}{4}$$

$$\Leftrightarrow \quad 5\sqrt{\frac{d}{T_t} \ln \frac{2}{\delta_{v,T_t}}} \leq c_v \frac{\Delta(x^*, x) + \varepsilon}{4}$$

By a coarse estimation, it is sufficient to have

$$T_t \geq \frac{\bar{c}_v d}{(\Delta(x^*, x) + \varepsilon)^2} \ln \frac{Kd/(\Delta(x^*, x) + \varepsilon)^2}{\delta} = O\left(\frac{d}{(\Delta(x^*, x) + \varepsilon)^2} \ln \frac{Kd/(\Delta(x^*, x) + \varepsilon)^2}{\delta}\right)$$

where $\bar{c}_v = \frac{6400}{c_v^2} \ln \frac{6400}{c_v^2}$.

**DE Term** The difficulty lies at the DE term.

$$
\sum_{j=1}^{N} \beta_{t,j} |\hat{\Delta}_{t,j}^{\mathrm{clip}_2}(x_\varepsilon, x) + \zeta_t(x_\varepsilon, x)|
$$

$$
\leq \sum_{j=1}^{N} \beta_{t,j} |\hat{\Delta}_{t,j}^{\mathrm{clip}_2}(x_\varepsilon, x) + \varepsilon|
$$

$$
\leq \sum_{j:\psi_{t,j} > \hat{p}_{t,j}} \beta_{t,j} |\hat{\Delta}_{t,j}^{\mathrm{clip}_2}(x_\varepsilon, x) + \varepsilon|
$$

$$
+ \sum_{j:\psi_{t,j} = \hat{p}_{t,j}} \beta_{t,j} |\Delta_j(x_\varepsilon, x) + \varepsilon| + \beta_{t,j} |\hat{\Delta}_{t,j}^{\mathrm{clip}_2}(x_\varepsilon, x) - \Delta_j(x_\varepsilon, x)|
$$

$$
\leq \sum_{j:\psi_{t,j} > \hat{p}_{t,j}} \beta_{t,j}(2 + \varepsilon) + \sum_{j:\psi_{t,j} = \hat{p}_{t,j}} \beta_{t,j} |\hat{\Delta}_{t,j}^{\mathrm{clip}_2}(x_\varepsilon, x) - \Delta_j(x_\varepsilon, x)|
$$

$$
+ \sum_{j:\psi_{t,j} = \hat{p}_{t,j}} \beta_{t,j} |\Delta_j(x_\varepsilon, x) + \varepsilon|
$$

$$
\leq 2\xi_t + \sum_{j:\psi_{t,j} = \hat{p}_{t,j}} \beta_{t,j} |\Delta_j(x_\varepsilon, x) + \varepsilon|
$$

We will upper bound $2\xi_t$ by $c_m \frac{\Delta(x^*, x) + \varepsilon}{2}$ first; the second term will be included in the RE term and analyzed later. Therefore, it is sufficient to have

$$\sum_{j:\psi_{t,j} = \hat{p}_{t,j}} \beta_{t,j} |\Delta_j(x_\varepsilon, x) + \varepsilon| \leq c_d \frac{\Delta(x^*, x) + \varepsilon}{2} \tag{J.4}$$

By the definition of $\beta_{t,j}$ in (I.2) and Lemma K.3, we get

$$\beta_{t,j} = \frac{5}{2}\sqrt{\frac{2\phi_{t,j}L_{\max}}{T_t}\ln\frac{2}{\delta_{d,T_t}}} \leq \frac{5}{2}\sqrt{\frac{2\min\left\{16p_j,\frac{1}{4}\right\}L_{\max}}{T_t}\ln\frac{2}{\delta_{d,T_t}}} \tag{J.5}$$

**Lemma K.3.** *If* $\psi_{t,j} = \hat{p}_{t,j}$, *then*

$$\phi_{t,j} \leq \min\left\{16p_j,\frac{1}{4}\right\}.$$

Therefore, in order for (J.4) to hold, it is sufficient to have

$$\sum_{j:\psi_{t,j,2}=\hat{p}_{t,j}}\frac{5}{2}\sqrt{\frac{2\min\left\{16p_j,\frac{1}{4}\right\}L_{\max}}{T_t}\ln\frac{2}{\delta_{d,T_t}}}|\Delta_j(x_\varepsilon,x)+\varepsilon| \leq c_d\frac{\Delta(x^*,x)+\varepsilon}{2}$$

$$\Leftrightarrow \quad \frac{5}{2}\sqrt{\frac{2L_{\max}}{T_t}\ln\frac{2}{\delta_{d,T_t}}}\sum_{j:\psi_{t,j,2}=\hat{p}_{t,j}}\sqrt{\min\left\{16p_j,\frac{1}{4}\right\}}|\Delta_j(x_\varepsilon,x)+\varepsilon| \leq c_d\frac{\Delta(x^*,x)+\varepsilon}{2}$$

$$\Leftarrow \quad \frac{5}{2}\sqrt{\frac{2L_{\max}}{T_t}\ln\frac{2}{\delta_{d,T_t}}}\sum_{j=1}^{N}\sqrt{\min\left\{16p_j,\frac{1}{4}\right\}}|\Delta_j(x_\varepsilon,x)+\varepsilon| \leq c_d\frac{\Delta(x^*,x)+\varepsilon}{2}$$

$$\Leftarrow \quad T_t \geq \bar{c}_d\mathrm{H}_{\mathrm{DE}}(x_\varepsilon,x)\ln\frac{N\mathrm{H}_{\mathrm{DE}}(x_\varepsilon,x)}{\delta} = O\left(\mathrm{H}_{\mathrm{DE}}(x_\varepsilon,x)\ln\frac{N\mathrm{H}_{\mathrm{DE}}(x_\varepsilon,x)}{\delta}\right)$$

where $\bar{c}_d = \frac{100}{c_d^2}\ln\frac{100}{c_d^2}$.

**RE Term** By the definition of $\xi_t$ in (I.2), it is sufficient to have

$$25\sqrt{2}\frac{NL_{\max}}{T_t}\ln\frac{2}{\delta_{m,T_t}} \leq c_m\frac{\Delta(x^*,x)+\varepsilon}{2}$$

$$\Leftarrow \quad T_t \geq \frac{\bar{c}_m\cdot NL_{\max}}{\Delta(x^*,x)+\varepsilon}\ln\frac{\frac{KNL_{\max}}{\Delta(x^*,x)+\varepsilon}}{\delta} = O\left(\frac{NL_{\max}}{\Delta(x^*,x)+\varepsilon}\ln\frac{\frac{KNL_{\max}}{\Delta(x^*,x)+\varepsilon}}{\delta}\right)$$

where $\bar{c}_m = \frac{400\sqrt{2}}{c_m}\ln\frac{400\sqrt{2}}{c_m}$.

By taking $c_v = \frac{3}{4}, c_d = \frac{1}{8}$ and $c_m = \frac{1}{8}$, the upper bound on $T_t$ can be concluded as

$$T_t = \frac{6400\ln 6400\cdot d}{(\Delta(x^*,x)+\varepsilon)^2}\ln\frac{Kd/(\Delta(x^*,x)+\varepsilon)^2}{\delta} + 6400\ln 6400\cdot\mathrm{H}_{\mathrm{DE}}(x_\varepsilon,x)\ln\frac{N\mathrm{H}_{\mathrm{DE}}(x_\varepsilon,x)}{\delta}$$

$$+\frac{3200\sqrt{2}\ln 3200\sqrt{2}\cdot NL_{\max}}{\Delta(x^*,x)+\varepsilon}\ln\frac{\frac{KNL_{\max}}{\Delta(x^*,x)+\varepsilon}}{\delta}$$

$$= O\left(\frac{d}{(\Delta(x^*,x)+\varepsilon)^2}\ln\frac{Kd/(\Delta(x^*,x)+\varepsilon)^2}{\delta}\right.$$

$$\left.+\mathrm{H}_{\mathrm{DE}}(x_\varepsilon,x)\ln\frac{N\mathrm{H}_{\mathrm{DE}}(x_\varepsilon,x)}{\delta}+\frac{NL_{\max}}{\Delta(x^*,x)+\varepsilon}\ln\frac{\frac{KNL_{\max}}{\Delta(x^*,x)+\varepsilon}}{\delta}\right)$$

$$= \tilde{O}\left(\frac{d}{(\Delta(x^*,x)+\varepsilon)^2}\ln\frac{1}{\delta}+\mathrm{H}_{\mathrm{DE}}(x_\varepsilon,x)\ln\frac{1}{\delta}+\frac{NL_{\max}}{\Delta(x^*,x)+\varepsilon}\ln\frac{1}{\delta}\right).$$

$$\square$$

**Step 3:**

**Theorem 3.2.** *Define the context distribution estimation (DE) hardness parameter*

$$\mathrm{H}_{\mathrm{DE}}(x_\varepsilon,x) := \frac{L_{\max}}{(\Delta(x^*,x)+\varepsilon)^2}\bar{H}(x_\varepsilon,x)$$

where $\bar{H}(x_\varepsilon, x) := \left( \sum_{j=1}^{N} \sqrt{\min\{16 p_j, 1/4\}} |\Delta_j(x_\varepsilon, x) + \varepsilon| \right)^2$. *Under Assumption 1, with probability at least $1 - \delta$, PS$\varepsilon$BAI identifies an $\varepsilon$-optimal arm and its sample complexity is*

$$\tilde{O}\left( \max_{\substack{x_\varepsilon \in \mathcal{X}_\varepsilon \\ x \neq x_\varepsilon, x^*}} \underbrace{\frac{d}{(\Delta(x^*, x) + \varepsilon)^2} \ln \frac{1}{\delta}}_{T_V(x)} + \underbrace{H_{DE}(x_\varepsilon, x) \ln \frac{1}{\delta}}_{T_D(x_\varepsilon, x)} + \underbrace{\frac{N L_{max}}{\Delta(x^*, x) + \varepsilon} \ln \frac{1}{\delta}}_{T_R(x)} \right). \tag{3.6}$$

*Proof of Theorem 3.2.* By Assumption 1, and the choice of the parameters, Lemma G.4 indicates Good is guaranteed with probability at least $1 - \frac{\delta}{2}$.

Note that some arm pulls are not counted in $T_t$:

- In every $\gamma$ time steps, Algorithm 1 enters the CD phase to select a CD sample.
- When a changepoint is detected, Algorithm 1 steps into the CA phase, where $\frac{w}{2}$ arms are sampled in Algorithm 4.
- During the CA phase, we abandon $\frac{w\gamma}{2}$ samples.

Therefore, when $t$ is large (or after the first stationary segment)

$$t \leq T_t \cdot \frac{\gamma}{\gamma - 1} \frac{L_{min}}{L_{min} - w\gamma - \frac{w}{2}} \leq T_t \cdot \frac{\gamma}{\gamma - 1} \frac{L_{min}}{L_{min} - \frac{3w\gamma}{2}}. \tag{J.6}$$

As $\gamma \geq 2$ and $w\gamma$ are set to be upper bounded by $\frac{L_{min}}{3}$, so the fractions above is upper bounded by an absolute constant 4, e.g., with $\gamma = 2$ and $w = \frac{L_{min}}{6}$, the constant is 4. Therefore, $t$ is of the same order as $T_t$. By Lemma G.6, (3.6) holds with probability $1 - \delta$. $\qquad\square$

# K  Analysis of PS$\varepsilon$BAI: Technical Lemmas

**Lemma K.1.** *Given any $j \in [N]$ and $T_t \geq \frac{2 L_{max}}{9} \ln \frac{2}{\delta_{d, T_t}}$, $\tilde{\beta}_{t,j} \leq \beta_{t,j}$.*

*Proof of Lemma K.1.* Recall

$$\phi_{t,j} = \min\left\{ 4 \max\left\{ \hat{p}_{t,j}, \frac{25}{4} \frac{L_{max}}{T_t} \ln \frac{2}{\delta_{d, T_t}} \right\}, \frac{1}{4} \right\}$$

The proof is scheduled in 2 steps.

**Step 1**: Upper bound $\sqrt{\frac{2 p_j (1 - p_j) L_{max}}{T_t} \ln \frac{2}{\delta_{d, T_t}}}$.

When $\frac{L_{max}}{3 T_t} \ln \frac{2}{\delta_{d, T_t}} \leq \sqrt{\frac{2 p_j (1 - p_j) L_{max}}{T_t} \ln \frac{2}{\delta_{d, T_t}}}$, we have

$$\tilde{\beta}_{t,j} = \frac{5}{2} \sqrt{\frac{2 p_j (1 - p_j) L_{max}}{T_t} \ln \frac{2}{\delta_{d, T_t}}}$$

As $p_j (1 - p_j) \leq \frac{1}{4}, \forall p_j \in (0, 1)$, this naïve bound gives $\tilde{\beta}_{t,j} \leq \frac{5}{2} \sqrt{\frac{2 \cdot \frac{1}{4} \cdot L_{max}}{T_t} \ln \frac{2}{\delta_{d, T_t}}}$.

If $T_{t,j} > 0$, according to Lemma K.2,

$$\tilde{\beta}_{t,j} = \frac{5}{2} \sqrt{\frac{2 p_j (1 - p_j) L_{max}}{T_t} \ln \frac{2}{\delta_{d, T_t}}}$$

$$\leq \frac{5}{2} \sqrt{\frac{2 p_j T_{t,j} L_{max}}{T_t T_{t,j}} \ln \frac{2}{\delta_{d, T_t}}}$$

$$\leq \frac{5}{2} \sqrt{\frac{2 \hat{p}_{t,j} L_{max}}{T_t} \cdot 4 \max\left\{ 1, \frac{25}{4} \frac{L_{max}}{T_{t,j}} \ln \frac{2}{\delta_{d, T_t}} \right\} \ln \frac{2}{\delta_{d, T_t}}},$$

thus we have

$$\tilde{\beta}_{t,j} \leq \frac{5}{2}\sqrt{\frac{2\phi_{t,j}L_{\max}}{T_t}\ln\frac{2}{\delta_{d,T_t}}} \tag{K.1}$$

If $\underline{T_{t,j}=0}$, i.e., the latent context $j$ has never been observed and $\hat{p}_{t,j}=0$, we have

$$\tilde{\beta}_{t,j} = \frac{5}{2}\sqrt{\frac{2p_j(1-p_j)L_{\max}}{T_t}\ln\frac{2}{\delta_{d,T_t}}}$$

$$\leq \frac{5}{2}\sqrt{\frac{2\tilde{\beta}_{t,j}(1-\tilde{\beta}_{t,j})L_{\max}}{T_t}\ln\frac{2}{\delta_{d,T_t}}}$$

$$\Rightarrow \quad \tilde{\beta}_j(t,\delta) \leq \frac{\frac{25}{8}L_{\max}\ln\frac{2}{\delta_{d,T_t}}}{T_t + \frac{25}{8}L_{\max}\ln\frac{2}{\delta_{d,T_t}}} \leq \frac{25}{8}\frac{L_{\max}}{T_t}\ln\frac{2}{\delta_{d,T_t}}$$

Take the trivial bound into consideration and observe that $\phi_{t,j} = \min\{\frac{25L_{\max}}{T_t}\ln\frac{2}{\delta_{d,T_t}}, \frac{1}{4}\}$ and that

$$\frac{25}{8}\frac{L_{\max}}{T_t}\ln\frac{2}{\delta_{d,T_t}} \leq \frac{25\sqrt{2}L_{\max}}{2T_t}\ln\frac{2}{\delta_{d,T_t}} = \frac{5}{2}\sqrt{\frac{2\phi_{t,j}L_{\max}}{T_t}\ln\frac{2}{\delta_{d,T_t}}}.$$

Thus, (K.1) also holds for $T_{t,j}=0$.

$\underline{\text{To conclude}}$, we have

$$\tilde{\beta}_{t,j} = \frac{5}{2}\max\left\{\frac{L_{\max}}{3T_t}\ln\frac{2}{\delta_{d,T_t}}, \sqrt{\frac{2p_j(1-p_j)L_{\max}}{T_t}\ln\frac{2}{\delta_{d,T_t}}}\right\}$$

$$\leq \frac{5}{2}\max\left\{\frac{L_{\max}}{3T_t}\ln\frac{2}{\delta_{d,T_t}}, \sqrt{\frac{2\phi_{t,j}L_{\max}}{T_t}\ln\frac{2}{\delta_{d,T_t}}}\right\}$$

**Step 2**: Simplify the bound. Recall that

$$\phi_{t,j} = \min\left\{4\max\left\{\hat{p}_{t,j}, \frac{25}{4}\frac{L_{\max}}{T_t}\ln\frac{2}{\delta_{d,T_t}}\right\}, \frac{1}{4}\right\}$$

As $T_t \geq \frac{2L_{\max}}{9}\ln\frac{2}{\delta_{d,T_t}}$, we have

$$\frac{L_{\max}}{3T_t}\ln\frac{2}{\delta_{d,T_t}} \leq \sqrt{\frac{2\cdot\frac{1}{4}L_{\max}}{T_t}\ln\frac{2}{\delta_{d,T_t}}}$$

According to the definition of $\phi$, if $\underline{\hat{p}_{t,j} \geq \frac{25}{4}\frac{L_{\max}}{T_t}\ln\frac{2}{\delta_{d,T_t}}}$, we have

$$\frac{L_{\max}}{3T_t}\ln\frac{2}{\delta_{d,T_t}} \leq \frac{5\sqrt{2}L_{\max}}{T_t}\ln\frac{2}{\delta_{d,T_t}} \leq \sqrt{\frac{2\cdot4\hat{p}_{t,j}L_{\max}}{T_t}\ln\frac{2}{\delta_{d,T_t}}}$$

If $\hat{p}_{t,j} \leq \frac{25}{4}\frac{L_{\max}}{T_t}\ln\frac{2}{\delta_{d,T_t}}$, we have

$$\frac{L_{\max}}{3T_t}\ln\frac{2}{\delta_{d,T_t}} \leq \frac{25\sqrt{2}L_{\max}}{T_t}\ln\frac{2}{\delta_{d,T_t}} = \sqrt{\frac{2L_{\max}}{T_t}\ln\frac{2}{\delta_{d,T_t}} \cdot 25\frac{L_{\max}}{T_t}\ln\frac{2}{\delta_{d,T_t}}}$$

Thus

$$\max\left\{\frac{L_{\max}}{3T_t}\ln\frac{2}{\delta_{d,T_t}}, \sqrt{\frac{2\phi_{t,j}L_{\max}}{T_t}\ln\frac{2}{\delta_{d,T_t}}}\right\} = \sqrt{\frac{2\phi_{t,j}L_{\max}}{T_t}\ln\frac{2}{\delta_{d,T_t}}}$$

This gives us the desired result. $\qquad\square$

**Lemma K.2.** *When* $\tilde{\beta}_{t,j} = \frac{5}{4}\sqrt{\frac{2p_j(1-p_j)L_{\max}}{T_t}\ln\frac{2}{\delta_{d,T_t}}}$ *and* $T_{t,j} > 0$, *we have*

$$\frac{p_j}{T_{t,j}} \leq \frac{4}{T_t}\max\left\{1, \frac{25}{16}\frac{L_{\max}}{T_{t,j}}\ln\frac{2}{\delta_{d,T_t}}\right\}$$

*Proof of Lemma K.2.*

$$\frac{p_j}{T_{t,j}} = \frac{p_j T_t}{T_{t,j} T_t}$$

$$\leq \frac{\hat{p}_{t,j} T_t + \frac{5}{4}\sqrt{2p_j(1-p_j)L_{\max}T_t \ln\frac{2}{\delta_{d,T_t}}}}{T_{t,j} T_t}$$

$$= \frac{1}{T_t} + \frac{5}{4}\sqrt{\frac{p_j}{T_{t,j}}} \cdot \sqrt{\frac{1}{T_{t,j}} - \frac{p_j}{T_{t,j}}} \cdot \sqrt{\frac{2L_{\max}}{T_t} \ln\frac{2}{\delta_{d,T_t}}}$$

$$\Rightarrow \quad \frac{p_j}{T_{t,j}} \leq \frac{4}{T_t}\max\left\{1, \frac{25}{16}\frac{L_{\max}}{T_{t,j}} \ln\frac{2}{\delta_{d,T_t}}\right\}$$

$\square$

**Lemma K.3.** *If $\psi_{t,j} = \hat{p}_{t,j}$, then*

$$\phi_{t,j} \leq \min\left\{16p_j, \frac{1}{4}\right\}.$$

*Proof of Lemma K.3.* Recall to the definition of $\phi_{t,j}$ in (I.2) and $\psi_{t,j}$ in (I.8),

$$\phi_{t,j} = \min\left\{4\max\left\{\hat{p}_{t,j}, \frac{25}{4}\frac{L_{\max}}{T_t} \ln\frac{2}{\delta_{d,T_t}}\right\}, \frac{1}{4}\right\}, \quad \psi_{t,j} = \max\left\{\hat{p}_{t,j}, \frac{25}{4}\frac{L_{\max}}{T_t} \ln\frac{2}{\delta_{d,T_t}}\right\},$$

we have

$$\hat{p}_{t,j} \geq \frac{25}{4}\frac{L_{\max}}{T_t} \ln\frac{2}{\delta_{d,T_t}}, \quad \phi_{t,j} = \min\left\{4\hat{p}_{t,j}, \frac{1}{4}\right\}$$

We only need to upper bound $\hat{p}_{t,j}$.

By (I.4),

$$\psi_{t,j} = \hat{p}_{t,j} \leq p_j + \frac{5}{2}\max\left\{\frac{L_{\max}}{3T_t}\ln\frac{2}{\delta_{d,T_t}}, \sqrt{\frac{2p_j(1-p_j)L_{\max}}{T_t}\ln\frac{2}{\delta_{d,T_t}}}\right\}$$

$$\Rightarrow \quad \psi_{t,j} \leq p_j + \frac{5}{2}\max\left\{\frac{4}{75}\psi_{t,j}, \sqrt{p_j(1-p_j)\frac{8}{25}\psi_{t,j}}\right\}$$

$$= p_j + \max\left\{\frac{4}{15}\psi_{t,j}, \sqrt{2p_j(1-p_j)\psi_{t,j}}\right\}$$

$$\Rightarrow \quad \psi_{t,j} \leq 4p_j$$

Thus $\hat{p}_{t,j} = \psi_{t,j} \leq 4p_j$, and $\phi_{t,j} \leq \min\left\{16p_j, \frac{1}{4}\right\}$. $\square$

## L  Upper Bound of $\mathrm{PS}\varepsilon\mathrm{BAI}^+$: Proof of Theorem 3.3

**Theorem 3.3.** *The $\mathrm{PS}\varepsilon\mathrm{BAI}^+$ algorithm is $(\varepsilon, \delta)$-PAC and its expected sample complexity is*

$$\tilde{O}\left(\min\left\{\max_{x_\varepsilon \in \mathcal{X}_\varepsilon, x \neq x_\varepsilon, x^*} T_V(x) + T_D(x_\varepsilon, x) + T_R(x), T_V^N + T_D^N\right\}\right).$$

*Proof.* We let $\tau_1$ denote the stopping time of $\mathrm{PS}\varepsilon\mathrm{BAI}$, $\tau_2$ denote the stopping time of $\mathrm{N}\varepsilon\mathrm{BAI}$, and $\tau$ denote the stopping time of $\mathrm{PS}\varepsilon\mathrm{BAI}^+$ when an identical sequence of samples and returns are applied to them. By the design of these algorithms, we have

$$\tau \leq \min\{\tau_1, \tau_2\}.$$

**Part I**: Prove that $\mathbb{P}(\mathring{x}_\varepsilon \notin \mathcal{X}_\varepsilon) \leq \delta$.

Let event $\mathcal{E}_1 = 1\{\tau_1 < \tau_2\} \cap 1\{\mathrm{PS}\varepsilon\mathrm{BAI} \text{ returns arm } x_\varepsilon \text{ when it terminates}\}$. Recall that $\dot{x}_\varepsilon$ denotes the recommended arm of $\mathrm{N}\varepsilon\mathrm{BAI}$ when it terminates and $\mathring{x}_\varepsilon$ denotes the recommended arm of

PS$\varepsilon$BAI$^+$ when it terminates. The design of algorithms indicates that:
$$\mathring{x}_\varepsilon = x_\varepsilon \text{ when } \mathcal{E}_1 \text{ occurs, and } \mathring{x}_\varepsilon = \dot{x}_\varepsilon \text{ when } \mathcal{E}_1^c \text{ occurs.}$$

Moreover, with the performance guarantees of N$\varepsilon$BAI and PS$\varepsilon$BAI(shown in Proposition 3.1 and Theorem 3.2 individually), we have

$$
\begin{aligned}
\mathbb{P}[\mathring{x}_\varepsilon \notin \mathcal{X}_\varepsilon] &= \mathbb{P}[\mathring{x}_\varepsilon \notin \mathcal{X}_\varepsilon | \mathcal{E}_1] \cdot \mathbb{P}[\mathcal{E}_1] + \mathbb{P}(\mathring{x}_\varepsilon \notin \mathcal{X}_\varepsilon | \mathcal{E}_1^c) \cdot \mathbb{P}[\mathcal{E}_1^c] \\
&= \mathbb{P}[x_\varepsilon \notin \mathcal{X}_\varepsilon | \mathcal{E}_1] \cdot \mathbb{P}[\mathcal{E}_1] + \mathbb{P}[\dot{x}_\varepsilon \notin \mathcal{X}_\varepsilon | \mathcal{E}_1^c] \cdot \mathbb{P}[\mathcal{E}_1^c] \\
&\le \delta \cdot \mathbb{P}[\mathcal{E}_1] + \delta \cdot \mathbb{P}[\mathcal{E}_1^c] = \delta \cdot (\mathbb{P}[\mathcal{E}_1] + \mathbb{P}[\mathcal{E}_1^c]) = \delta.
\end{aligned}
$$

**Part II**: Derive the expected sample complexity of PS$\varepsilon$BAI$^+$.

**We first derive the conditional expectation of the stopping time of** PS$\varepsilon$BAI**.** By the same argument as in (J.6), $T_t \le t \le 4T_t$ for any $t$. Therefore, it is sufficient to upper bound $\mathbb{E}[T_{\tau_1}]$ and $\mathbb{E}[\tau_1]$ can be upper bounded by $4\mathbb{E}[T_{\tau_1}]$.

According to Lemma G.6 and Lemma G.6, when both events Good and CI occur, PS$\varepsilon$BAI terminates with $T_{\tau_1}$ satisfying $T_{\tau_1} \le \tilde{T}$, where $\tilde{T}$ is defined in (J.3) and as below:

$$
\begin{aligned}
\tilde{T} := \max_{x_\varepsilon \in \mathcal{X}_\varepsilon, x \ne x_\varepsilon, x^*} & \frac{6400 \ln 6400 \cdot d}{(\Delta(x^*, x) + \varepsilon)^2} \ln \frac{Kd/(\Delta(x^*, x) + \varepsilon)^2}{\delta} \\
& + 6400 \ln 6400 \cdot \mathrm{H}_{\mathrm{DE}}(x_\varepsilon, x) \ln \frac{N \mathrm{H}_{\mathrm{DE}}(x_\varepsilon, x)}{\delta} \\
& + \frac{3200\sqrt{2} \ln 3200\sqrt{2} \cdot NL_{\max}}{\Delta(x^*, x) + \varepsilon} \ln \frac{\frac{KNL_{\max}}{\Delta(x^*, x) + \varepsilon}}{\delta}.
\end{aligned}
$$

Given time step $t$ with $T_t \ge 2\tilde{T}$ (thus $t \ge 8\tilde{T}$), we will upper bound $\mathbb{P}[\tau_1 > t]$ in the following. We firstly upper bound $\mathbb{P}[T_{\tau_1} > T_t]$. Note that

$$
\begin{aligned}
\mathbb{P}[T_{\tau_1} > T_t] = \ & \mathbb{P}\left[T_{\tau_1} > T_t | T_{\tau_1} \ge \frac{T_t}{2} + 1\right] \mathbb{P}\left[T_{\tau_1} \ge \frac{T_t}{2} + 1\right] \\
& + \mathbb{P}\left[T_{\tau_1} > T_t | T_{\tau_1} < \frac{T_t}{2} + 1\right] \mathbb{P}\left[T_{\tau_1} < \frac{T_t}{2} + 1\right] \\
\le \ & \mathbb{P}\left[T_{\tau_1} > T_t | T_{\tau_1} \ge \frac{T_t}{2} + 1\right].
\end{aligned}
$$

If PS$\varepsilon$BAI fails to terminate with $T_{\tau_1} \le \frac{T_t}{2}$, it implies that event $\text{Good} \bigcap \left(\bigcap_{s \in \mathcal{I}} \mathrm{CI}_s\right)$ fails, where $\mathcal{I} = \{s : \frac{T_t}{2} + 1 \le T_s \le T_t\}$. Hence,

$$
\begin{aligned}
\mathbb{P}[T_{\tau_1} > T_t] &\le \mathbb{P}\left[T_{\tau_1} > T_t \middle| T_{\tau_1} \ge \frac{T_t}{2} + 1\right] \\
&\le \mathbb{P}\left[\text{Good}^c \bigcup \left(\bigcup_{s \in \mathcal{I}} \mathrm{CI}_s^c\right)\right] \\
&\le \mathbb{P}[\text{Good}^c] + \mathbb{P}\left[\text{Good} \bigcap \left(\bigcup_{s \in \mathcal{I}} \mathrm{CI}_s^c\right)\right] \\
&\le \mathbb{P}[\text{Good}^c] + \mathbb{P}\left[\left(\bigcup_{s \in \mathcal{I}} \mathrm{CI}_s^c\right) \middle| \text{Good}\right] \\
&\overset{(a)}{\le} \frac{\delta}{2\tau^*} + 2 \sum_{T_s = T_t/2 + 1}^{T_t} (K\delta_{v, T_s} + N\delta_{d, T_s} + KN\delta_{m, T_s}) \\
&\overset{(b)}{\le} \frac{\delta}{2\tau^*} + 2 \sum_{T_s = T_t/2 + 1}^{T_t} \frac{\delta}{5T_s^3} \le \frac{\delta}{2\tau^*} + 2 \int_{T_t/2}^{T_t} \frac{\delta}{5x^3} \, \mathrm{d}x = \frac{\delta}{2\tau^*} + \frac{3\delta}{5T_t^2},
\end{aligned}
$$

where $(a)$ is obtained by applying Lemma G.5 to time steps in $\mathcal{I} \cap \mathcal{T}_t$ (i.e., the Exp time steps in $\mathcal{I}$) and note that there are at most two time steps $t_1 < t_2$ with $T_{t_1} = T_{t_2}$ due to the reversion step; $(b)$ is derived by substituting $\delta_{v,T_t} = \frac{\delta}{15KT_t^3}, \delta_{d,T_t} = \frac{\delta}{15NT_t^3}, \delta_{m,T_t} = \frac{\delta}{15KNT_t^3}$ which are used to define the confidence radius $\rho$ in (3.3). By using the fact that $4T_{\tau_1} \geq \tau_1$, for any $T_t \geq 2\tilde{T}$, we have

$$\mathbb{P}\left[\tau_1 > 4T_t\right] \leq \mathbb{P}\left[T_{\tau_1} > T_t\right] \leq \frac{\delta}{2\tau^*} + \frac{3\delta}{5T_t^2}$$

$$\Rightarrow \quad \mathbb{P}\left[\tau_1 > 4T_t + i\right] \leq \frac{\delta}{2\tau^*} + \frac{3\delta}{5T_t^2}, \quad i = 0,1,2,3$$

$$\Rightarrow \quad \mathbb{P}\left[\tau_1 > t\right] \leq \frac{\delta}{2\tau^*} + \frac{48\delta}{5(t-4)^2}, \quad \forall 8\tilde{T} \leq t \leq \tau^*.$$

This indicates for $t \geq 8\tilde{T}$, the probability that PS$\varepsilon$BAI does not stop after $t$ time steps is $\frac{\delta}{2\tau^*} + \frac{48\delta}{5(t-4)^2}$. Therefore, by Tonelli's Theorem, we have

$$\mathbb{E}[\tau_1|8\tilde{T} < \tau_1 \leq \tau^*] \leq \sum_{t=8\tilde{T}}^{\tau^*-1} \mathbb{P}[\tau_1 > t] \leq \sum_{t=8\tilde{T}}^{\tau^*-1} \left(\frac{\delta}{2\tau^*} + \frac{48\delta}{5(t-4)^2}\right)$$

$$\leq 1 + \int_{8\tilde{T}-1}^{\tau^*-1} \frac{48\delta}{5(t-4)^2}\, \mathrm{d}t \leq 2. \tag{L.1}$$

**Next, we bound $\mathbb{E}\tau$, the expected sample complexity of PS$\varepsilon$BAI$^+$.** $\mathbb{E}\tau$ can be decomposed as below:

$$\mathbb{E}\tau \leq 8\tilde{T} + \mathbb{E}[\tau|8\tilde{T} < \tau \leq \tau^*] + \mathbb{E}[\tau|\tau > \tau^*]. \tag{L.2}$$

Since $\tau = \min\{\tau_1, \tau_2\}$ and $\mathbb{P}[\min\{\tau_1, \tau_2\} > t] \leq \mathbb{P}[\tau_1 > t]$ for all $t$, we have

$$\mathbb{E}[\tau|8\tilde{T} < \tau \leq \tau^*] = \sum_{t=8\tilde{T}}^{\tau^*-1} \mathbb{P}[\tau > t] = \sum_{t=8\tilde{T}}^{\tau^*-1} \mathbb{P}[\min\{\tau_1, \tau_2\} > t]$$

$$\leq \sum_{t=8\tilde{T}}^{\tau^*-1} \mathbb{P}[\tau_1 > t] = \mathbb{E}[\tau_1|8\tilde{T} < \tau_1 \leq \tau^*]. \tag{L.3}$$

Besides, since PS$\varepsilon$BAI will terminate after $\tau^*$ time steps, i.e., $\mathbb{P}(\tau_1 \leq \tau^*) = 1$. We have

$$\mathbb{E}[\tau|\tau \geq \tau^*] = \sum_{t=\tau^*}^{\infty} \mathbb{P}[\tau \geq t] \leq 1 + \sum_{t=\tau^*+1}^{\infty} \mathbb{P}[\tau_2 \geq t] = 1 + \mathbb{E}[\tau_2|\tau_2 > \tau^*]. \tag{L.4}$$

Lemma F.1 indicates that $\mathbb{P}[\tau_2 \geq t] \leq \frac{\delta}{(\alpha-1)C_3(t/2)^2}$ for all $t \geq T_0$, where

$$T_0 = \frac{768(8L_{\max} + 25d)}{(\varepsilon + \Delta_{\min})^2} \ln \frac{768KC_3(8L_{\max} + 25d)}{(\varepsilon + \Delta_{\min})^2 \delta}, \quad C_3 = \sum_{n=1}^{\infty} n^{-3}.$$

Since the order of $T_0$ is apparently larger than $\tau^*$, we have $T_0 \leq \tau^*$. Then, with the same method as in the proof of Proposition 3.1, the conditional sample complexity of N$\varepsilon$BAI can be bounded as follows:

$$\mathbb{E}[\tau_2|\tau_2 > \tau^*] \leq \int_{\tau^*}^{+\infty} \mathbb{P}(\tau_2 \geq x)\, \mathrm{d}x \leq \int_{\tau^*}^{+\infty} \frac{\delta}{(\alpha-1)C_3(x/2)^2}\, \mathrm{d}x = \frac{\delta}{C_3\tau^*}. \tag{L.5}$$

Substituting terms in (L.2) with (L.1), (L.3), (L.4) and (L.5), we have

$$\mathbb{E}\tau \leq 8\tilde{T} + 3 + \frac{\delta}{C_3\tau^*}.$$

Besides,

$$\mathbb{E}\tau \leq \mathbb{E}\tau_2 \overset{(a)}{\leq} T_0 + \frac{\delta}{(\alpha-1)(\alpha-2)(T_0/2)^{\alpha-2}},$$

Where (a) is shown in (F.1).

**Altogether**, we have

$$\mathbb{E}\tau \leq \min\left\{8\tilde{T} + 3 + \frac{\delta}{C_3\tau^*},\ 8\tilde{T} + 3 + \frac{\delta}{C_3\tau^*}\right\}.$$

$\square$

# M Analysis of Lower Bound: Proof of Theorem 4.1

The dynamics for under our PSLB model is displayed in Dynamics 1.

---
**Dynamics 1** Dynamics for piecewise-stationary linear bandits
---
1: The instance: $\Lambda = (\mathcal{X}, \Theta, P_\theta, \mathcal{C})$
2: **while** the algorithm does not stop at time step $t$ **do**
3:    **if** $t \in \mathcal{C}$ **then**
4:       The environment samples $\theta^*_{j_t} \sim P_\theta$
5:    **else**
6:       $\theta^*_{j_t} = \theta^*_{j_{t-1}}$ (the environment does not change)
7:    **end if**
8:    The agent samples an arm $x_t$ based on the history up to time $t-1$.
9:    The reward $Y_{t,x_t} = x_t^\top \theta^*_{j_t} + \eta_t$ is revealed to the agent.
10: **end while**
11: Recommend an $\varepsilon$-best arm $\hat{x}_\varepsilon$.

---

To derive the lower bound in Theorem 4.1, we investigate two environments different from the one defined in Section 2 (and as in Dynamics 1):

• Dynamics 2: the agent observes the index of current context $j_t$, and the environment reduces to contextual linear bandits; the definition of Dynamics 2 and the lower bound under it are detailed in Appendix M.1.

• Dynamics 3: the agent observes the changepoints in $\mathcal{C}$ and context vector $\theta^*_{j_t}$'s, and hence she solely needs to estimate the distribution of contexts; the definition of Dynamics 3 and the lower bound under it are detailed in Appendix M.2.

We **first** derive a lower bound for $(\varepsilon, \delta)$-BAI algorithms in Dynamics 2, that is, in contextual linear bandits.

**Corollary M.1.** *For any $(\varepsilon, \delta)$-PAC algorithm $\pi$, there exists an instance $\Lambda = (\mathcal{X}, \Theta, P_\theta, \mathcal{C})$ with Dynamics 2 such that*

$$\mathbb{E}[\tau] \geq T_\varepsilon(\Lambda) \log \frac{1}{2.4\delta}.$$

*where* $\quad T_\varepsilon(\Lambda)^{-1} = \max_{\{v_j \in \boldsymbol{\Delta}_\mathcal{X}\}_{j=1}^N} \min_{\Lambda' \in \mathrm{Alt}_\Theta(\Lambda)} \sum_{j=1}^N p_j \sum_{x \in \mathcal{X}} v_{j,x} \frac{(x^\top(\theta^*_j - \theta'_j))^2}{2}$

*is defined in Theorem 4.1. In addition, when $|\mathcal{X}_\varepsilon| = 1$,*

$$T_\varepsilon(\Lambda) = \min_{\{v_j \in \boldsymbol{\Delta}_\mathcal{X}\}_{j=1}^N} \max_{x \neq x^*} \frac{\sum_{j=1}^N p_j \|x^* - x\|^2_{A(v_j)^{-1}}}{(\Delta(x^*, x) + \varepsilon)^2}.$$

This lower bound generalizes the result of [16] to the linear bandit setting.

We **next** study Dynamics 3. In this setting, the agent solely needs to estimate the distribution of contexts $P_\theta$ with context samples. Once the agent obtains a good estimate of $P_\theta$, she can identify an $\varepsilon$-optimal arm w.h.p. Hence, the lower bound on the complexity of an $(\varepsilon, \delta)$-BAI algorithm is the product of the minimum length of a stationary segment and the minimum number of context samples/changepoints needed for distribution estimation.

**Corollary M.2.** *For any $(\varepsilon, \delta)$-PAC algorithm $\pi$, there exists an instance $\Lambda = (\mathcal{X}, \Theta, P_\theta, \mathcal{C})$ with Dynamics 3 such that $\mathbb{E}\tau \geq c_{N_C}$, where $c_{N_C}$ is the $N_C^{th}$ changepoint in the changepoint sequence $\mathcal{C}$.*

**Altogether,** the sample complexity of an $(\varepsilon, \delta)$-BAI algorithm in Dynamics 2 and 3 build up the lower bound in Theorem 4.1.

## M.1 Lower Bound for $\varepsilon$-BAI in Contextual Linear Bandits

In this section, we consider a sub-problem where the index of the current context $j_t$ is revealed at each time step $t$. The original piecewise-stationary problem becomes a contextual problem whose dynamics is presented in Algorithm 2. As more information is provided to the agent, the lower

---

**Dynamics 2** Dynamics for contextual linear bandits

1: The instance: $\Lambda = (\mathcal{X}, \Theta, P_\theta, \mathcal{C})$
2: **while** the algorithm does not stop at time step $t$ **do**
3:    **if** $t \in \mathcal{C}$ **then**
4:       The environment samples $\theta^*_{j_t} \sim P_\theta$
5:    **else**
6:       $\theta^*_{j_t} = \theta^*_{j_{t-1}}$ (the environment does not change)
7:    **end if**
8:    Reveal $j_t$ to the agent.
9:    The agent samples an arm $x_t$ based on the history up to time $t - 1$.
10:   The reward $Y_{t,x_t} = x_t^\top \theta^*_{j_t} + \eta_t$ is revealed to the agent.
11: **end while**
12: Recommend an $\varepsilon$-best arm $\hat{x}_\varepsilon$.

---

bound for this sub-problem is smaller than the one for the original problem.

**Corollary M.1.** *For any $(\varepsilon, \delta)$-PAC algorithm $\pi$, there exists an instance $\Lambda = (\mathcal{X}, \Theta, P_\theta, \mathcal{C})$ with Dynamics 2 such that*

$$\mathbb{E}[\tau] \geq T_\varepsilon(\Lambda) \log \frac{1}{2.4\delta}.$$

$$where \quad T_\varepsilon(\Lambda)^{-1} = \max_{\{v_j \in \boldsymbol{\Delta}_\mathcal{X}\}_{j=1}^N} \min_{\Lambda' \in \mathrm{Alt}_\Theta(\Lambda)} \sum_{j=1}^N p_j \sum_{x \in \mathcal{X}} v_{j,x} \frac{(x^\top(\theta^*_j - \theta'_j))^2}{2}$$

*is defined in Theorem 4.1. In addition, when $|\mathcal{X}_\varepsilon| = 1$,*

$$T_\varepsilon(\Lambda) = \min_{\{v_j \in \boldsymbol{\Delta}_\mathcal{X}\}_{j=1}^N} \max_{x \neq x^*} \frac{\sum_{j=1}^N p_j \|x^* - x\|^2_{A(v_j)^{-1}}}{(\Delta(x^*, x) + \varepsilon)^2}.$$

For simplicity, we consider the noise model is the Clipped Gaussian Distribution $CN(1)$, i.e., $\eta \sim CN(1)$ or $P_\eta = CN(1)$.

**Definition M.3.** *A random variable $x$ follows the Clipped Gaussian Distribution with parameter $\sigma$, denoted by $x \sim CN(\sigma)$, if it has the probability distribution function*

$$f(x) = \frac{1}{(2\Phi(\sigma) - 1) \cdot \sqrt{2\pi\sigma^2}} \exp\left(-\frac{x^2}{2\sigma^2}\right), \quad \forall x \in [-1, 1]$$

*where $\Phi(x)$ is the cumulative distribution function of the standard Gaussian distribution.*

Some notations are introduced here:

- Given an instance $\Lambda = (\mathcal{X}, \Theta, P_\theta, \mathcal{C})$, define the alternative instance $\Lambda' = (\mathcal{X}, \Theta', P_{\theta'}, \mathcal{C})$ with respect to $\Lambda$, where $\Theta' = (\theta'_1, \ldots, \theta'_n) \in \mathbb{R}^{d \times N}$ and $P_{\theta'}[\theta'_j] = P_\theta[\theta^*_j]$, s.t. there $\exists x \in \mathcal{X} \setminus \mathcal{X}_\varepsilon, \forall x_\varepsilon \in \mathcal{X}_\varepsilon, s.t., x_\varepsilon^\top \mathbb{E}_{\theta' \sim P_{\theta'}} < x^\top \mathbb{E}_{\theta' \sim P_{\theta'}} - \varepsilon$. We denote the set containing all the alternative instance (w.r.t. $\Lambda$) as $\mathrm{Alt}_\Theta(\Lambda)$.
- $\mathcal{H}_t = (x_s, Y_{s,x_s}, j_s)_{s=1}^{t-1}$ is the observation history up to but not include time $t$.
- $N_{j,x}(t) = \sum_{s=1}^t \mathbb{1}\{x_s = x, j_s = j\}$ is the number of times arm in which $x$ is sampled under context $j$. And $N_j(t) := \sum_{x \in \mathcal{X}} N_{j,x}(t)$ is the number of times in which context $j$ appears.
- $\mathrm{kl}(p, q) = \mathrm{KL}(\mathrm{Bern}(p), \mathrm{Bern}(q))$ is the KL divergence between two Bernoulli distributions with parameters $p$ and $q$.

Therefore, the probability of the observation history $\mathcal{H}_{t+1} = (x_s, Y_{s,x_s}, j_s)_{s=1}^{t}$ is

$$P_{\pi\Lambda}\left[(x_s, Y_{s,x_s}, j_s)_{s=1}^{t}\right] = \prod_{l=1}^{l_t} P_\theta[\theta_{j_{c_l}}^*] \prod_{s=0}^{L_l-1} \pi(x_{c_l+s}|\mathcal{H}_{c_l+s})P_\eta(Y_{c_l+s,x_{c_l+s}}|x_{c_l+s}, \theta_{j_{c_l}}^*)$$

Then the log-likelihood between the two instance up to time $t$, given the observed data, is

$$
\begin{aligned}
&L_t\left[(x_s, Y_{s,x_s}, j_s)_{s=1}^{t}\right] \\
&= \log \frac{P_{\pi\Lambda}\left[(x_s, Y_{s,x_s}, j_s)_{s=1}^{t}\right]}{P_{\pi\Lambda'}\left[(x_s, Y_{s,x_s}, j_s)_{s=1}^{t}\right]} \\
&\overset{(a)}{=} \log \left( \frac{\prod_{l=1}^{l_t} P_\theta[\theta_{j_{c_l}}^*] \prod_{s=0}^{L_l-1} \pi(x_{c_l+s}|\mathcal{H}_{c_l+s})P(Y_{c_l+s,x_{c_l+s}}|x_{c_l+s}, \theta_{j_{c_l}}^*)}{\prod_{l=1}^{l_t} P_{\theta'}[\theta'_{j_{c_l}}] \prod_{s=0}^{L_l-1} \pi(x_{c_l+s}|\mathcal{H}_{c_l+s})P(Y_{c_l+s,x_{c_l+s}}|x_{c_l+s}, \theta'_{j_{c_l}})} \right) \\
&= \log \left( \frac{\prod_{l=1}^{l_t} \prod_{s=0}^{L_l-1} P_\eta(Y_{c_l+s,x_{c_l+s}} - x_{c_l+s}^\top \theta_{j_{c_l}}^*)}{\prod_{l=1}^{l_t} \prod_{s=0}^{L_l-1} P_\eta(Y_{c_l+s,x_{c_l+s}} - x_{c_l+s}^\top \theta'_{j_{c_l}})} \right) \\
&= \prod_{l=1}^{l_t} \sum_{s=0}^{L_l-1} \log \left( \frac{\exp(-\eta_{c_l+s}^2/2)}{\exp(-(\eta'_{c_l+s})^2/2)} \right) \\
&\overset{(b)}{=} \sum_{l=1}^{l_t} \sum_{s=0}^{L_l-1} \frac{-(Y_{c_l+s,x_{c_l+s}} - x_{c_l+s}^\top \theta_{j_{c_l}}^*)^2 + (Y_{c_l+s,x_{c_l+s}} - x_{c_l+s}^\top \theta'_{j_{c_l}})^2}{2} \\
&\overset{(c)}{=} \sum_{l=1}^{l_t} \sum_{s=0}^{L_l-1} \frac{x_{c_l+s}^\top \vartheta_{j_{c_l}}(2\eta_{c_l+s} + x_{c_l+s}^\top \vartheta_{j_{c_l}})}{2}
\end{aligned}
$$

where $(a)$ utilizes $P_\theta[\theta_{j_{c_l}}^*] = P_{\theta'}[\theta'_{j_{c_l}}]$, $(b)$ makes use of the relationship between the arm and observation and in $(c)$ $\vartheta_{j_{c_l}} := \theta_{j_{c_l}}^* - \theta'_{j_{c_l}}$. Thus, the expectation of the log-likelihood is (in the following, the expectation is taken under instance $\Lambda$ and algorithm $\pi$)

$$
\begin{aligned}
\mathbb{E}[L_t] &= \mathbb{E}\left[ \sum_{l=1}^{l_t} \sum_{s=0}^{L_l-1} \frac{x_{c_l+s}^\top \vartheta_{j_{c_l}}(2\eta_{c_l+s} + x_{c_l+s}^\top \vartheta_{j_{c_l}})}{2} \right] \\
&= \mathbb{E}\left[ \sum_{l=1}^{l_t} \sum_{s=0}^{L_l-1} \frac{\vartheta_{j_{c_l}}^\top x_{c_l+s} x_{c_l+s}^\top \vartheta_{j_{c_l}}}{2} \right] \\
&= \frac{1}{2}\mathbb{E}\left[ \sum_{l=1}^{l_t} \vartheta_{j_{c_l}}^\top \left( \sum_{s=0}^{L_l-1} x_{c_l+s} x_{c_l+s}^\top \right) \vartheta_{j_{c_l}} \right] \\
&= \frac{1}{2}\mathbb{E}\left[ \sum_{x\in\mathcal{X}} \mathbb{1}\{x_{c_l+s} = x\} \sum_{j=1}^{N} \mathbb{1}\{j_{c_l} = j\} \sum_{l=1}^{l_t} \vartheta_{j_{c_l}}^\top \left( \sum_{s=0}^{L_l-1} x_{c_l+s} x_{c_l+s}^\top \right) \vartheta_{j_{c_l}} \right] \\
&= \frac{1}{2} \sum_{x\in\mathcal{X}} \sum_{j=1}^{N} \mathbb{E}\left[ N_{j,x}(t) \right] \vartheta_j^\top x x^\top \vartheta_j \\
&= \frac{1}{2}\mathbb{E}[t] \sum_{j=1}^{N} \frac{\mathbb{E}[N_j(t)]}{\mathbb{E}[t]} \sum_{x\in\mathcal{X}} \frac{\mathbb{E}[N_{j,x}(t)]}{\mathbb{E}[N_j(t)]} \vartheta_j^\top x x^\top \vartheta_j
\end{aligned}
\tag{M.8}
$$

As the lengths of all the stationary phases are upper bounded, i.e., $L_l \leq L_{\max}, \forall l \in \mathbb{N}$, then by Wald's Lemma,

$$\frac{\mathbb{E}[N_j(t)]}{\mathbb{E}[t]} = \frac{\mathbb{E}[t]P_\theta[\theta_j^*]}{\mathbb{E}[t]} = P_\theta[\theta_j^*] = p_j \tag{M.9}$$

The above also holds for $t = \tau$ where $\tau$ is a stopping time. According to Lemma 19 in [31],

$$\mathbb{E}[L_\tau] \geq \sup_{\mathcal{E} \in \mathcal{F}_\tau} \mathrm{kl}(P_{\pi\Lambda}[\mathcal{E}], P_{\pi\Lambda'}[\mathcal{E}]) \tag{M.10}$$

where $\mathcal{F}_\tau = \sigma(\mathcal{H}_{\tau+1})$. In addition, let $\mathcal{E} = \{\text{the recommended arm } \hat{x}_\varepsilon \notin \mathcal{X}_\varepsilon\}$, as the algorithm $\pi$ is $\delta$-PAC, then

$$\mathrm{kl}(P_{\pi\Lambda}[\mathcal{E}], P_{\pi\Lambda'}[\mathcal{E}]) \geq \mathrm{kl}(1 - \delta, \delta) \geq \log \frac{1}{2.4\delta} \tag{M.11}$$

From (M.8), (M.9), (M.10) and (M.11), we conclude that

$$\frac{1}{2}\mathbb{E}[\tau] \sum_{j=1}^{N} p_j \sum_{x \in \mathcal{X}} \frac{\mathbb{E}\left[N_{j,x}(\tau)\right]}{\mathbb{E}[N_j(\tau)]} \vartheta_j^\top x x^\top \vartheta_j \geq \log \frac{1}{2.4\delta}$$

$$\Rightarrow \quad \min_{\Lambda' \in \mathrm{Alt}_\Theta(\Lambda)} \frac{1}{2}\mathbb{E}[\tau] \sum_{j=1}^{N} p_j \sum_{x \in \mathcal{X}} \frac{\mathbb{E}\left[N_{j,x}(\tau)\right]}{\mathbb{E}[N_j(\tau)]} \vartheta_j^\top x x^\top \vartheta_j \geq \log \frac{1}{2.4\delta}$$

$$\Rightarrow \quad \max_{\{v_j \in \boldsymbol{\Delta}_\mathcal{X}\}_{j=1}^N} \min_{\Lambda' \in \mathrm{Alt}_\Theta(\Lambda)} \frac{1}{2}\mathbb{E}[\tau] \sum_{j=1}^{N} p_j \sum_{x \in \mathcal{X}} v_{j,x} \vartheta_j^\top x x^\top \vartheta_j \geq \log \frac{1}{2.4\delta}$$

$$\Leftrightarrow \quad \mathbb{E}[\tau] \geq T_\varepsilon(\Lambda) \log \frac{1}{2.4\delta}$$

where

$$T_\varepsilon(\Lambda)^{-1} = \max_{\{v_j \in \boldsymbol{\Delta}_\mathcal{X}\}_{j=1}^N} \min_{\Lambda' \in \mathrm{Alt}_\Theta(\Lambda)} \frac{1}{2} \sum_{j=1}^{N} p_j \sum_{x \in \mathcal{X}} v_{j,x} \vartheta_j^\top x x^\top \vartheta_j.$$

The solution to the above optimization problem is in general intractable, even for the stationary case [32]. We can establish a connection of the above problem to the stationary $\varepsilon$-best identification problem in linear bandits when we assume the change of the latent vectors are the same, i.e., $\vartheta_1 = \ldots = \vartheta_N$.

### M.1.1 Connection with the Stationary Case

In the alternative instance $\mathrm{Alt}_\Theta(\Lambda)$, $\exists x \in \mathcal{X} \setminus \mathcal{X}_\varepsilon$, for any arm $x_\varepsilon \in \mathcal{X}_\varepsilon$, s.t.

$$\sum_{j=1}^{N} p_j x_\varepsilon^\top \theta_j' + \varepsilon < \sum_{j=1}^{N} p_j x^\top \theta_j' \tag{M.12}$$

$$\Leftrightarrow \quad \sum_{j=1}^{N} p_j (x_\varepsilon - x)^\top \theta_j' + \varepsilon < 0$$

$$\Leftrightarrow \quad -\sum_{j=1}^{N} p_j (x_\varepsilon - x)^\top \theta_j' + \sum_{j=1}^{N} p_j (x_\varepsilon - x)^\top \theta_j^* > \sum_{j=1}^{N} p_j (x_\varepsilon - x)^\top \theta_j^* + \varepsilon$$

$$\Leftrightarrow \quad \sum_{j=1}^{N} p_j (x_\varepsilon - x)^\top \vartheta_j > \sum_{j=1}^{N} p_j (x_\varepsilon - x)^\top \theta_j^* + \varepsilon = \Delta(x_\varepsilon, x) + \varepsilon$$

Thus,
$\mathrm{Alt}_\Theta(\Lambda)$

$$= \left\{ (\mathcal{X}, \Theta', P_\theta, \mathcal{C}) : \theta_j' = \theta_j^* - \vartheta_j, \exists x \in \mathcal{X} \setminus \mathcal{X}_\varepsilon, \sum_{j=1}^{N} p_j (x_\varepsilon - x)^\top \vartheta_j > \Delta(x_\varepsilon, x) + \varepsilon, \forall x_\varepsilon \in \mathcal{X}_\varepsilon \right\}.$$

Define $\mathrm{Alt}(\Lambda)_{\mathrm{restricted}} \subset \mathrm{Alt}_\Theta(\Lambda)$, with the additional constraint that $\vartheta_1 = \vartheta_2 = \cdots = \vartheta_N$. Then we have

$\mathrm{Alt}(\Lambda)_{\mathrm{restricted}}$

$$= \left\{ (\mathcal{X}, \Theta', P_\theta, \mathcal{C}) : \theta'_j = \theta^*_j - \vartheta_1, \exists x \in \mathcal{X} \setminus \mathcal{X}_\varepsilon, \sum_{j=1}^N p_j (x_\varepsilon - x)^\top \vartheta_j > \Delta(x_\varepsilon, x) + \varepsilon, \forall x_\varepsilon \in \mathcal{X}_\varepsilon \right\}.$$

Note that in stationary linear bandits, the instance can be characterized by the arm set $\mathcal{X} \subset \mathbb{R}^d$ and the latent vector $\theta \in \mathbb{R}^d$. Define the alternative instance in linear bandits [32] for the instance $(\mathcal{X}, \mathbb{E}_{\theta \sim P_\theta} \theta)$

$$\mathrm{Alt}((\mathcal{X}, \mathbb{E}_{\theta \sim P_\theta} \theta))_{\mathrm{stationary}}$$
$$= \left\{ (\mathcal{X}, \theta') : \theta' = \mathbb{E}_{\theta \sim P_\theta} \theta - \vartheta_1, (x_\varepsilon - x)^\top \vartheta_1 > (x_\varepsilon - x)^\top \mathbb{E}_{\theta \sim P_\theta} + \varepsilon, \forall x_\varepsilon \in \mathcal{X}_\varepsilon \right\}.$$

Note that

$$\max_{\{v_j \in \boldsymbol{\Delta}_\mathcal{X}\}_{j=1}^N} \min_{\Lambda' \in \mathrm{Alt}_\Theta(\Lambda)} \frac{1}{2} \mathbb{E}[\tau] \sum_{j=1}^N p_j \sum_{x \in \mathcal{X}} v_{j,x} \vartheta_j^\top x x^\top \vartheta_j$$

$$\leq \max_{\{v_j \in \boldsymbol{\Delta}_\mathcal{X}\}_{j=1}^N} \min_{\Lambda' \in \mathrm{Alt}(\Lambda)_{\mathrm{restricted}}} \frac{1}{2} \mathbb{E}[\tau] \vartheta_1^\top \left( \sum_{x \in \mathcal{X}} \left( \sum_{j=1}^N p_j v_{j,x} \right) x x^\top \right) \vartheta_1$$

$$\overset{(a)}{=} \max_{\bar{v} \in \boldsymbol{\Delta}_\mathcal{X}} \min_{\Lambda' \in \mathrm{Alt}(\Lambda)_{\mathrm{restricted}}} \frac{1}{2} \mathbb{E}[\tau] \vartheta_1^\top \left( \sum_{x \in \mathcal{X}} \bar{v}_i x x^\top \right) \vartheta_1$$

$$= \max_{\bar{v} \in \boldsymbol{\Delta}_\mathcal{X}} \min_{\Lambda' \in \mathrm{Alt}((\mathcal{X}, \mathbb{E}_{\theta \sim P_\theta} \theta))_{\mathrm{stationary}}} \frac{1}{2} \mathbb{E}[\tau] \vartheta_1^\top \left( \sum_{x \in \mathcal{X}} \bar{v}_i x x^\top \right) \vartheta_1$$

where in $(a)$ we denote $\bar{v} := \sum_{j=1}^N p_j v_j$ as a mixture of $\{v_j\}_{j=1}^N$. In other words, the max-min problem becomes the one for the $\varepsilon$-best arm identification problem in stationary linear bandits. According to [32], the solution to the last optimization problem above is in general intractable.

However, the optimization problem can be simplified under some simple cases, e.g., the set of $\varepsilon$-best arm is a singleton [32]. In the next subsection, a lower bound for the original problem with $\mathcal{X}_\varepsilon$ being a singleton will be derived.

**M.1.2  A Simple Case: $|\mathcal{X}_\varepsilon| = 1$**

Assume that the set of $\varepsilon$-best arm is a singleton, i.e., $\mathcal{X}_\varepsilon = \{x^*\}$, we will solve the original optimization problem:

$$\min_{\Lambda' \in \mathrm{Alt}_\Theta(\Lambda)} \frac{1}{2} \sum_{j=1}^N p_j \sum_{x \in \mathcal{X}} v_{j,x} \vartheta_j^\top x x^\top \vartheta_j$$

we extend the procedures in [1] to the piecewise-stationary setup.

**Lemma M.4.**

$$\min_{\Lambda' \in \mathrm{Alt}_\Theta(\Lambda)} \frac{1}{2} \sum_{j=1}^N p_j \sum_{x \in \mathcal{X}} v_{j,x} \vartheta_j^\top x x^\top \vartheta_j \geq \frac{1}{2} \min_{x \neq x^*} \frac{(\Delta(x^*, x))^2}{\sum_{j=1}^N p_j \|x^* - x\|^2_{A(v_j)^{-1}}}$$

*Proof.* Note that $\Lambda'$ differs from $\Lambda$ in the context matrix $\Theta$. By the derivations in (M.12), there exists arm $x \neq x^*$, s.t.,

$$\sum_{j=1}^N p_j (x^* - x)^\top \vartheta_j > \Delta(x_\varepsilon, x) + \varepsilon$$

Therefore, the optimization problem becomes

$$\min_{\vartheta_1, \dots, \vartheta_N} \frac{1}{2} \sum_{j=1}^N p_j \sum_{x \in \mathcal{X}} v_{j,x} \vartheta_j^\top x x^\top \vartheta_j$$

$$\text{s.t.} \quad \exists x \in \mathcal{X} \setminus \{x^*\}, \sum_{j=1}^N p_j (x^* - x)^\top \vartheta_j \geq \Delta(x^*, x) + \varepsilon + a =: \Delta_{a+\varepsilon}(x^*, x) \quad \text{(M.13)}$$

where $a > 0$. The Lagrangian function for this problem is

$$L(\vartheta_1, \ldots, \vartheta_N, \lambda) = \frac{1}{2} \sum_{j=1}^{N} p_j \sum_{x \in \mathcal{X}} v_{j,x} \vartheta_j^\top x x^\top \vartheta_j + \lambda(-\sum_{j=1}^{N} p_j (x^* - x)^\top \vartheta_j + \Delta_{a+\varepsilon}(x^*, x))$$

Then

$$\frac{\partial L}{\partial \vartheta_j} = 0 \implies p_j \sum_{x \in \mathcal{X}} v_{j,x} x x^\top \vartheta_j - \lambda p_j (x^* - x) = 0 \implies A(v_j)^{1/2} \vartheta_j = \lambda A(v_j)^{-1/2}(x^* - x)$$

$$\frac{\partial L}{\partial \lambda} = 0 \implies \sum_{j=1}^{N} p_j (x^* - x)^\top \vartheta_j = \Delta_{a+\varepsilon}(x^*, x)$$

This indicates that

$$\sum_{j=1}^{N} p_j \sum_{x \in \mathcal{X}} v_{j,x} \vartheta_j^\top x x^\top \vartheta_j = \sum_{j=1}^{N} p_j \|\vartheta_j\|^2_{A(v_j)}$$

$$\geq \frac{\left(\sum_{j=1}^{N} p_j \|\vartheta_j\|_{A(v_j)} \|x^* - x\|_{A(v_j)^{-1}}\right)^2}{\sum_{j=1}^{N} w_j \|x^* - x\|^2_{A(v_j)^{-1}}}$$

$$= \frac{\left(\sum_{j=1}^{N} p_j \vartheta_j^\top (x^* - x)\right)^2}{\sum_{j=1}^{N} p_j \|x^* - x\|^2_{A(v_j)^{-1}}}$$

$$= \frac{(\Delta_{a+\varepsilon}(x^*, x))^2}{\sum_{j=1}^{N} p_j \|x^* - x\|^2_{A(v_j)^{-1}}}$$

Let $a \to 0$, we have

$$\sum_{j=1}^{N} p_j \sum_{x \in \mathcal{X}} v_{j,x} \vartheta_j^\top x x^\top \vartheta_j \geq \frac{(\Delta(x^*, x) + \varepsilon)^2}{\sum_{j=1}^{N} p_j \|x^* - x\|^2_{A(v_j)^{-1}}}$$

Due to (M.13), we only require there exists $x \neq x^*$, such that the constraint is satisfied, therefore,

$$\frac{1}{2} \sum_{j=1}^{N} p_j \|\vartheta_j\|^2_{A(v_j)} \geq \frac{1}{2} \min_{x \neq x^*} \frac{(\Delta(x^*, x) + \varepsilon)^2}{\sum_{j=1}^{N} p_j \|x^* - x\|^2_{A(v_j)^{-1}}}$$

$\square$

By Lemma M.4, the stopping time can be lower bounded as

$$\mathbb{E}[\tau] \geq 2 \log \frac{1}{2.4\delta} \min_{\{v_j \in \mathbf{\Delta}_{\mathcal{X}}\}_{j=1}^{N}} \max_{x \neq x^*} \frac{\sum_{j=1}^{N} p_j \|x^* - x\|^2_{A(v_j)^{-1}}}{(\Delta(x^*, x) + \varepsilon)^2}$$

This lower bound indicates that, (1) a good algorithm should actively detects and makes use of the contextual information to facilitate the arm identification process. (2) our lower bound extends the result of [16] to the linear bandits case.

The above lower bound can be further lower bounded if we restrict $v_1 = \ldots = v_N$.

**Lemma M.5.** *Let* $\mathrm{SPD}(d) := \{A : A \in \mathbb{R}^{d \times d}, A > 0\}$ *denote the set of SPD matrices of dimension* $d \times d$. *Given any* $x \in \mathbb{R}^d$, *define the function* $f : \mathrm{SPD}(d) \to \mathbb{R}$, $f(A) = x^\top A^{-1} x$, *then* $f$ *is convex.*

*Proof of Lemma M.5.* Given any $A, B \in \mathrm{SPD}(d)$, define

$$g : \{t \in \mathbb{R} : A + tB \in \mathrm{SPD}(d)\} \to \mathbb{R}, g(t) = x^\top (A + tB)^{-1} x$$

It is suffice to prove $g$ is convex.

$$g'(t) = -x^\top (A + tB)^{-1} B (A + tB)^{-1} x$$
$$g''(t) = 2x^\top (A + tB)^{-1} B (A + tB)^{-1} B (A + tB)^{-1} x$$

$$= 2 \left( x^\top (A + tB)^{-1} B \right) (A + tB)^{-1} \left( B(A + tB)^{-1} x \right) \geq 0$$

Therefore, $g$ is convex. $\qquad\square$

Given Lemma M.5, we have

$$\sum_{j=1}^{N} p_j \|x^* - x\|_{A(v_j)^{-1}}^2 = (x^* - x)^\top \left( \sum_{j=1}^{N} p_j A(v_j)^{-1} \right) (x^* - x)$$

$$\geq (x^* - x)^\top \left( \sum_{j=1}^{N} p_j A(v_j) \right)^{-1} (x^* - x)$$

$$= (x^* - x)^\top \left( \sum_{j=1}^{N} p_j \sum_{x \in \mathcal{X}} v_{j,x} x x^\top \right)^{-1} (x^* - x)$$

$$= (x^* - x)^\top \left( \sum_{x \in \mathcal{X}} \left( \sum_{j=1}^{N} p_j v_{j,x} \right) x x^\top \right)^{-1} (x^* - x)$$

$$= (x^* - x)^\top A(\bar{v})^{-1} (x^* - x)$$

where $\bar{v} := \sum_{j=1}^{N} p_j v_j \in \mathbf{\Delta}_{\mathcal{X}}$ and the inequality becomes equality when $v_1 = \cdots = v_n$. Thus,

$$\mathbb{E}[\tau] \geq 2 \log \frac{1}{2.4\delta} \min_{\{v_j \in \mathbf{\Delta}_{\mathcal{X}}\}_{j=1}^N} \max_{x \neq x^*} \frac{\sum_{j=1}^{N} p_j \|x^* - x\|_{A(v_j)^{-1}}^2}{(\Delta(x^*, x) + \varepsilon)^2}$$

$$\geq 2 \log \frac{1}{2.4\delta} \min_{\bar{v} \in \mathbf{\Delta}_{\mathcal{X}}} \max_{x \neq x^*} \frac{\|x^* - x\|_{A(\bar{v})^{-1}}^2}{(\Delta(x^*, x) + \varepsilon)^2}. \qquad (\text{M.14})$$

The last term mimics the lower bound in the stationary linear bandits with the latent vector $\mathbb{E}_{\theta \sim P_\theta} \theta$.

In addition, if we let $p_1 = 1$ and $p_j = 0$ for all $j \neq 1$ in the instance, our lower bound can be simplified to the lower bound in stationary linear bandits with latent vector $\theta_1^*$ [24]

$$\mathbb{E}[\tau] \geq 2 \log \frac{1}{2.4\delta} \min_{\{v_j \in \mathbf{\Delta}_{\mathcal{X}}\}_{j=1}^N} \max_{x \neq x^*} \frac{\sum_{j=1}^{N} p_j \|x^* - x\|_{A(v_j)^{-1}}^2}{(\Delta(x^*, x) + \varepsilon)^2}$$

$$= 2 \log \frac{1}{2.4\delta} \min_{v_1 \in \mathbf{\Delta}_{\mathcal{X}}} \max_{x \neq x^*} \frac{\|x^* - x\|_{A(v_1)^{-1}}^2}{(\Delta_1(x^*, x) + \varepsilon)^2}. \qquad (\text{M.15})$$

### M.2 Lower Bound on the Number of changepoints

In this section, we consider an even easier problem: all the contextual information is known to the agent, except for the distribution $P_\theta$. The dynamics is displayed in Dynamics 3.

---

**Dynamics 3** Dynamics for an easier problem

---

1: The instance: $\Lambda = (\mathcal{X}, \Theta, P_\theta, \mathcal{C})$
2: **while** the algorithm does not stop at time step $t$ **do**
3:    **if** $t \in \mathcal{C}$ **then**
4:       The agent acknowledges $t \in \mathcal{C}$.
5:       The environment samples $\theta_{j_t}^* \sim P_\theta$
6:    **else**
7:       $\theta_{j_t}^* = \theta_{j_{t-1}}^*$ (the environment does not change)
8:    **end if**
9:    The agent observes $\theta_{j_t}^*$
10: **end while**
11: Recommend an $\varepsilon$-best arm $\hat{x}_\varepsilon$.

---

We are going to consider the alternative instances with respect to the distribution on $\theta$, i.e., $\Lambda' = (\mathcal{X}, \Theta, P'_\theta, \mathcal{C})$. where $\exists x \in \mathcal{X} \setminus \mathcal{X}_\varepsilon, \forall x_\varepsilon \in \mathcal{X}_\varepsilon, s.t., x_\varepsilon^\top \mathbb{E}_{\theta \sim P'_\theta} < x^\top \mathbb{E}_{\theta \sim P'_\theta} - \varepsilon$. Denote the set of all the alternative instance (w.r.t. $\Lambda$) as $\text{Alt}_P(\Lambda)$. According to the Pinsker's inequality, let $\mathcal{E} := \{\text{the recommended arm } \hat{x}_\varepsilon \notin \mathcal{X}_\varepsilon\}$, then for any $\delta$-PAC algorithm $\pi$,

$$P_{\pi\Lambda}[\mathcal{E}] + P_{\pi\Lambda'}[\mathcal{E}^c] \geq \frac{1}{2} \exp\left(-\text{KL}(P_{\pi\Lambda}, P_{\pi\Lambda'})\right), \quad \forall \Lambda' \in \text{Alt}_P(\Lambda)$$

$$\Rightarrow \quad 4\delta \geq \exp\left(-\text{KL}(P_{\pi\Lambda}, P_{\pi\Lambda'})\right), \quad \forall \Lambda' \in \text{Alt}_P(\Lambda)$$

$$\Rightarrow \quad \text{KL}(P_{\pi\Lambda}, P_{\pi\Lambda'}) \geq \ln \frac{1}{4\delta}, \quad \forall \Lambda' \in \text{Alt}_P(\Lambda)$$

$$\Rightarrow \quad \min_{\Lambda' \in \text{Alt}_P(\Lambda)} \text{KL}(P_{\pi\Lambda}, P_{\pi\Lambda'}) \geq \ln \frac{1}{4\delta}$$

$$\Rightarrow \quad \mathbb{E}[l_t] \min_{\Lambda' \in \text{Alt}_P(\Lambda)} \text{KL}(P_\theta, P'_\theta) \geq \ln \frac{1}{4\delta}$$

$$\Rightarrow \quad \mathbb{E}[l_t] \geq \frac{1}{\min_{\Lambda' \in \text{Alt}_P(\Lambda)} \text{KL}(P_\theta, P'_\theta)} \ln \frac{1}{4\delta} \tag{M.16}$$

We will give an upper bound on the denominator. Note that

$$\min_{\Lambda' \in \text{Alt}_P(\Lambda)} \text{KL}(P_\theta, P'_\theta) = \min_{x \in \mathcal{X} \setminus \mathcal{X}_\varepsilon} \min_{\Lambda_x} \text{KL}(P_\theta, P_\theta^x) \tag{M.17}$$

where $\Lambda_x$ is an alternative instance with distribution $P_\theta^x$ s.t. $x^\top \mathbb{E}_{\theta \sim P_\theta^x} \theta - \varepsilon > x_\varepsilon^\top \mathbb{E}_{\theta \sim P_\theta^x} \theta, \forall x_\varepsilon \in \mathcal{X}_\varepsilon$.

Given any $x \notin \mathcal{X}_\varepsilon$, we denote the shorthand notation $q_j = P_\theta^x[\theta_j^*]$. We have

$$\sum_{j=1}^N q_j \Delta_j(x, x_\varepsilon) \geq \varepsilon + a, \text{ for any } a > 0, x_\varepsilon \in \mathcal{X}_\varepsilon$$

Fix $a > 0$ which is sufficiently small, we have the following optimization problem:

$$\min_{q \in \mathbf{\Delta}_N} \sum_{j=1}^N p_j \ln \frac{p_j}{q_j}$$

$$s.t. \sum_{j=1}^N q_j = 1$$

$$\sum_{j=1}^N q_j \Delta_j(x_\varepsilon, x) + \varepsilon + a \leq 0, \forall x_\varepsilon \in \mathcal{X}_\varepsilon$$

If such alternative distribution $q$ exists, let $L$ denote the augmented Lagrangian function

$$L(q, \lambda, \{\lambda_{x_\varepsilon}\}_{x_\varepsilon \in \mathcal{X}_\varepsilon}) = \sum_{j=1}^N p_j \ln \frac{p_j}{q_j} + \lambda\left(\sum_{j=1}^N q_j - 1\right) + \sum_{x_\varepsilon \in \mathcal{X}_\varepsilon} \lambda_{x_\varepsilon}(q_j \Delta_j(x_\varepsilon, x) + \varepsilon + a)$$

By the KKT conditions, we have:

$$\frac{\partial L}{\partial q_j} = \frac{-p_j}{q_j} + \lambda + \sum_{x_\varepsilon \in \mathcal{X}_\varepsilon} \lambda_{x_\varepsilon} \Delta_j(x_\varepsilon, x) = 0, \ \forall j \in [N] \tag{M.18}$$

$$\frac{\partial L}{\partial \lambda} = \sum_{j=1}^N q_j - 1 = 0$$

$$\frac{\partial L}{\partial \lambda_{x_\varepsilon}} = \sum_{j=1}^N q_j \Delta_j(x_\varepsilon, x) + \varepsilon + a = 0, \forall x_\varepsilon \in \mathcal{X}_\varepsilon$$

or $\lambda_{x_\varepsilon} = 0$ for some $x_\varepsilon \in \mathcal{X}_\varepsilon$, which indicates some conditions are not satisfied. The equations above give

$$q_j = \frac{p_j}{1 + \sum_{x_\varepsilon \in \mathcal{X}_\varepsilon} \lambda_{x_\varepsilon}(\Delta_j(x_\varepsilon, x) + \varepsilon + a)}, \ \forall j \in [N] \tag{M.19}$$

By solving

$$\sum_{j=1}^{N} \frac{p_j}{1 + \sum_{x_\varepsilon \in \mathcal{X}_\varepsilon} \lambda_{x_\varepsilon}(\Delta_j(x_\varepsilon, x) + \varepsilon + a)} - 1 = 0 \tag{M.20}$$

which is a polynomial with $N - 1$ degree. Thus, an explicit solution for any instance is not applicable. However, there are some cases where we can get an estimate of $\lambda_{x_\varepsilon}$

**Lemma M.6.** *Denote* $j_0 = \arg\min_{j \in [N]} \Delta_j(x^*, x)$ *and* $j_1 = \arg\max_{j \in [N]} \Delta_j(x^*, x)$. *When* $|\mathcal{X}_\varepsilon| = 1$, $-\Delta_{j_0}(x^*, x) - 2\varepsilon > \Delta_{j_1}(x^*, x) > \varepsilon$ *and* $\sum_{j=1}^{N} p_j(\Delta_j(x^*, x) + \varepsilon + a)^3 \leq 0$, *the solution to* (M.20) *is upper bounded by*

$$\lambda_{i^*} \leq \frac{\Delta(x^*, x) + \varepsilon + a}{\sum_{j=1}^{N} p_j(\Delta_j(x^*, x) + \varepsilon + a)^2}$$

*Proof of Lemma M.6.* When the $\varepsilon$-best arm set is a singleton, (M.20) becomes (where $a$ is sufficiently small)

$$\sum_{j=1}^{N} \frac{p_j}{1 + \lambda_{x^*}(\Delta_j(x^*, x) + \varepsilon + a)} - 1 = 0$$

As (M.19) is a probability distribution, we need $1 + \lambda_{x^*}(\Delta_{j_0}(x^*, x) + \varepsilon + a) \geq 0$, which indicates $\lambda_{x^*} \leq \frac{-1}{\Delta_j(x^*, x) + \varepsilon + a}$. By the condition on $j_0$ and $j_1$, $0 < \lambda_{x^*}(\Delta_j(x^*, x) + \varepsilon + a) < 1, \forall j \in [N]$.

Therefore, by using $\frac{1}{1+x} = 1 - x + x^2 - x^3 + \frac{x^4}{1+x}$ for $|x| < 1$, we can expand the equation as follows

$$
\begin{aligned}
1 &= \sum_{j=1}^{N} \frac{p_j}{1 + \lambda_{x^*}(\Delta_j(x^*, x) + \varepsilon + a)} \\
&\geq \sum_{j=1}^{N} p_j \left( \left(1 - \lambda_{x^*}(\Delta_j(x^*, x) + \varepsilon + a) + (\lambda_{x^*}(\Delta_j(x^*, x) + \varepsilon + a))^2 \right. \right. \\
&\quad \left. \left. - (\lambda_{x^*}(\Delta_j(x^*, x) + \varepsilon + a))^3 \right) \right) \\
\Leftrightarrow \quad 0 &\geq -\lambda_{x^*}^2 \sum_{j=1}^{N} p_j(\Delta_j(x^*, x) + \varepsilon + a)^3 + \lambda_{x^*} \sum_{j=1}^{N} p_j(\Delta_j(x^*, x) + \varepsilon + a)^2 \\
&\quad - \sum_{j=1}^{N} p_j(\Delta_j(x^*, x) + \varepsilon + a) \\
\Rightarrow \quad \lambda_{x^*} &\leq \frac{\sum_{j=1}^{N} p_j(\Delta_j(x^*, x) + \varepsilon + a)}{\sum_{j=1}^{N} p_j(\Delta_j(x^*, x) + \varepsilon + a)^2} = \frac{\Delta(x^*, x) + \varepsilon + a}{\sum_{j=1}^{N} p_j(\Delta_j(x^*, x) + \varepsilon + a)^2}.
\end{aligned}
$$

$\square$

In general, when we get the $\{\lambda_{x_\varepsilon}\}_{x_\varepsilon \in \mathcal{X}_\varepsilon}$ and plug it in (M.19), the alternative distribution $q$ is obtained. Finally, we let $a \to 0$.

This gives the lower bound for the number of changepoints that need to be observed. A coarse estimation of the KL divergence without solving (M.20) can be done as follows

$$
\begin{aligned}
\sum_{j=1}^{N} p_j \ln \frac{p_j}{q_j} &= \sum_{j=1}^{N} p_j \ln \left(1 + \sum_{x_\varepsilon \in \mathcal{X}_\varepsilon} \lambda_{x_\varepsilon}(\Delta_j(x_\varepsilon, x) + \varepsilon) \right) \\
&\leq \sum_{j=1}^{N} p_j \sum_{x_\varepsilon \in \mathcal{X}_\varepsilon} \lambda_{x_\varepsilon}(\Delta_j(x_\varepsilon, x) + \varepsilon) \\
&= \sum_{x_\varepsilon \in \mathcal{X}_\varepsilon} \lambda_{x_\varepsilon}(\Delta(x_\varepsilon, x) + \varepsilon).
\end{aligned}
$$

Note that the solution of $\lambda_{x_\varepsilon}$ depends on $\Delta(x_\varepsilon, x)$ (e.g. Lemma M.6), so the final solution is of order $(\Delta(x_\varepsilon, x) + \varepsilon)^2$ for a given $x \notin \mathcal{X}$. This is the solution to the inside minimization problem in (M.17). The final lower bound will be

$$\min_{\Lambda' \in \mathrm{Alt}_P(\Lambda)} \mathrm{KL}(P_\theta, P'_\theta) = \min_{x \in \mathcal{X} \setminus \mathcal{X}_\varepsilon} \min_{\Lambda_x} \mathrm{KL}(P_\theta, P_\theta^x)$$

$$\leq \min_{x \notin \mathcal{X}_\varepsilon} \sum_{x_\varepsilon \in \mathcal{X}_\varepsilon} \lambda_{x_\varepsilon}(\Delta(x_\varepsilon, x) + \varepsilon).$$

The lower bound is

$$\mathbb{E}[l_t] \geq \max_{x \notin \mathcal{X}_\varepsilon} \frac{1}{\sum_{x_\varepsilon \in \mathcal{X}_\varepsilon} \lambda_{x_\varepsilon}(\Delta(x_\varepsilon, x) + \varepsilon)} \ln \frac{1}{4\delta}$$

where $\lambda_{x_\varepsilon}$ is the solution to (M.20).

With the setup in Lemma M.6, we have

$$\min_{\Lambda' \in \mathrm{Alt}_P(\Lambda)} \mathrm{KL}(P_\theta, P'_\theta) = \min_{x \in \mathcal{X} \setminus \mathcal{X}_\varepsilon} \min_{\Lambda_x} \mathrm{KL}(P_\theta, P_\theta^x)$$

$$\leq \min_{x \neq x_\varepsilon} \frac{(\Delta(x^*, x) + \varepsilon)^2}{\sum_{j=1}^N p_j(\Delta_j(x^*, x) + \varepsilon)^2}$$

and the lower bound is

$$\mathbb{E}[l_t] \geq \max_{x \neq x_\varepsilon} \frac{\sum_{j=1}^N p_j(\Delta_j(x^*, x) + \varepsilon)^2}{(\Delta(x^*, x) + \varepsilon)^2} \ln \frac{1}{4\delta}.$$

**Remark M.7.** *We give some comments on the existence and uniqueness of the solution to* (M.20).

*Existence:*

- *It is possible that* (M.20) *does not have a solution. For instance, consider a three-arm instance: $x_{(1)} = (1, 0.5), x_{(2)} = (0.5, 1), x_{(3)} = (0.6, 0.6), \theta_1^* = (1, 0), \theta_2^* = (0, 1), P_\theta = (0.5, 0.5), \varepsilon = 0.1$. We have $x_{(1)}^\top \mathbb{E}_{\theta \sim P_\theta} = x_{(2)}^\top \mathbb{E}_{\theta \sim P_\theta} = 0.75$ and $x_{(3)}^\top \mathbb{E}_{\theta \sim P_\theta} = 0.6$, thus $x_{(3)}$ is not an $\varepsilon$-best arm. Furthermore, there does not exist an alternative distribution $q$, such that $x_{(3)}$ is the best arm and neither $x_{(1)}, x_{(2)}$ is $\varepsilon$-best. Under such case, the lower bound on $\mathbb{E}[l_t]$ is 0 (we regard $\min_{x \in \emptyset} f(x) = +\infty$ by convention).*
- *It is possible that* (M.20) *does not have a solution and it is unnecessary to estimate $P_\theta$. For instance, consider a two-arm instance: $x_{(1)} = (1, 0.5), x_{(2)} = (0.5, 0.1), \theta_1^* = (1, 0), \theta_2^* = (0, 1), P_\theta = (0.5, 0.5), \varepsilon = 0.1$. Arm $x_{(1)}$ is better than arm $x_{(2)}$ under all contexts. Therefore, no matter what $P_\theta$ is, arm $x_{(1)}$ is the $\varepsilon$-best arm. Under such cases, there may exist an algorithm and it is sufficient for it to determine the best arm if the context vectors are well-approximated. There is no need to estimate $P_\theta$.*
- *A necessary condition for the existence of the solution of* (M.20) *is: there exists $x \notin \mathcal{X}_\varepsilon$, for any $x_\varepsilon \in \mathcal{X}_\varepsilon$, there $\exists j(x_\varepsilon) \in [N], s.t. \Delta_{j(x_\varepsilon)}(x_\varepsilon, x) < -\varepsilon$. In other words, for each $\varepsilon$-best arm $x_\varepsilon$, there is at least one context in which the alternative arm $x$ is better than $x_\varepsilon$ by at least $\varepsilon$.*
- *A sufficient condition for the existence of the solution of* (M.20) *is: there exists $x \notin \mathcal{X}_\varepsilon$ and $j \in [N], s.t. \Delta_j(x_\varepsilon, x) < -\varepsilon, \forall x_\varepsilon \in \mathcal{X}_\varepsilon$. In other words, $x$ is better than any $\varepsilon$-best arm under context $j$. In the alternative instance, we can lift $P'_\theta[\theta_j^*]$ close to 1 so that arm $x$ becomes the $\varepsilon$-best arm and $\mathcal{X}'_\varepsilon \cap \mathcal{X}_\varepsilon = \emptyset$.*

*Uniqueness:*

- *The uniqueness of the solution is not guaranteed, as the KKT conditions* (M.18) *is only a necessary condition for the solution. We need to look for the solution that minimizes the* KL *divergence.*
- *A sufficient condition for the uniqueness of the solution is (if it exists, which indicates $\exists j \in [N], s.t. \Delta_j(x^*, x) < -\varepsilon$): $|\mathcal{X}_\varepsilon| = 1$. Specifically, denote $f(\lambda_{x^*}) := \sum_{j=1}^N \frac{p_j}{1 + \lambda_{x^*}(\Delta_j(x^*, x) + \varepsilon + a)}$, we have*

$$\frac{\partial f}{\partial \lambda_{x^*}} = \sum_{j=1}^N \frac{-p_j(1 + \lambda_{x^*}(\Delta_j(x^*, x) + \varepsilon + a))}{1 + \lambda_{x^*}(\Delta_j(x^*, x) + \varepsilon + a)}$$

$$\frac{\partial^2 f}{\partial \lambda_{x^*}^2} = \sum_{j=1}^{N} \frac{2p_j(1 + \lambda_{x^*}(\Delta_j(x^*, x) + \varepsilon + a))^2}{1 + \lambda_{x^*}(\Delta_j(x^*, x) + \varepsilon + a)} > 0$$

As $f(0) = 1$, $\frac{\partial f}{\partial \lambda_{x^*}}(0) = -1$ and $\frac{\partial^2 f}{\partial \lambda_{x^*}^2} > 0$, so there is exactly 1 solution $\lambda_{x^*} > 0$.

## N    More Examples and Details

In this section, we **firstly** provide one more example to illustrate the tightness of our derived upper bound lower bounds, indicating the efficiency of our PS$\varepsilon$BAI$^+$ algorithm. In addition, we can observe how the upper and lower bounds are affected by the level of piecewise non-stationarity and whether our PS$\varepsilon$BAI$^+$ algorithm can reduce the influence manifested by $L_{\max}$. **After that**, we present a proof sketch of the results in Corollaries 5.1 and N.1 in Appendix N.1.

**Example 2.** *Instance* $\Lambda = (\mathcal{X}, \Theta, P_\theta, \mathcal{C})$ *is with*

- *$d$ arms: $x_{(i)} = \mathbf{e}_i, i \in [d]$.*
- *$N = d$ contexts: $\theta_1^* = (a, 0, 0, \ldots, 0)^\top$, $\theta_2^* = (a, b, b, \ldots, b)^\top - b\mathbf{e}_j, j \geq 2$, where $b > a > \varepsilon, b - a > \varepsilon$.*
- *Context distribution: $p_j = p, j \geq 2$ and $p_1 = 1 - (N-1)p$, where $p \in (0, \frac{a-\varepsilon}{(N-2)b})$.*

Under Example 2, we have

$$\Delta(x_{(1)}, x_{(i)}) = (1 - (N-2)p) \cdot a - (N-2)p \cdot (b - a) > \varepsilon$$
$$\Delta_j(x_{(1)}, x_{(i)}) = -b + a < -\varepsilon, \quad i \geq 2, j \neq 1, i$$

Thus, (1) $x_{(1)}$ is the unique $\varepsilon$-best arm; (2) $\{x_{(i)}\}_{i\geq 2}$ are equivalent, and $\Delta_{\min} := \Delta(x_{(1)}, x_{(i)})$; (3) for any $i \geq 2$, $x_{(i)}$ can be an $\varepsilon$-optimal arm under some alternative distributions.

**Corollary N.1.** *Firstly, for the instances defined in Example 2, we have*

$$\mathrm{H_{DE}} \leq \frac{16(N-2)L_{\max}}{(\Delta_{\min} + \varepsilon)^2}\left((a + \varepsilon)^2 + (b - a - \varepsilon)^2\right),$$

*and the sample complexity of the PS$\varepsilon$BAI$^+$ is tight up to $(NL_{\max}/L_{\min})$ and logarithmic factors. We also further observe some specific instances: (i) when $p \to 0^+$, with $\Delta_{\min} = \min_{x \neq x^*} \Delta(x^*, x)$, we have*

$$\frac{\mathbb{E}[\tau]^*}{\ln(1/\delta)} \in \tilde{O}\left(\min\left\{\frac{d}{(\Delta_{\min} + \varepsilon)^2} + \frac{NL_{\max}}{\Delta_{\min} + \varepsilon}, \frac{L_{\max} + d}{(\Delta_{\min} + \varepsilon)^2}\right\}\right)$$

$$\bigcap \Omega\left\{\max\left\{\frac{d}{(\Delta_{\min} + \varepsilon)^2}, \frac{L_{\min}(b - a - \varepsilon)}{\Delta_{\min} + \varepsilon}\right\}\right\}.$$

*(ii) When $p \to \left(\frac{a-\varepsilon}{(N-2)b}\right)^-$ and $(a + \varepsilon)^2 + (b - a - \varepsilon)^2 = \Omega(1)$, we have $\mathrm{H_{DE}} = \frac{NL_{\max}}{\left(\Delta(x_{(1)}, x_{(i)}) + \varepsilon\right)^2}$ and*

$$\frac{\mathbb{E}[\tau]^*}{\ln(1/\delta)} \in \tilde{O}\left(\min\left\{\mathrm{H_{DE}}, \frac{d + L_{\max}}{\varepsilon^2}\right\}\right) \bigcap \Omega\left(\frac{d + L_{\min}}{\varepsilon^2}\right).$$

*The upper bounds are achieved by the PS$\varepsilon$BAI$^+$ algorithm.*

We can observe from Corollary N.1 that

- In case (i), when $p \to 0^+$, $p_1 \to 1 - \frac{(N-1)(a-\varepsilon)}{(N-2)b}$ and $p_j \to \left(\frac{a-\varepsilon}{(N-2)b}\right)$, and thus the instance tends to be non-stationary and $\Delta_{\min} \to \varepsilon$. We will obtain

$$\frac{\mathbb{E}\tau}{\ln(1/\delta)} \in \tilde{\Theta}\left(\frac{d}{(\Delta_{\min} + \varepsilon)^2}\right),$$

  indicating that our algorithm can also reduce the impact of $L_{\max}$.

- In case (ii), the upper and lower bounds are with the same order, and the difference is solely manifested by a additive term $L_{\max} - L_{\min}$, suggesting that PS$\varepsilon$BAI$^+$ is near optimal and again, PS$\varepsilon$BAI$^+$ mitigates the impact of $L_{\max}$.

## N.1 Analysis of examples

Recall the instance $\Lambda = (\mathcal{X}, \Theta, P_\theta, \mathcal{C})$ in Example 1:

**Example 1.** *Instance* $\Lambda = (\mathcal{X}, \Theta, P_\theta, \mathcal{C})$ *is with (i)* $2d-1$ *arms:* $x_{(1)} = \mathbf{e}_1, x_{(i)} = \mathbf{e}_i, x_{(d+i-1)} = \mathbf{e}_1 \cos \phi + \mathbf{e}_i \sin \phi$ *for all* $i \in \{2, \dots, d\}$ *where* $\phi \in [0, \pi/4)$, *(ii)* $2d-2$ *contexts:* $\theta^*_{j\pm} = \mathbf{e}_1 \cos \phi \pm \mathbf{e}_{j+1} \sin \phi$ *for all* $j \in [d-1]$, *(iii) Context distribution:* $p_j = 1/N$ *for all* $j \in [N]$.

**Corollary 5.1.** *For the instance defined in Example 1, we have* $\mathrm{H}_{\mathrm{DE}}(x_\varepsilon, x) = \tilde{O}(NL_{\max})$ *for all* $(x_\varepsilon, x) \in \mathcal{X}_\varepsilon \times (\mathcal{X} \setminus \mathcal{X}_\varepsilon)$. *In addition, if* $\varepsilon < (\cos \phi)(1 - \cos \phi)$, *we have*

$$\frac{\mathbb{E}[\tau]^*}{\ln(1/\delta)} \in \tilde{\Theta}\left( (1 + f(\phi)) \cdot \frac{d}{(\Delta_{x_{(1)}, x_{(d+1)}} + \varepsilon)^2} \right), \tag{5.1}$$

*where* $\mathbb{E}[\tau]^*$ *is the minimal expected sample complexity over all* $(\varepsilon, \delta)$-*PAC algorithms and* $f : \mathbb{R} \to \mathbb{R}$ *satisfies* $f(\phi) \to 0$ *as* $\phi \to 0^+$. *The upper bound in (5.1) is achieved by* PS$\varepsilon$BAI$^+$.

*Proof of Corollary 5.1.* As $\mu_{x_{(1)}} = \cos \phi, \mu_{x_{(i)}} = 0, \mu_{x_{(d+i)}} = \cos^2 \phi$ for $i = 2, \dots, d$ and $\varepsilon \in (1 - \cos \phi, \cos \phi)$, $x_{(1)}$ is the best arm and $x_{(i)}, i = 2, \dots, d$ are not $\varepsilon$-best arms and $x_{(d+i)}, i = 2, \dots, d$ are $\varepsilon$-best arms. $\Delta_{\min} = \cos \phi - \cos^2 \phi$.

When $\varepsilon < \cos \phi - \cos^2 \phi = \Delta_{\min}$, $x_{(1)}$ is the unique $\varepsilon$-best arm. By solving (M.14), we see that there exists a continuous function $f(\phi)$ such that $f(\phi) \to 0$ as $\phi \to 0$ and we can get the lower bound for the VE term as

$$\Omega\left( (1 + f(\phi)) \cdot \frac{d}{(\Delta(x_{(1)}, x_{(d+1)}) + \varepsilon)^2} \ln \frac{1}{\delta} \right). \tag{N.1}$$

As $(x_{(1)} - x_{(i)})^\top \theta^*_{j\pm} > 0$ for $i = 2, \dots, d, j \in [d-1]$, $x_{(i)}$ cannot be an $\varepsilon$-best arm under any alternative distribution, we only need to consider $x_{(d+i)}, i = 2, \dots, d$, which are equivalent. Given any $x_{(d+i)}$, by solving (M.20), we can upper bound

$$\lambda_{x_{(1)}} \leq \frac{\Delta_{\min} + \varepsilon + \sin^2 \phi}{-(\Delta_{i+}(x_{(1)}, x_{(d+i)}) + \varepsilon)(\Delta_{i-}(x_{(1)}, x_{(d+i)}) + \varepsilon)},$$

and thus the number of change points or context samples can be lower bounded as

$$\mathbb{E}[l_\tau] \geq \frac{-(\Delta_{i+}(x_{(1)}, x_{(d+i)}) - \varepsilon)(\Delta_{i-}(x_{(1)}, x_{(d+i)}) + \varepsilon)}{(\Delta_{\min} + \varepsilon + \sin^2 \phi)(\Delta_{\min} + \varepsilon)}$$

$$= \frac{(1 - \cos \phi - \varepsilon)(1 + \cos \phi - 2\cos^2 \phi + \varepsilon)}{(\Delta_{\min} + \varepsilon + \sin^2 \phi)(\Delta_{\min} + \varepsilon)} = \frac{1 - \cos \phi - \varepsilon}{\Delta_{\min} + \varepsilon} = O(1).$$

According to Theorem 3.3, the upper bound is

$$\tilde{O}\left( \frac{d}{(\Delta(x_{(1)}, x_{(d+1)}) + \varepsilon)^2} \ln \frac{1}{\delta} + \mathrm{H}_{\mathrm{DE}}(\Delta(x_{(1)}, x_{(d+1)})) \ln \frac{1}{\delta} + \frac{NL_{\max}}{\Delta(x_{(1)}, x_{(d+1)}) + \varepsilon} \ln \frac{1}{\delta} \right)$$

$$= \tilde{O}\left( \frac{d}{(\Delta_{\min} + \varepsilon)^2} \ln \frac{1}{\delta} + \mathrm{H}_{\mathrm{DE}}(\Delta(x_{(1)}, x_{(d+1)})) \ln \frac{1}{\delta} + \frac{NL_{\max}}{\Delta_{\min} + \varepsilon} \ln \frac{1}{\delta} \right)$$

where

$$\mathrm{H}_{\mathrm{DE}}(\Delta(x_{(1)}, x_{(d+1)})) = \begin{cases} 16NL_{\max}, & \varepsilon \geq 1 - \cos \phi \\ \dfrac{16NL_{\max}(\Delta_{\min} + \varepsilon + 2(1 - \cos \phi)/N)^2}{(\Delta_{\min} + \varepsilon)^2}, & \varepsilon < 1 - \cos \phi \end{cases}$$

As $\Delta_{\min} = \cos \phi - \cos^2 \phi = \cos \phi(1 - \cos \phi)$, thus $1 - \cos \phi = \frac{\Delta_{\min}}{\cos \phi} \leq \sqrt{2}\Delta_{\min}$, and $\mathrm{H}_{\mathrm{DE}}(\Delta(x_{(1)}, x_{(d+1)})) < 144NL_{\max}$ for any choice of $\varepsilon$.

Hence, the upper bound is

$$\mathbb{E}[\tau] = \tilde{O}\left(\frac{d}{(\Delta_{\min} + \varepsilon)^2} \ln\frac{1}{\delta} + NL_{\max}\ln\frac{1}{\delta} + \frac{NL_{\max}}{\Delta_{\min} + \varepsilon}\ln\frac{1}{\delta}\right)$$

$$= \tilde{O}\left(\frac{d}{(\Delta_{\min} + \varepsilon)^2}\ln\frac{1}{\delta} + \frac{NL_{\max}}{\Delta_{\min} + \varepsilon}\ln\frac{1}{\delta}\right).$$

As $\phi \to 0$, the first term (the VE term) dominates and it matches the lower bound in (N.1). Therefore, the upper bound is asymptotically tight up to logarithmic terms. $\qquad\square$

**Corollary N.1.** *Firstly, for the instances defined in Example 2, we have*

$$\mathrm{H_{DE}} \le \frac{16(N-2)L_{\max}}{(\Delta_{\min} + \varepsilon)^2}\left((a+\varepsilon)^2 + (b-a-\varepsilon)^2\right),$$

*and the sample complexity of the $\mathrm{PS}\varepsilon\mathrm{BAI}^+$ is tight up to $(NL_{\max}/L_{\min})$ and logarithmic factors. We also further observe some specific instances: (i) when $p \to 0^+$, with $\Delta_{\min} = \min_{x \ne x^*}\Delta(x^*, x)$, we have*

$$\frac{\mathbb{E}[\tau]^*}{\ln(1/\delta)} \in \tilde{O}\left(\min\left\{\frac{d}{(\Delta_{\min} + \varepsilon)^2} + \frac{NL_{\max}}{\Delta_{\min} + \varepsilon}, \frac{L_{\max} + d}{(\Delta_{\min} + \varepsilon)^2}\right\}\right)$$

$$\bigcap\Omega\left\{\max\left\{\frac{d}{(\Delta_{\min} + \varepsilon)^2}, \frac{L_{\min}(b-a-\varepsilon)}{\Delta_{\min} + \varepsilon}\right\}\right\}.$$

*(ii) When $p \to \left(\frac{a-\varepsilon)}{(N-2)b}\right)^-$ and $(a+\varepsilon)^2 + (b-a-\varepsilon)^2 = \Omega(1)$, we have $\mathrm{H_{DE}} = \frac{NL_{\max}}{\left(\Delta(x_{(1)}, x_{(i)}) + \varepsilon\right)^2}$ and*

$$\frac{\mathbb{E}[\tau]^*}{\ln(1/\delta)} \in \tilde{O}\left(\min\left\{\mathrm{H_{DE}}, \frac{d + L_{\max}}{\varepsilon^2}\right\}\right)\bigcap\Omega\left(\frac{d + L_{\min}}{\varepsilon^2}\right).$$

*The upper bounds are achieved by the $\mathrm{PS}\varepsilon\mathrm{BAI}^+$ algorithm.*

*Proof of Corollary N.1.* Under Example 2, we have

$$\Delta(x_{(1)}, x_{(i)}) = (1 - (N-2)p) \cdot a - (N-2)p \cdot (b-a) > \varepsilon$$
$$\Delta_j(x_{(1)}, x_{(i)}) = -b + a < -\varepsilon, \quad i \ge 2, j \ne 1, i$$

Thus, (1) $x_{(1)}$ is the unique $\varepsilon$-best arm; (2) $\{x_{(i)}\}_{i\ge2}$ are equivalent, and $\Delta_{\min} := \Delta(x_{(1)}, x_{(i)})$; (3) for any $i \ge 2$, $x_{(i)}$ can be an $\varepsilon$-best arm under some alternative distributions.

For any $i \ge 2$, by solving (M.20) with $x_\varepsilon = x_{(1)}, x = x_{(i)}$, we obtain $\lambda_{x_{(1)}} = \frac{\Delta(x_{(1)}, x_{(i)}) + \varepsilon}{(a+\varepsilon)(b-a-\varepsilon)}$ and the alternative distribution

$$P'_\theta[\theta_j^*] = \frac{p_j}{1 + \lambda_{x_{(1)}}(\Delta_j(x_{(1)}, x_{(i)}) + \varepsilon)}.$$

The lower bound on the expected number of changepoints is

$$\mathbb{E}[l_\tau] \ge \frac{(a+\varepsilon)(b-a-\varepsilon)}{\left(\Delta(x_{(1)}, x_{(i)}) + \varepsilon\right)^2}\ln\frac{4}{\delta}.$$

Thus the time complexity is lower bounded by

$$\mathbb{E}[\tau] \ge L_{\min} \cdot \frac{(a+\varepsilon)(b-a-\varepsilon)}{\left(\Delta(x_{(1)}, x_{(i)}) + \varepsilon\right)^2}\ln\frac{4}{\delta}.$$

Furthermore, by solving (M.14), we obtain the lower bound on the expected sample complexity when the context index $j_t$ is revealed:

$$\mathbb{E}[\tau] \ge 2\log\frac{1}{2.4\delta}\min_{\bar{v}\in\Delta_K}\max_{i\ne1}\frac{\frac{1}{\bar{v}_1} + \frac{1}{\bar{v}_i}}{\left(\Delta(x_{(1)}, x_{(i)}) + \varepsilon\right)^2} \ge 2\log\frac{1}{2.4\delta}\frac{(\sqrt{d-1}+1)^2}{\left(\Delta(x_{(1)}, x_{(i)}) + \varepsilon\right)^2}$$

$$\ge \frac{d}{\left(\Delta(x_{(1)}, x_{(i)}) + \varepsilon\right)^2} \cdot 2\log\frac{1}{2.4\delta}.$$

We get the lower bound on the expected sample complexity:

$$\mathbb{E}[\tau] \geq \max\left\{\frac{d}{(\Delta_{\min} + \varepsilon)^2} \cdot 2\log\frac{1}{2.4\delta}, \frac{L_{\min}(a + \varepsilon)(b - a - \varepsilon)}{(\Delta_{\min} + \varepsilon)^2}\ln\frac{4}{\delta}\right\}. \tag{N.2}$$

According to Theorem 3.3, the upper bound on the expected sample complexity is

$$\tilde{O}\left(\min\left\{\frac{d}{(\Delta_{\min} + \varepsilon)^2}\ln\frac{1}{\delta} + H_{DE}\ln\frac{1}{\delta} + \frac{NL_{\max}}{\Delta_{\min} + \varepsilon}\ln\frac{1}{\delta}, \frac{L_{\max} + d}{(\Delta_{\min} + \varepsilon)^2}\ln\frac{1}{\delta}\right\}\right) \tag{N.3}$$

$$\leq \tilde{O}\left(\min\left\{\frac{d}{(\Delta_{\min} + \varepsilon)^2}\ln\frac{1}{\delta} + \frac{NL_{\max}\left((a + \varepsilon)^2 + (b - a - \varepsilon)^2\right)}{(\Delta_{\min} + \varepsilon)^2}\ln\frac{1}{\delta}\right.\right.$$

$$\left.\left. + \frac{NL_{\max}}{\Delta_{\min} + \varepsilon}\ln\frac{1}{\delta}, \frac{L_{\max} + d}{(\Delta_{\min} + \varepsilon)^2}\ln\frac{1}{\delta}\right\}\right)$$

where we utilize

$$H_{DE} = \frac{L_{\max}}{(\Delta_{\min} + \varepsilon)^2}\left(\sqrt{\min\left\{16p_j, \frac{1}{4}\right\}}|a + \varepsilon| + \sqrt{\min\left\{16p, \frac{1}{4}\right\}}|a + \varepsilon|\right.$$

$$\left. + (N - 2)\sqrt{\min\left\{16p, \frac{1}{4}\right\}}|b - a - \varepsilon|\right)^2$$

$$\leq \frac{16(N - 2)L_{\max}}{(\Delta_{\min} + \varepsilon)^2}\left((a + \varepsilon)^2 + (b - a - \varepsilon)^2\right). \tag{N.4}$$

By comparing the lower bound in (N.2) and the upper bound (N.3), we conclude that

- the sample complexity of PS$\varepsilon$BAI$^+$ is tight up to $\frac{NL_{\max}}{L_{\min}}$ and logarithmic factors.
- When the mean gap is small, the sample complexity of PS$\varepsilon$BAI$^+$ is dominated by the former term, i.e., the design of PS$\varepsilon$BAI.
- When $p \to 0^+$, then $p_1 \to 1, p_j \to 0$ for $j \geq 2$, so the instance tends to be stationary and $\Delta_{\min} \to a$. The lower bound in (N.2) becomes

$$\mathbb{E}[\tau] \geq \max\left\{\frac{d}{(\Delta_{\min} + \varepsilon)^2} \cdot 2\log\frac{1}{2.4\delta}, \frac{L_{\min}(b - a - \varepsilon)}{\Delta_{\min} + \varepsilon}\ln\frac{4}{\delta}\right\}$$

and the upper bound in (N.3) turns into

$$\tilde{O}\left(\min\left\{\frac{d}{(\Delta_{\min} + \varepsilon)^2}\ln\frac{1}{\delta} + \frac{NL_{\max}}{\Delta_{\min} + \varepsilon}\ln\frac{1}{\delta}, \frac{L_{\max} + d}{(\Delta_{\min} + \varepsilon)^2}\ln\frac{1}{\delta}\right\}\right).$$

The vector estimation term dominates the sample complexity and our upper bound is tight.

- When $p \to \left(\frac{a - \varepsilon}{(N - 2)b}\right)^-$, then $p_1 \to 1 - \frac{(N - 1)(a - \varepsilon)}{(N - 2)b}$ and $p_j \to \left(\frac{a - \varepsilon}{(N - 2)b}\right)$, so the instance tends to be non-stationary and $\Delta_{\min} \to \varepsilon$. The lower bound in (N.2) becomes

$$\mathbb{E}[\tau] \geq \max\left\{\frac{d}{4\varepsilon^2} \cdot 2\log\frac{1}{2.4\delta}, \frac{L_{\min}(a + \varepsilon)(b - a - \varepsilon)}{4\varepsilon^2}\ln\frac{4}{\delta}\right\}$$

and the upper bound in (N.3) turns into

$$\tilde{O}\left(\min\left\{\frac{d}{4\varepsilon^2}\ln\frac{1}{\delta} + H_{DE} \cdot \ln\frac{1}{\delta} + \frac{NL_{\max}}{2\varepsilon}\ln\frac{1}{\delta}, \frac{L_{\max} + d}{4\varepsilon^2}\ln\frac{1}{\delta}\right\}\right)$$

where $H_{DE}$ is upper bounded by (N.4).

(1) If $\left((a + \varepsilon)^2 + (b - a - \varepsilon)^2\right) = O(4\varepsilon^2 + NL_{\max})$, DE is independent of $\varepsilon$ and VE increases as $\varepsilon$ decreases, thus the expected sample complexity is dominated by vector estimation term when $\varepsilon$ is small;

(2) If $\left((a + \varepsilon)^2 + (b - a - \varepsilon)^2\right) = \Omega(1)$, the expected sample complexity is dominated by

distribution estimation term.

Under both scenarios, our upper bound is tight.

$\square$

# O    Experimental Details

For the computation of the G-optimal allocation, we adopt the Wolfe–Atwood Algorithm with the Kumar–Yildirim start introduced in [33], where the input to the function is the arm set and the output is the G-optimal allocation. All experiments are conducted via MATLAB R2021b on a MacBook Pro with Apple M1 Pro chip and 16 GB memory.

To shorten the execution time,

- Before PS$\varepsilon$BAI stops, PS$\varepsilon$BAI and N$\varepsilon$BAI are conducted in parallel (i.e., we run PS$\varepsilon$BAI$^+$). After PS$\varepsilon$BAI stops, N$\varepsilon$BAI continues and is run in a batch manner, i.e., we sample $L_{\min}$ samples according to the G-optimal allocation one time and update the statistics. As the sample complexity of N$\varepsilon$BAI is of order at least $10^7$ and each stationary segment is of order $10^4$, the effect of this batch sampling procedure can be largely ignored.
- D$\varepsilon$BAI and D$\varepsilon$BAI$_\beta$ are both conducted in segments, because the latent context vector can be observed by the two algorithms and the latent vector does not change within a stationary segment.
- As the experiment in Section 6 illustrates that PS$\varepsilon$BAI outperforms N$\varepsilon$BAI and dominates the PS$\varepsilon$BAI$^+$ algorithm, we run PS$\varepsilon$BAI instead of PS$\varepsilon$BAI$^+$ for the addition experiments in Section O.2 and Section O.3.

To increase the robustness of the algorithm, the window size for LCD is doubled. Note that this will only influence an absolute constant in the sample complexity of the proposed algorithm and the order of the sample complexity remains.

## O.1    Modification of Confidence Radii in PS$\varepsilon$BAI and PS$\varepsilon$BAI$^+$

During the proof of the upper bound, we have relaxed the absolute constants in the confidence radii to simplify the proof and increase the readability. In the experiments, we utilize the tighter confidence radii to gain better empirical performance. Note that when these tighter confidence radii are utilized by our algorithm, it still enjoys the current theoretical guarantee. The choice of confidence radii are as follows:

- $\alpha_t$: according to Lemma I.1 and Lemma E.1, $\alpha$ can be tightened to be

$$\alpha_t^{\mathrm{alg}} = \frac{d}{T_t} \ln \frac{2}{\delta_{v,T_t}} + \sqrt{\left( \frac{d}{T_t} \ln \frac{2}{\delta_{v,T_t}} \right)^2 + \frac{4d}{T_t} \ln \frac{2}{\delta_{v,T_t}}} \le 5\sqrt{\frac{d}{T_t} \ln \frac{2}{\delta_{v,T_t}}}.$$

- $\beta_{t,j}$: according to (I.3) and Lemma K.1, $\beta_{t,j}$ can be tightened to be

$$\beta_{t,j}^{\mathrm{alg}} = \min \left\{ \frac{\frac{1}{3} L_{\max} \ln \frac{2}{\delta_{d,T_t}} + \sqrt{\left( \frac{1}{3} L_{\max} \ln \frac{2}{\delta_{d,T_t}} \right)^2 + \frac{2\phi_{t,j} L_{\max}}{T_t} \ln \frac{2}{\delta_{d,T_t}}}}{T_t}, 1 \right\},$$

where $\phi_{t,j} := \min \left\{ 4 \max \left\{ \hat{p}_{t,j}, \frac{25}{4} \frac{L_{\max}}{T_t} \ln \frac{2}{\delta_{d,T_t}} \right\}, \frac{1}{4} \right\}$.

- $\xi_t$: instead of using $25\sqrt{2} \frac{N L_{\max}}{T_t} \ln \frac{2}{\delta_{m,T_t}}$, we turn to bound the residual error by (I.9):

$$\xi_t^{\mathrm{alg}} = \sum_{j:\psi_{t,j,1} > \hat{p}_{t,j}} 4 \cdot \beta_{t,j}^{\mathrm{alg}} + \sum_{\substack{j:\psi_{t,j,1} = \hat{p}_{t,j}, \\ \psi_{t,j,2} > \hat{p}_{t,j}}} \min \left\{ 4, 2\tilde{\xi}_{t,j}^{\mathrm{alg}} \right\} \cdot \beta_{t,j}^{\mathrm{alg}}$$

$$+ \sum_{j:\psi_{t,j,3} = \hat{p}_{t,j}} 10\sqrt{\frac{d}{T_{t,j}} \ln \frac{2}{\delta_{m,T_t}}} \cdot \beta_{t,j}^{\mathrm{alg}}$$

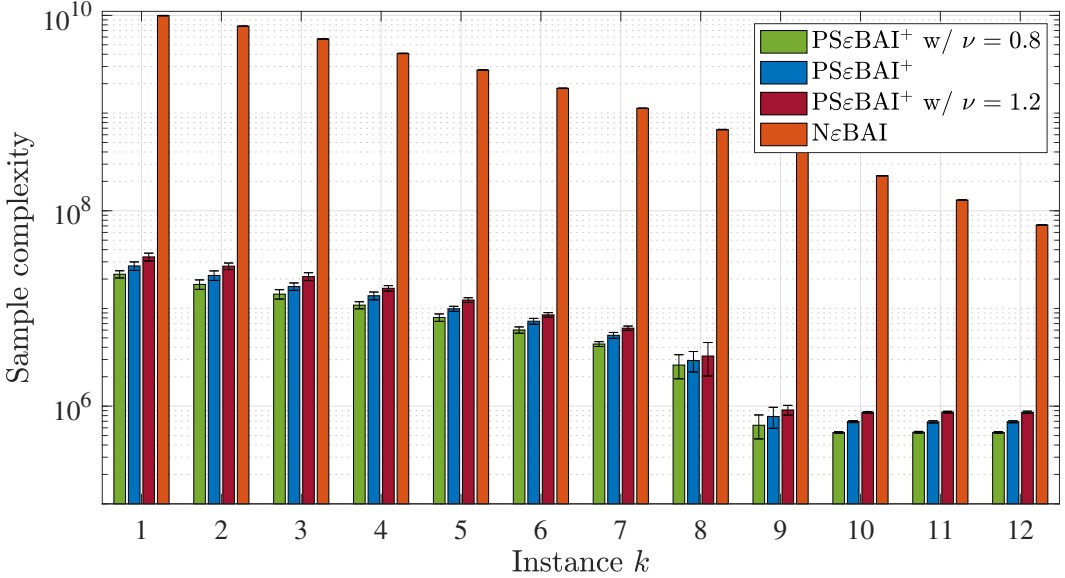

Figure 3: Misspeficied $L_{\max}$ and $L_{\min}$.

where

$$\psi_{t,j,1} := \max\left\{\hat{p}_{t,j}, \frac{d}{4T_t}\ln\frac{2}{\delta_{m,T_t}}\right\}, \ \psi_{t,j,2} := \max\left\{\hat{p}_{t,j}, \frac{25}{4}\frac{L_{\max}}{T_t}\ln\frac{2}{\delta_{d,T_t}}\right\},$$

$$\tilde{\xi}_{t,j}^{\text{alg}} = \frac{d}{T_{t,j}}\ln\frac{2}{\delta_{m,T_t}} + \sqrt{\left(\frac{d}{T_{t,j}}\ln\frac{2}{\delta_{m,T_t}}\right)^2 + \frac{4d}{T_{t,j}}\ln\frac{2}{\delta_{m,T_t}}} \leq 5\sqrt{\frac{d}{T_{t,j}}\ln\frac{2}{\delta_{m,T_t}}}.$$

$\tilde{\xi}_{t,j}^{\text{alg}}$ is obtained in a similar manner as $\alpha_t^{\text{alg}}$. $\xi_t^{\text{alg}}$ characterizes the confidence radii of each context at a finer level.

These finer confidence bounds can save a constant of $2.5$ when $t$ is large.

## O.2 Misspecified $L_{\max}$ and $L_{\min}$

As PS$\varepsilon$BAI$^+$ requires the knowledge of $L_{\max}$, we empirically test the robustness of the algorithm towards $L_{\max}$ on the instances in section 6. We run PS$\varepsilon$BAI with $\tilde{L}_{\min} = \nu L_{\min}$ and $\tilde{L}_{\max} = \nu L_{\max}$ where $\nu = 0.8$ or $1.2$. An $\varepsilon$-best arm is recommended in all experiments. The sample complexities are presented in Figure 3. The result indicates that PS$\varepsilon$BAI (thus PS$\varepsilon$BAI$^+$) is robust towards the knowledge of $L_{\max}$ and its superiority over N$\varepsilon$BAI is maintained.

## O.3 Robustness towards $w$ and $b$

According to the distinguishability condition (Assumption 1), point (2) indicates we can set $w = \frac{L_{\min}}{3\gamma}$, where $\gamma$ is the change detection frequency, and (3.5) indicates $w$ and $b$ are coupled. We denote $\tilde{w} := \frac{L_{\min}}{18}$.

To exam the robustness towards $w$ and $b$, we choose to vary the choice of $\gamma$, thus, $w$ and $b$ will change accordingly. Specifically, we select $\gamma \in \{2, 3, 6, 12\}$ and the corresponding $w \in \{3\tilde{w}, 2\tilde{w}, \tilde{w}, \frac{\tilde{w}}{2}\}$. The other parameters remain unchanged. The experiment result is presented in Figure 4. When $\gamma = 18, w = \frac{\tilde{w}}{3}$, Assumption 1 is severely violated and the result is not desirable.

The result indicates that, while smaller $\gamma$ and greater $w$ can result in slightly greater sample complexity, the overall sample complexity does not vary much and the superiority of our algorithm over the naive uniform sampling algorithm N$\varepsilon$BAI is maintained. We conclude that our algorithm is robust against these choices, as long as Assumption 1 is not severely violated.

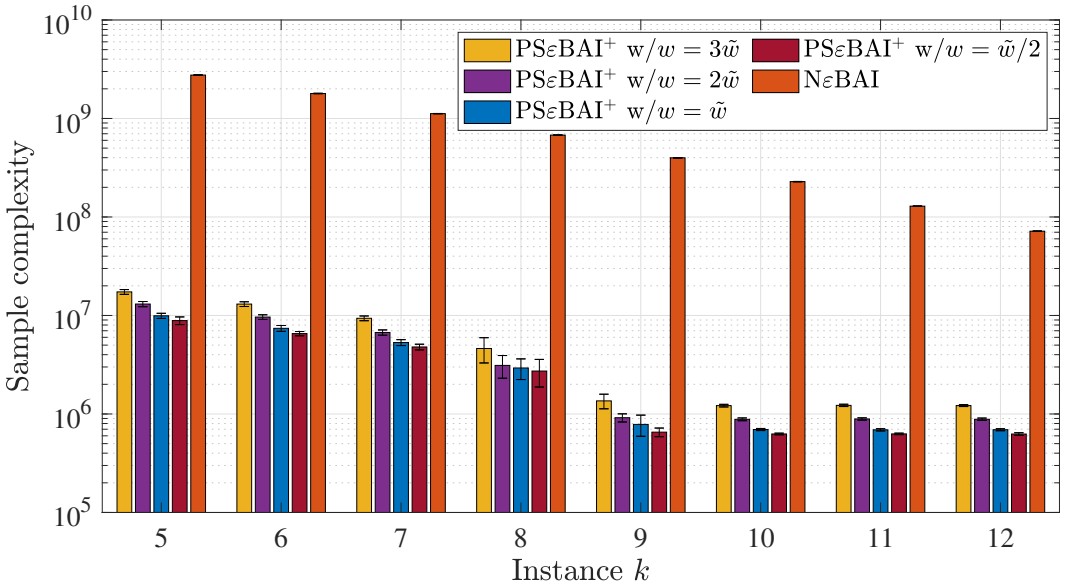

Figure 4: Robustness towards $w$ and $b$.

## O.4 Benchmarks: D$\varepsilon$BAI and D$\varepsilon$BAI$_\beta$

We present the Distribution $\varepsilon$-Best Arm Identification (D$\varepsilon$BAI) in Algorithm 8 and its variant D$\varepsilon$BAI$_\beta$ in Algorithm 9. According to Dynamics 3, the agent has access to the context vector $\theta^*_{j_t}$ and the changepoint $t \in \mathcal{C}$. Thus only the distribution $P_\theta$ remains unknown and the agent needs to estimate it via the observed contexts.

---

**Algorithm 8** DISTRIBUTION $\varepsilon$-BEST ARM IDENTIFICATION (D$\varepsilon$BAI)

---

1: **Input:** the arm set $\mathcal{X}$, the latent vector matrix $\Theta$, the slackness parameter $\varepsilon$, the confidence parameter $\delta$.
2: **Initialize**: Compute the G-optimal allocation $\lambda^*$.
3: Compute $C_3 = \sum_{n=1}^{\infty} n^{-3}$.
4: Observe $\theta^*_{j_1}$.
5: Compute

$$\ddot{p}_{t,j} = \sum_{l_s=1}^{l_t} \frac{\mathbb{1}\{c_{l_s} = c_j\}}{l_t}, \quad \ddot{x}_t = \arg\max_{x \in \mathcal{X}} x^\top \Theta \ddot{p}_t \tag{O.1}$$

$$\ddot{\beta}_t = \sqrt{\frac{1}{2l_t} \ln \frac{2C_3 N l_t^3}{\delta}}, \quad \ddot{\rho}_t(\ddot{x}_t, x) = \ddot{\beta}_t \min_{\zeta(\ddot{x}_t, x) \in \mathbb{R}} \sum_{j=1}^{N} |\Delta_j(\ddot{x}_t, x) + \zeta(\ddot{x}_t, x)|$$

6: **while** $\min_{x \neq \ddot{x}_t} (\ddot{x}_t - x)^\top \Theta \ddot{p}_t - \ddot{\rho}_t < -\varepsilon$ or $t \notin \mathcal{C}$ **do**
7:    Observe $\theta^*_{j_t}$ and $\mathbb{1}\{t \in \mathcal{C}\}$ update $t = t + 1$.
8:    Update $\ddot{x}_t$ and $\ddot{\rho}_t$ with (O.1) and update $l_t$ if $t \in \mathcal{C}$.
9: **end while**
10: Recommend arm $\ddot{x}_\varepsilon = \ddot{x}_t$.

---

The design is straightforward except for $\zeta(\ddot{x}_t, x)$. It minimizes the summation $\zeta(\ddot{x}_t, x) := \arg\min_{y \in \mathbb{R}} \sum_{j=1}^{N} |\Delta_j(\ddot{x}_t, x) + \zeta(\ddot{x}_t, x)|$. This trick has also been utilized in the design of PS$\varepsilon$BAI (see (I.5)). It helps to better exploit the structure of the latent vectors $\Theta$ and facilitates the identification process. For easy implementation, we choose a proxy $\zeta(\ddot{x}_t, x) = \arg\min_{y \in \mathbb{R}} \sum_{j=1}^{N} (\Delta_j(\ddot{x}_t, x) + \zeta(\ddot{x}_t, x))^2 = -\frac{1}{N} \sum_{j=1}^{N} \Delta_j(\ddot{x}_t, x)$

---

**Algorithm 9** DISTRIBUTION $\varepsilon$-BEST ARM IDENTIFICATION-$\beta$ (D$\varepsilon$BAI$_\beta$)

---

1: **Input:** the arm set $\mathcal{X}$, the latent vector matrix $\Theta$, the slackness parameter $\varepsilon$, the confidence parameter $\delta$.
2: **Initialize**: Compute the G-optimal allocation $\lambda^*$.
3: Compute $C_3 = \sum_{n=1}^{\infty} n^{-3}$.
4: Observe $\theta_{j_1}^*$.
5: Compute

$$\mathring{p}_{t,j} = \sum_{l_s=1}^{l_t} \frac{L_{l_s}\mathbb{1}\{c_{l_s} = c_j\}}{t}, \quad \mathring{x}_t = \arg\max_{x \in \mathcal{X}} x^\top \Theta \mathring{p}_t, \tag{O.2}$$

$$\mathring{\beta}_{t,j} := \min\left\{ \frac{\frac{1}{3} L_{\max}\ln\frac{2}{\delta_t} + \sqrt{\left(\frac{1}{3}L_{\max}\ln\frac{2}{\delta_t}\right)^2 + \frac{2\phi_{t,j}L_{\max}}{t}\ln\frac{2}{\delta_t}}}{t}, 1\right\},$$

$$\mathring{\phi}_{t,j} := \min\left\{4\max\left\{\mathring{p}_{t,j}, \frac{25}{4}\frac{L_{\max}}{t}\ln\frac{2}{\delta_t}\right\}, \frac{1}{4}\right\}, \quad \delta_t = \frac{\delta}{C_3 N t^3},$$

$$\mathring{\rho}_t(\ddot{x}_t, x) = \min_{\zeta(\mathring{x}_t, x) \in \mathbb{R}} \sum_{j=1}^{N} |\Delta_j(\mathring{x}_t, x) + \zeta(\mathring{x}, x)| \mathring{\beta}_{t,j}.$$

6: **while** $\min_{x \neq \mathring{x}_t} (\mathring{x}_t - x)^\top \Theta \mathring{p}_t - \mathring{\rho}_t < -\varepsilon$ **do**
7:     Observe $\theta_{j_t}^*$ and $\mathbb{1}\{t \in \mathcal{C}\}$ and let $t = t + 1$.
8:     Update $\mathring{x}_t$ and $\mathring{\rho}_t$ with (O.2).
9: **end while**
10: Recommend arm $\mathring{x}_\varepsilon = \mathring{x}_t$.

---

D$\varepsilon$BAI$_\beta$ utilizes the techniques we have used to bound the deviation between the true distribution $p_j$ and the estimated ones $\mathring{p}_{t,j}$ for all $j \in [N]$. Similarly, we let $\zeta(\mathring{x}, x) = \arg\min_{\zeta(\mathring{x},x)\in\mathbb{R}} \sum_{j=1}^{N} \mathring{\beta}_{t,j}(\Delta_j(\mathring{x}_t, x) + \zeta(\mathring{x}, x))^2 = -\frac{\sum_{j=1}^{N} \mathring{\beta}_{t,j}\Delta_j(\mathring{x}_t, x)}{\sum_{j=1}^{N} \mathring{\beta}_{t,j}}$ for efficient computing.

As the theoretical guarantee of D$\varepsilon$BAI$_\beta$ can be derived following a similar manner as the proof of the DE term of PS$\varepsilon$BAI in Appendix I.2 and in the proof of Lemma G.6, for simplicity, we just present the result and omit the proof here here.

**Theorem O.1.** D$\varepsilon$BAI$_\beta$ *identifies an $\varepsilon$-best arm within*

$$\tilde{O}\left(\max_{x_\varepsilon \in \mathcal{X}, x \neq x_\varepsilon, x^*} \frac{L_{\max}\mathring{H}^2(x_\varepsilon, x)}{(\Delta_{\min} + \varepsilon)^2}\ln\frac{1}{\delta}\right),$$

*time steps with probability at least $1 - \delta$ and in expectation, where*

$$\mathring{H}(x_\varepsilon, x) := \min_{\zeta(\mathring{x}_\varepsilon, x) \in \mathbb{R}} \sum_{j=1}^{N} \sqrt{\min\{16p_j, \frac{1}{4}\}} |\Delta_j(\mathring{x}_\varepsilon, x) + \zeta(\mathring{x}_\varepsilon, x)|.$$

We present a theorem, along with a proof sketch, for the theoretical guarantee of D$\varepsilon$BAI below.

**Theorem O.2.** D$\varepsilon$BAI *identifies an $\varepsilon$-best arm within*

$$\tilde{O}\left(\max_{x_\varepsilon \in \mathcal{X}, x \neq x_\varepsilon, x^*} \frac{L_{\max}\ddot{H}^2(x_\varepsilon, x)}{(\Delta_{\min} + \varepsilon)^2}\ln\frac{1}{\delta}\right),$$

*time steps with probability at least $1 - \delta$ and in expectation, where* $\ddot{H}(x_\varepsilon, x) := \min_{\zeta(\ddot{x}_\varepsilon, x) \in \mathbb{R}} \sum_{j=1}^{N} |\Delta_j(\ddot{x}_\varepsilon, x) + \zeta(\ddot{x}_\varepsilon, x)|.$

*Proof.* For the sake of conciseness and simplicity, only a proof sketch is provided for this Theorem. Similar results can be found in the referred contents. The proof is composed by 4 steps.

- Step 1: bound the deviation of $P_\theta[\theta_j^*]$ by Lemma O.3 and Remark O.4.
- Step 2: prove the recommended arm is an $\varepsilon$-best arm. Conditional on Step 1, we can prove this following the procedures in Lemma J.1.
- Step 3: give a sufficient condition and then obtain a high probability upper bound. This can be seen from (J.1) and by solving

$$\ddot{\rho}_t(\ddot{x}_t, x) = \ddot{\beta}_t \sum_{j=1}^{N} |\Delta_j(\ddot{x}_t, x) + \zeta(\ddot{x}_t, x)| \leq \frac{\Delta_{\min} + \varepsilon}{2}$$

$$\Rightarrow \quad l_t = \tilde{O}\left( \frac{\ddot{H}^2(x_\varepsilon, x)}{(\Delta_{\min} + \varepsilon)^2} \ln \frac{1}{\delta} \right).$$

$$\Rightarrow \quad t = \tilde{O}\left( \frac{L_{\max} \ddot{H}^2(x_\varepsilon, x)}{(\Delta_{\min} + \varepsilon)^2} \ln \frac{1}{\delta} \right).$$

The high probability result is obtained by maximizing the above over $x_\varepsilon \in \mathcal{X}, x \neq x_\varepsilon, x^*$.
- Step 4: the expected result can be derived in a similar method as in the proof of N$\varepsilon$BAI in Appendix F. The expected sample complexity and the high-probability sample complexity is of the same order.

$\square$

**Lemma O.3.** *With probability at least* $1 - \delta$, $\sup_{j \in [N]} |p_j - \ddot{p}_{t,j}| \leq \ddot{\beta}_t$ *for all* $t \in \mathbb{N}$, *where* $\ddot{\beta}_t = \sqrt{\frac{2}{l_t} \ln \frac{2}{\delta_{l_t}}}$ *and* $\delta_{l_t} = \frac{\delta}{C_3 l_t^3}$.

*Proof.* We may define the cumulative distribution function (CDF) and the empirical CDF for $P_\theta$ as $F_\theta(j) = \sum_{k=1}^{j} P_\theta[\theta_j^*]$ and $\ddot{F}_t(j) = \sum_{l_s=1}^{l_t} \mathbb{1}\{c_{l_s} = c_j, l_s \leq j\}$ respectively for $j \in [N]$. According to the DKW inequality, we have

$$\mathbb{P}\left[ \sup_{j \in [N]} |\ddot{F}_t(j) - F(j)| \geq \epsilon \right] \leq 2 \exp\left(-2l_t \epsilon^2\right).$$

By the implication

$$\epsilon \leq |p_j - \ddot{p}_{t,j}| = |F(j) - F(j-1) - \ddot{F}_t(j) + \ddot{F}_t(j-1)|$$
$$\leq |F(j) - \ddot{F}_t(j)| + |F(j-1) - \ddot{F}_t(j-1)|$$
$$\Rightarrow \quad |F(j) - \ddot{F}_t(j)| \geq \frac{\epsilon}{2} \quad \text{or} \quad |F(j-1) - \ddot{F}_t(j-1)| \geq \frac{\epsilon}{2},$$

we have that

$$\mathbb{P}\left[ \sup_{j \in [N]} |p_j - \ddot{p}_{t,j}| \geq \epsilon \right] \leq \mathbb{P}\left[ \sup_{j \in [N]} |\ddot{F}_t(j) - F(j)| \geq \frac{\epsilon}{2} \right] \leq 2 \exp\left(-2l_t \left(\frac{\epsilon}{2}\right)^2\right)$$
$$= 2 \exp\left(-\frac{l_t \epsilon^2}{2}\right).$$

This is equivalent to

$$\mathbb{P}\left[ \sup_{j \in [N]} |p_j - \ddot{p}_{t,j}| \geq \ddot{\beta}_t \right] \leq \delta_{l_t}.$$

A union bound gives an upper bound of the failure probability $\sum_{l_t=1}^{\infty} \delta_{l_t} = \delta$. $\square$

**Remark O.4.** *The use of the DKW inequality gives a union bound over the deviation of the distribution estimation all contexts. This avoid the $N$ factor in the logarithm in $\ddot{\beta}_t$ and is beneficial when the number of contexts $N$ is large.*

*When there are a few contexts, it is also possible to directly bound the deviation for each context via Azuma-Hoeffding's inequality, i.e.,*

$$\mathbb{P}\left[|p_j - \ddot{p}_{t,j}| \leq \ddot{\beta}_t\right] \leq \delta_{l_t}$$

*where $\ddot{\beta}_t := \sqrt{\frac{1}{2l_t} \ln \frac{2C_3 N l_t^3}{\delta}}$. A union bound gives that with probability at least $1-\delta$, $|p_j - \ddot{p}_{t,j}| \leq \ddot{\beta}_t$ for all $t \in \mathbb{N}, j \in [N]$. As this new confidence radius is the same as the one in Lemma O.3 up to constant and logarithmic terms, the sample complexity should be of the same order.*

*We adopt this confidence radius $\ddot{\beta}_t := \sqrt{\frac{1}{2l_t} \ln \frac{2C_3 N l_t^3}{\delta}}$ in the experiment.*

## P   Additional Discussions

In this section, we provide additional discussions on

- the related methods for BAI in the nonstationary bandits literature in subsection P.1,
- the upper on the sample complexity in Theorem 3.2 and Theorem 3.3 in subsection P.2,
- the connection between the piecewise-stationary linear bandits model to the stationary linear bandits model in subsection P.3,
- the special case where the number of context $N = 1$ in subsection P.4,

### P.1   Discussion on the Related Methods in Nonstationary Bandits

To the best of our knowledge, there is limited literature investigating the BAI in the nonstationary bandits setup (there is comparatively a much richer literature for regret minimization in non-stationary environments): [14] studies BAI in the *fixed-horizon* setup and [15] assumes the *best arm remains unchanged* after certain time step and explores the fixed-confidence setting. Both of the works are not directly comparable to our proposed piecewise-stationary setup.

Additionally, we provide strong baselines algorithms $D\varepsilon BAI$ and $D\varepsilon BAI_\beta$ in Section 6, which are detailed in Section O.4. These two baselines are given oracle information about the context (see Dynamics 3), while PS$\varepsilon$BAI$^+$ is not. As indicated by Figure 2(b), PS$\varepsilon$BAI$^+$ is competitive compared to these strong baselines and is much better than the naive approach.

### P.2   More Discussions on the Upper Bound

Firstly, we emphasize that the sample complexity in Theorem 3.2 and Theorem 3.3 are instance-dependent, in particular, the term $H_{DE}$ characterizes the difficulty in estimating the distribution of the context. Furthermore, the presence of the gaps $\Delta(x, x^*)$ also underscores that our upper bounds are functions of the instance.

Secondly, our algorithm adopts the G-optimal design, which is minimax optimal for the BAI in standard linear bandits problem [1, 2]. Although we have not proved it, we believe that the sample complexity of the proposed algorithm may not be instance-dependent optimal.

Lastly, we provide some remarks on the instance-dependent optimal algorithms:
(1) The G-optimal design is the cornerstone for the more efficient and adaptive rules like $\mathcal{XY}$-allocation and $\mathcal{XY}$-Adaptive Allocation in [1]. Therefore, our algorithm provides a framework for more sophisticated algorithms in the piecewise-stationary linear bandits problem with the BAI task. Empirically, one can attempt to replace the G-optimal design by the $\mathcal{XY}$-allocation during implementation, which can possibly yield empirical benefits.
(2) The current lower bound is established with two simpler problems (see Section 4), which is sufficient to show our algorithm is minimax optimal. However, a tighter instance-dependent lower bound based on the original problem may be required if one wishes to show an algorithm is instance-dependent optimal. This can be challenging because the distribution of the arms, the contexts and the changepoints are unknown, and the characterization of the "alternative instance" (i.e., the $\varepsilon$-best arm set is changed) is difficult.

To conclude, we believe an instance-dependent optimal algorithm is appealing and can lead to future research.

### P.3  Connections to the Stationary Bandits

**Firstly**, we would like to clarify that, in the piecewise-stationary linear bandits model, the instance tends towards a stationary one when $\max_{j\in[N]} p_j \to 1$, instead of the scenario in which $L_{\max} \to \infty$.

When $L_{\min}$ and $L_{\max}$ are large, estimating each latent vector becomes easier since there are more observations in a single stationary segment. However, the overall reward of an arm also depends on $P_\theta$, the distribution of latent context vectors, which is also assumed to be unknown in the setup. In order to estimate $P_\theta$, a learning agent can get one (unobservable) sample from $P_\theta$ only at the (unobservable) change points. In other words, $L_{\max}$ charaterizes the "sparsity" of the context samples: the larger $L_{\max}$ is, the sparser the context samples are. A larger $L_{\max}$ indicates that samples from $P_\theta$ are likely to be generated less frequently, which will result in the increase of the overall sample complexity. Consider an extreme example of Dynamics 3 where the context vector is revealed at every time step and $L_{\min}$ and $L_{\max}$ are very large. If at least $l_\tau$ context samples from $P_\theta$ are required to identify the best arm, then a sample complexity of $l_\tau L_{\min}$ is unavoidable in the our setup under this extreme case.

**Secondly**, while $\max_{j\in[N]} p_j$ is important in characterizing the (non)stationarity of the instance, we would like to emphasize that **both** the distribution $P_\theta$ and latent context vectors are essential in characterizing the sample complexity, as shown in the term $\bar{H}(x_\epsilon, x)$ in Theorem 3.2. To further illustrate this, we provide a simple but illuminating instance as follows: The instance is composed by $N = 3$ contexts and $K = 2$ arms, and $\Delta_j$ denotes the mean gap between arm 1 and arm 2 under context $j \in [N]$ (to be specified below) and $\Delta := \sum_{j\in[N]} p_j \Delta_j$ denotes the weighted mean gap between arm 1 and arm 2. The probability of context 1 is $p_1 = \max_{j\in[N]} p_j = 1 - p_2 - p_3$, and $p_1$ is close to 1. Let $\hat{x}$ denote the empirical version of the statistics $x$ in the following arguments.

In such a case, consider this question: *if an algorithm has only detected context 1 in the first $t$ time steps and $t$ is large, should it stop or not?*

As no change point has occurred, the current dynamics behaves similarly to that of a stationary bandit environment up till now. Thus the empirical probabilities of the 3 contexts are $\hat{p}_1 = 1, \hat{p}_2 = \hat{p}_3 = 0$ and the empirical mean gap under context 1 $\hat{\Delta}_1$ is close to $\Delta_1$. Consider the following two cases:

- Case 1: $\Delta_1 = p_2$ and $\Delta_2 = \Delta_3 = -p_1$. The true weighted mean gap between arm 1 and arm 2 is $\Delta = \sum_{j\in[N]} p_j \Delta_j = -p_1 p_3 < 0$, and thus arm 2 is the best arm. This indicates that, in order to estimate $\Delta_2$ and $\Delta_3$, an algorithm needs to observe contexts 2 and 3 despite their small probabilities. But currently $\hat{p}_2 = \hat{p}_3 = 0, \hat{\Delta} = \hat{\Delta}_1 \approx \Delta_1 > 0$, so the algorithm should not terminate.
- Case 2: $\Delta_1 = 1$ and $\Delta_2, \Delta_3 \in [-2, 2]$. Thus $\hat{\Delta} \approx \Delta \geq p_1 - 2p_2 - 2p_3 > 0$. This indicates there is no need to observe contexts 2 and 3, because arm 1 is the best even in the worst case ($\Delta_2 = \Delta_3 = -2$). In this case, as long as the algorithm gets a good estimate of $\Delta_1$, it can confidently terminate.

As these two cases indicate, in addition to $P_\theta$, the latent vectors (which determine the means of the arms) are equally important for the stopping time. Our algorithm takes both factors into account, as reflected by the summation term in equation (3.4) and $\hat{\Delta}_t(x_t^*, x)$ in (3.5) for the algorithm design, and by $\bar{H}(x_\epsilon, x)$ for the theoretical upper bound.

We **conclude** that

- The unknown distribution $P_\theta$ and the latent vectors **jointly** determine the sample complexity. $\bar{H}(x_\epsilon, x)$ in Theorem 3.2 characterizes their roles, where the contexts with larger probabilities have commensurately greater influence on the sample complexity of the algorithm.
- When $\max_{j\in[N]} p_j \to 1$ with other parameters fixed, the piecewise-stationary linear bandit instance reduces to a stationary one. $H_{DE}(x_\epsilon, x)$ in Theorem 3.2 becomes $L_{\max}/4$, which implies that a **constant** number of change points need to be observed, and thus $T_D(x_\epsilon, x)$ can be regarded as a constant. The dominant term in our upper bound, the $T_V(x)$ term, recovers the upper bound in the stationary bandits, that is, $\max_{x \neq x^*} \frac{d}{(\Delta(x^*, x) + \varepsilon)^2} \ln(1/\delta)$ [32].

## P.4   Special Case: $N = 1$

We only consider $N > 1$ as the bandit model is only **truely** piecewise-stationary when $N > 1$. Thus, we did not specifically derive upper and lower bounds for the extreme case where $N = 1$. Nevertheless, our analysis can cover this case due to the following reasons:

- The (minimax) lower bound in Theorem 4.1 is not directly applicable since it is not derived for this extreme instance. However, if we step back to the analysis equation (M.16), the feasible set in the optimization problem is empty, and thus the lower bound on $\mathbb{E}[l_t]$ ($N_C$ in Theorem4.1) is 0. In this case, the lower bound in Theorem4.1 reduces to the lower bound in the stationary bandits.
- Regarding the upper bound in Theorem 3.2, when $N = 1$, we have $p_1 = 1$ and $H_{DE} = \frac{L_{\max}}{4}$. In this case, as $H_{DE}$ does not depend on the mean gaps and the mean gaps are among the fundamental quantities to characterize the difficulty of an instance, the $T_D$ term can be regarded as a constant. Therefore, the dominant term in the upper bound is $T_V$. This recovers the bound in the stationary setup.
- In addition, as we assume $N$ is an input to our algorithm, when we are aware $N = 1$, we can actually adopt the algorithms for the stationary setting, e.g., the $G$-allocation or $\mathcal{XY}$-allocation rule [1].

## Q   $\varepsilon$-Best Arm Tuple Identification Problem

In the main paper, we consider the identification of an $\varepsilon$-best arm in terms of the "ensemble" quality $\mu_x := \mathbb{E}_{\theta \sim P_\theta}[x^\top \theta] = \sum_{j=1}^N P_\theta[\theta_j^*] x^\top \theta_j^*$. The curious reader may wonder whether we can identify the $\varepsilon$-best arm tuple $\mathcal{X}_\varepsilon^{\text{tuple}} := (x_1^\varepsilon, \ldots, x_N^\varepsilon)$, where $x_j^\varepsilon$ is an $\varepsilon$-best arm under context $j$, i.e., $x_j^\varepsilon \in \{x : x^\top \theta_j^* \geq \max_{x \in \mathcal{X}} x^\top \theta_j^* - \varepsilon\}$. Given the tools we developed in this manuscript, we answer this problem in the affirmative.

**Intuitions:** Let $x_j^* := \arg\max_{x \in \mathcal{X}} x^\top \theta_j^*$ and $\Delta_j^* := \min_{x \neq x_j^*} \Delta_j(x_j^*, x)$. We expect to have an upper bound taking the form of

$$\tilde{O}\left( \max_{j \in [N]} \frac{L_{\max}}{p_j} \cdot \frac{d}{\left(\Delta_j^* + \varepsilon\right)^2 L_{\min}} \ln \frac{1}{\delta} \right)$$

where $O(\frac{d}{(\Delta_j^* + \varepsilon)^2 L_{\min}} \ln \frac{1}{\delta})$ is an upper bound on the number of context samples $j$ and context $j$ will occur once among every $\frac{1}{p_j}$ contexts in expectation.

**Analysis of the problem:** The change detection and context alignment subroutines are only effective within $\tau^*$ time steps (Line 2 in Algorithm 1). However, if a context is with small occurrence probability $p_j$, it may not appear before $\tau^*$ time steps.

Regarding this scenario, we only expect to obtain a high-probability upper bound on the sample complexity, whereas the expected sample complexity requires more techniques beyond our parallel execution trick (which is used to design PS$\varepsilon$BAI$^+$) and is an interesting direction left for future research.

Besides, it is more feasible to consider the identification of $\varepsilon$-best arms under contexts with high occurrence probability, i.e., we aim to identify

$$\mathcal{X}_{\varepsilon, \bar{p}}^{\text{tuple}} := \{x_j^\varepsilon : p_j > \bar{p}\}$$

where $\bar{p}$ is a threshold on the occurrence probability of contexts. Let $\Theta_{\bar{p}} = \{\theta_j^* : p_j > \bar{p}\}$ denote those context with high occurrence probability.

**Goal:** We aim to devise an algorithm with the minimum sample complexity (arm pulls) to ascertain either (1) an $\varepsilon$-best arm under context $j$, or (2) $\theta_j^* \notin \Theta_{\bar{p}}$.

With the above goal, we expect to identify an $\varepsilon$-best arm for all $j$ with $\theta_j^* \in \Theta_{\bar{p}}$; for those $j$ with $\theta_j^* \notin \Theta_{\bar{p}}$, either an $\varepsilon$-best arm is identified or $\theta_j^* \notin \Theta_{\bar{p}}$ is ascertained.

We propose Algorithm 10 for this problem, which is quite similar to Algorithm 1, except for a few changes:

**Algorithm 10** PIECEWISE-STATIONARY $\varepsilon$-BEST ARM TUPLE IDENTIFICATION

---

1: **Input:** arm set $\mathcal{X}$, size of the set of latent vectors $N$, bounds on the segment lengths $L_{\min}$ and $L_{\max}$, slackness parameter $\varepsilon$, confidence parameter $\delta$, sampling parameter $\gamma$ and window size $w$, threshold $b$, probability threshold $\bar{p}$.

2: **Initialize**: Compute the G-optimal allocation $\lambda^*$ and $\tau^* = \frac{38400 \ln(80) N L_{\max}}{\varepsilon^2} \ln \frac{N^2 K L_{\max}}{\delta \varepsilon^2}$ and Flag $= [N]$ and Hold $= [\,]$ and Output $= [\,]$.

3: Set $\mathrm{CD}_{\mathrm{sample}} = [\,]$, $\mathrm{CA}_{\mathrm{id}} = \{\,\}$. Set $t_{\mathrm{CD}} = +\infty$.

4: Set $\mathcal{T}_{t,j} = \emptyset$ and initialize $\mathcal{T}_t, T_{t,j}, T_t$ with (3.1) for all $t \leq \tau^*$, $j \in [N]$.

5: Sample $\frac{w}{2}$ arms $\{x_s\}_{s=1}^{\frac{w}{2}} \sim \lambda^*$ and observe the associated returns $\{Y_{s,x_s}\}_{s=1}^{\frac{w}{2}}$, $t = \frac{w}{2}$, $t_{\mathrm{CA}} = \frac{w}{2}$.

6: $\mathrm{CA}_{\mathrm{id}} = \{1 : [(x_s, Y_{s,x_s})]_{s=1}^{\frac{w}{2}}\}$, $\hat{j}_t = 1$.

7: **while** $t \leq \tau^*$ and Flag $\neq \emptyset$ **do**

8:     $t = t + 1$

9:     Sample an arm $x_t \sim \lambda^*$ and observe return $Y_{t,x_t}$.

10:     **if** $\mathrm{mod}\,(t - t_{\mathrm{CA}}, \gamma) \neq 0$ **then**

11:         Update $\hat{j}_t = \hat{j}_{t-1}$, $\mathcal{T}_{t,\hat{j}_t} = \mathcal{T}_{t-1,\hat{j}_t} \cup \{t\}$, $\mathcal{T}_{t,j} = \mathcal{T}_{t-1,j}$ for $j \neq \hat{j}_t$.

12:     **else**

13:         $\mathrm{CD}_{\mathrm{sample}} = \mathrm{CD}_{\mathrm{sample}} + [(x_t, Y_{t,x_t})]$.

14:         Update $\hat{j}_t = \hat{j}_{t-1}$, $\mathcal{T}_{t,j} = \mathcal{T}_{t-1,j}$ for all $j \in [N]$.

15:         **if** $|\mathrm{CD}_{\mathrm{sample}}| \geq w$ **then**

16:           **if** $\mathrm{LCD}(\mathcal{X}, w, b, \mathrm{CD}_{\mathrm{sample}}[-w:])$ **then**

17:             $\mathrm{CD}_{\mathrm{sample}} = [\,]$.

18:             $t = t + \frac{w}{2}$, $t_{\mathrm{CA}} = t$, $t_{\mathrm{CD}} = +\infty$.

19:             $\hat{j}_t, \mathrm{CA}_{\mathrm{id}} = \mathrm{LCA}(\mathcal{X}, w, b, \mathrm{CA}_{\mathrm{id}})$.

20:             **if** $\hat{j}_t = N + 1$ **then break**.

21:             Revert $\mathcal{T}_{t,j} = \mathcal{T}_{t - \frac{w(\gamma+1)}{2}, j}$ for all $j \in [N]$.

22:           **end if**

23:         **end if**

24:     **end if**

25:     Update the estimates with (3.1), (3.2) and (Q.1).

26:     **if** $\exists j \in$ Flag condition (Q.2) is met for $j$ and $t_{\mathrm{CD}} = +\infty$ **then**

27:         **for** $j \in$ Flag **do**

28:           **if** $\min_{x:x \neq x_{t,j}^*} \hat{\Delta}_{t,j}(x_{t,j}^*, x) - \alpha_{t,j} \geq -\varepsilon$ **then**

29:             Record $j$ and $\hat{x}_{j,\varepsilon} := \arg\max_{x \in \mathcal{X}} x^\top \hat{\theta}_{t,j}$ in Hold.

30:             Flag $=$ Flag $\setminus \{j\}$ and $t_{\mathrm{CD}} = |\mathrm{CD}_{\mathrm{sample}}|$.

31:           **else if** $\hat{p}_j + \beta_{t,j} < \bar{p}$ **then**

32:             Record $j$ and $\hat{x}_{j,\varepsilon} = \mathrm{NAN}$ in Hold.

33:             Flag $=$ Flag $\setminus \{j\}$ and $t_{\mathrm{CD}} = |\mathrm{CD}_{\mathrm{sample}}|$.

34:           **end if**

35:         **end for**

36:     **else if** $t_{\mathrm{CD}} = |\mathrm{CD}_{\mathrm{sample}}| - \frac{w}{2}$ **then**

37:         **for** $(j, \hat{x}_{j,\varepsilon})$ in Hold **do**

38:           Output$[j] = \hat{x}_{j,\varepsilon}$

39:         **end for**

40:     **end if**

41: **end while**

42: Recommend Output

---

- On Line 7, the algorithm stops when an $\varepsilon$-best arm or $p_j \leq \bar{p}$ is identified for all contexts.
- On Line 25, it computes the confidence radii for the arms in context $j_t$ as well as the confidence radii for the occurrence probabilities

$$\alpha_{t,j} = \frac{d}{n} \ln \frac{2}{\delta_{v,T_t}} + \sqrt{\left(\frac{d}{n} \ln \frac{2}{\delta_{v,T_t}}\right)^2 + \frac{4d}{n} \ln \frac{2}{\delta_{v,T_t}}}, \tag{Q.1}$$

$$\beta_{t,j} := \min\left\{\frac{5}{2}\sqrt{\frac{2\phi_{t,j}L_{\max}}{T_t}\ln\frac{2}{\delta_{d,T_t}}}, 1\right\},$$

$$\text{where } \phi_{t,j} := \min\left\{4\max\left\{\hat{p}_{t,j}, \frac{25}{4}\frac{L_{\max}}{T_t}\ln\frac{2}{\delta_{d,T_t}}\right\}, \frac{1}{4}\right\},$$

- On Line 26 to 40, the stopping rule is changed:
  - Stopping condition **(I)** on Line 26

$$\left(\min_{x:x\neq x_{t,j}^*}\hat{\Delta}_{t,j}(x_{t,j}^*, x) - \alpha_{t,j} \geq -\varepsilon \quad \textbf{or} \quad \hat{p}_j + \beta_{t,j} < \bar{p}\right) \tag{Q.2}$$
$$\text{and} \quad T_t \geq (2L_{\max}/9)\ln(2/\delta_{d,T_t})$$

  where $x_{t,j}^* := \arg\max_{x\in\mathcal{X}} x^\top\hat{\theta}_{t,j}$.
  - Lines 27 to 35: identify an $\varepsilon$-best arm or ascertain $p_j \leq \bar{p}$ among the remaining contexts, and record these observations in Hold for easy access in stopping condition **(II)**.
  - Lines 36 to 40: the recommended $\varepsilon$-best arm is recorded, where we adopt NAN to flag those contexts with small occurrence probabilities.

**Theorem Q.1.** *Given an instance $\Lambda$, with probability at least $1-\delta$, Algorithm 10 can recommend an $\varepsilon$-best arm for all context $j \in \Theta_{\bar{p}}$ with sample complexity*

$$\max\left\{\tilde{O}\left(\max_{\theta_j^*\in\Theta_{\bar{p}}}\max\left\{L_{\max}, \frac{d}{\left(\Delta_j^* + \varepsilon\right)^2}\right\} \cdot \frac{\ln(1/\delta)}{p_j}\right), \tilde{O}\left(\max_{\theta_j^*\notin\Theta_{\bar{p}/2}}\frac{\min\left\{p_j, \frac{1}{64}\right\}L_{\max}\ln(1/\delta)}{(p_j - \bar{p})^2}\right),\right.$$

$$\left.\tilde{O}\left(\max_{\theta_j^*\in\Theta_{\bar{p}/2}\setminus\Theta_{\bar{p}}}\min\left\{\max\left\{L_{\max}, \frac{d}{\left(\Delta_j^* + \varepsilon\right)^2}\right\}\frac{\ln(1/\delta)}{p_j}, \frac{\min\left\{p_j, \frac{1}{64}\right\}L_{\max}\ln(1/\delta)}{(p_j - \bar{p})^2}\right\}\right)\right\},$$

*which can be simplified as*

$$\tilde{O}\left(\max\left\{L_{\max}, \frac{d}{\varepsilon^2}\right\} \cdot \frac{\ln(1/\delta)}{\bar{p}}\right).$$

*Proof of Theorem Q.1.* We provide a concise proof sketch for this theorem.

By adapting the stopping rule for best arm identification in stationary linear bandits to $\varepsilon$-best arm identification, we observe that the number of arm pulls needed for $\varepsilon$-best arm identification under context $j$ is

$$T_{t,j} = \tilde{O}\left(\frac{d}{\left(\Delta_j^* + \varepsilon\right)^2}\ln\frac{1}{\delta}\right).$$

The remaining problem is to determine how many context samples/changepoints are needed such that the above number of arm samples can be achieved.

Recall that $\hat{p}_{t,j} = \frac{T_{t,j}}{T_t}$ (3.2) and Lemma I.2, we have

$$\mathbb{P}\left[|T_{t,j} - T_t p_j| \geq T_t\beta_{t,j}\big|\text{Good}\right] \leq \delta_{d,T_t}.$$

In addition, we have an upper bound on $\beta_{t,j}$ as in (J.5), so there would be sufficient arm samples for $\varepsilon$-best arm identification under context $j$ if

$$T_t \cdot \left(p_j - 2\cdot\frac{5}{2}\sqrt{\frac{2\min\left\{16p_j, \frac{1}{4}\right\}L_{\max}}{T_t}\ln\frac{2}{\delta_{d,T_t}}}\right) \geq \tilde{O}\left(\frac{d}{\left(\Delta_j^* + \varepsilon\right)^2}\ln\frac{1}{\delta}\right)$$

By solving this inequality in terms of $T_t$, we obtain

$$T_t = \tilde{O}\left(\max\left\{L_{\max}, \frac{d}{\left(\Delta_j^* + \varepsilon\right)^2}\right\} \cdot \frac{\ln(1/\delta)}{p_j}\right).$$

As we aim to identify the $\varepsilon$-best arms under each context $\theta_j^* \in \Theta_{\bar{p}}$, the upper bound is at least

$$T_t = \tilde{O}\left(\max_{\theta_j^* \in \Theta_{\bar{p}}} \max\left\{L_{\max}, \frac{d}{(\Delta_j^* + \varepsilon)^2}\right\} \cdot \frac{\ln(1/\delta)}{p_j}\right) \tag{Q.3}$$

even if the set $\Theta_{\bar{p}}$ is given.

Similarly, for those contexts with small occurrence probability $\theta_j^* \notin \Theta_{\bar{p}}$, by Lemma I.2 and (J.5), we can identify them when

$$p_j + 2 \cdot \frac{5}{2}\sqrt{\frac{2\min\left\{16p_j, \frac{1}{4}\right\}L_{\max}}{T_t}\ln\frac{2}{\delta_{d,T_t}}} \leq \bar{p}$$

$$\Rightarrow \quad T_t = \tilde{O}\left(\frac{\min\left\{p_j, \frac{1}{64}\right\}L_{\max}\ln(1/\delta)}{(p_j - \bar{p})^2}\right).$$

Careful readers may notice that the denominator depends on a "probability gap", which can be very small. In practice, if we can actually identify the $\varepsilon$-best arm in those contexts, it is also acceptable. Therefore, we instead only choose not to identify the $\varepsilon$-best arm in those contexts $\theta_j^* \in \Theta_{\bar{p}/2}$, which yields an upper bound

$$T_t = \tilde{O}\left(\max_{\theta_j^* \notin \Theta_{\bar{p}/2}} \frac{\min\left\{p_j, \frac{1}{64}\right\}L_{\max}\ln(1/\delta)}{(p_j - \bar{p})^2}\right). \tag{Q.4}$$

In this case, the bound is at most $\tilde{O}\left(\frac{L_{\max}\ln(1/\delta)}{\bar{p}}\right)$. For the rest contexts $\theta_j^* \notin \Theta_{\bar{p}/2} \setminus \Theta_{\bar{p}}$, either identifying the $\varepsilon$-best arm or ascertaining its small occurrence probability suffices, that is,

$$T_t = \tilde{O}\left(\max_{\theta_j^* \in \Theta_{\bar{p}/2}\setminus\Theta_{\bar{p}}} \min\left\{\max\left\{L_{\max}, \frac{d}{(\Delta_j^* + \varepsilon)^2}\right\}\frac{\ln(1/\delta)}{p_j}, \frac{\min\left\{p_j, \frac{1}{64}\right\}L_{\max}\ln(1/\delta)}{(p_j - \bar{p})^2}\right\}\right). \tag{Q.5}$$

The above bound is upper bounded by $\tilde{O}\left(\max\left\{L_{\max}, \frac{d}{\varepsilon^2}\right\} \cdot \frac{\ln(1/\delta)}{\bar{p}}\right)$.

By taking the maximum of (Q.3), (Q.4) and (Q.5), we can obtain a high-probability problem-dependent upper bound on the sample complexity. In addition, we can get a high-probability problem-independent upper bound

$$\tilde{O}\left(\max\left\{L_{\max}, \frac{d}{\varepsilon^2}\right\} \cdot \frac{\ln(1/\delta)}{\bar{p}}\right).$$

By setting the threshold $\bar{p}$ carefully, the algorithm is guaranteed to terminate before $\tau^*$ given the good event Good (G.1). $\qquad\square$

