# OpenReview forum: "Almost Minimax Optimal Best Arm Identification in Piecewise Stationary Linear Bandits"
_NeurIPS.cc/2024/Conference — NeurIPS 2024 poster_

### Official Review · Reviewer_tHAq · 2024-07-10

**Soundness:** 3
**Presentation:** 3
**Contribution:** 3
**Rating:** 6
**Confidence:** 3

**Summary:**

he paper introduces a novel algorithm, PSεBAI+, designed for ε-Best Arm Identification in Piecewise Stationary Linear Bandits (PSLB). The algorithm addresses the challenge of identifying an arm with an average return close to the optimal under varying contexts and unknown changepoints. It incorporates change detection and context alignment mechanisms to adapt to the non-stationary nature of the environment efficiently. The paper provides theoretical guarantees for the algorithm's performance and supports these claims with empirical results.

**Strengths:**

1. The paper proposes a new algorithm that combines change detection and context alignment in a PSLB setting. The algorithm is optimized to identify ε-optimal arms with a minimal number of samples, which is crucial for practical applications.
2. It offers a thorough theoretical analysis, including a proof of the algorithm's high-probability performance and sample complexity bounds.
3.The paper backs up the theoretical findings with empirical experiments, demonstrating the algorithm's efficiency and robustness.

**Weaknesses:**

1. The paper does not provide a comparison with the current state-of-the-art methods, which is essential for establishing the novelty and superiority of the proposed approach.

2. Parameter Sensitivity Analysis:
- There is no ablation study on the parameters w and b, which, based on past experiences replicating reference [10], could have a significant impact on the algorithm's performance.

**Questions:**

1. About the context alignment:
- When a new distribution is detected, how does the algorithm decide when to perform context alignment?
- What methods are used to judge the similarity between the new context and existing ones? The concept of context alignment is common, but the paper lacks a detailed description of the design specifics.
- How does the algorithm confirm the reliability of the estimated new context feature vectors?

2. Could the authors elaborate on the specific application of the G-optimal design in the algorithm?

**Limitations:**

Yes.

---

> ### Author Rebuttal · Authors · 2024-08-05
>
> >Comparison with the current state-of-the-art methods.
>
> We thank the reviewer for the question. As we stated in Line 101 in the manuscript, to the best of our knowledge, there is limited literature investigating the BAI problem in the nonstationary bandit setup (there is comparatively a much richer literature for regret minimization in non-stationary environments): [14] studies BAI in the **fixed-horizon** setup and [15] assumes the **best arm remains unchanged** after certain time step and explores the fixed-confidence setting. Both of the works are not directly comparable to our proposed piecewise-stationary setup.
>
> Additionally, we provide strong baseline algorithms $D\varepsilon BAI$ and $D\varepsilon BAI_\beta$ in Section 6, which are detailed in Section O.3 on page 58. These two baselines are given oracle information about the context (see Dynamics 3 on page 49), while $PS\varepsilon BAI^+$ is not. As indicated by Figure 2(b), $PS\varepsilon BAI^+$ is competitive compared to these strong baselines and is much better than the naive approach.
>
> >Ablation study on the parameters w and b
>
> We thank the reviewer for the suggestion and we design an experiment to study the choice of $w$ and $b$.
>
> According to the distinguishability condition (Assumption 1 on page 6), point (2) indicates we can set $w=\frac{L_{\min}}{3\gamma}$, where $\gamma$ is the change detection frequency, and point (3) indicates $w$ and $b$ are coupled. In the original experiment, we have fixed $\gamma=6$ (see Line 327) so $w = \tilde{w}:=\frac{L_{\min}}{18}$.
>
> In the ablation study on $w$ and $b$, we choose to vary the choice of $\gamma$, and thus, $w$ and $b$ will change accordingly. Specifically, we select $\gamma\in\\{2,3,6,12\\}$ and the corresponding $w\in\\{3\tilde{w},2\tilde{w},\tilde{w},\frac{\tilde{w}}{2}\\}$. The other parameters remain unchanged. The experiment result is presented in the attached file in the global response. When $\gamma=18,w = \frac{\tilde{w}}{3}$, Assumption 1 is severely violated and the result is not desirable.
>
> The result indicates that, while smaller $\gamma$ and greater $w$ can result in slightly greater sample complexity, the overall sample complexity does not vary much and the superiority of our algorithm over the naive uniform sampling algorithm $N\varepsilon BAI$ is maintained. We conclude that our algorithm is robust against these choices, as long as Assumption 1 is not severely violated.
>
> We thank the reviewer for the valuable suggestion again and we will incorporate the above discussions in the revised manuscript.
>
>
>
> >About the context alignment:
> (1) When to perform context alignment?
> (2) What methods are used to judge the similarity between the new context and existing ones? Detailed description of the context alignment design.
> (3) How does the algorithm confirm the reliability of the estimated new context feature vectors?
>
> (1) We believe the reviewer is asking "When a new **context** is detected, how does the algorithm decide when to perform context alignment?"
>
> When a new context is detected, i.e., Line 16 of Algorithm 1 is True (Step 1 in Figure 1(b)), the algorithm immediately enters the Context Alignment phase in Lines 17 to 22 (Step 2 in Figure 1(b)).
>
> (2) The methods used to judge the similarity between the new context and the existing ones are encompassed in the $LCA$ subroutine. Due to the page limit, we postpone the detailed description of $LCA$ to page 15 with vivid explanations in Lines 565 to 576. We politely ask the reviewer to kindly refer to the description therein.
>
> (3) The estimated index $\hat{j}\_t$ of the new context is given by the $LCA$ subroutine, whose performance is guaranteed by Lemma G.3 on page 23. With the estimated context index, the new samples (up to the next changepoint) are collected by the corresponding time indices collector $\mathcal{T}\_{t,\hat{j}\_t}$ in Line 11 of Algorithm 1, which are later used to estimate the context feature vectors in Line 25. We adopt the concentration Lemma E.1 in the proof of Lemma I.1 on page 30, which quantifies how well the estimated context vectors concentrate around the groundtruth.
>
> >Could the authors elaborate on the specific application of the G-optimal design in the algorithm?
>
> We thank the reviewer for the question. As there can be two interpretations of the proposed question, we answer both of them to avoid any misunderstandings. One interpretation concerns the exact **implementation** of the G-optimal design; the other concerns the **theoretical justification** for why we use the G-optimal design within $PS\varepsilon BAI$.
>
> In terms of the **implementation** of the G-optimal design, as stated in Line 1288, we adopt the Wolfe--Atwood Algorithm with the Kumar--Yildirim start introduced in [33]. We kindly ask the reviewer to kindly refer to the description therein. In Algorithm 1, the G-optimal allocation is computed in Line 2 and is stored in $\lambda^*$. A new arm will be sampled according to $\lambda^*$ in Line 9.
>
> Perhaps the reviewer is wondering why we use the G-optimal design at all in our algorithm. From a **theoretical justification**, we make use of properties of the G-optimal design (i.e., Equation (E.3) on page 18) in the proof the concentration lemmas (Lemma E.1 on page 17 and Lemma H.1 on page 25), which have been further used in the proof of the upper bound (e.g., Lines 851 and 880) and in the guarantees of the $LCD$ and $LCA$ subroutines (e.g. Lemma G.1).
>
> We note that the G-optimal design has been widely used in the linear bandits literature [1,2,14]. The G-optimal design takes the geometry of the arm set into account and reduces the uncertainty of the estimated latent context vector in all directions uniformly, thus enhancing overall efficiency/sample complexity. Therefore, we are inspired to incorporate the G-optimal design into our algorithm design and the proposed algorithm achieves the minimax optimality, as expected (but requires work to prove).

---

> > ### Comment · Reviewer_tHAq · 2024-08-12
> >
> > Thanks for the authors' detailed feedback which answers my questions! I have no further questions.

---

### Official Review · Reviewer_comL · 2024-07-16

**Soundness:** 3
**Presentation:** 3
**Contribution:** 3
**Rating:** 6
**Confidence:** 4

**Summary:**

The paper introduces a piecewise stationary linear bandit (PSLB) model and presents the PSεBAI+ algorithm for identifying the best arm with minimal sample complexity. The PSLB model accounts for environments where contexts change at unknown points and are drawn from an unknown distribution. The PSεBAI+ algorithm, combining PSεBAI and NεBAI subroutines, aims to detect changepoints and align contexts. Theoretical analysis indicates that PSεBAI+ achieves an expected sample complexity close to the optimal. The paper also includes numerical experiments comparing PSεBAI+ with baseline algorithms.

**Strengths:**

- Introduces a new framework for best arm identification in piecewise stationary environments.
- The proposed PSεBAI+ algorithm combines changepoint detection and context alignment.
- Provides proofs that the expected sample complexity of PSεBAI+ is nearly optimal.
- Includes numerical experiments showing the algorithm's performance relative to baseline methods.

**Weaknesses:**

- The justification for using a piecewise stationary model could be more robust, as real-world scenarios may involve more complex conditions.
- The discussion on computational complexity is limited. The authors should discuss the computation required for each major step of the proposed algorithm.

**Questions:**

- Is $\theta_j^*$ sampled IID from $P_{\theta}$ each time? If so, it should be clearly stated "IID"

**Limitations:**

Discussion on limitations can be improved, with designated section.

---

> ### Author Rebuttal · Authors · 2024-08-05
>
> >The justification for using a piecewise stationary model could be more robust, as real-world scenarios may involve more complex conditions.
>
> We agree that the real-world scenarios may involve more complex conditions. However, even under our relatively benign setup, the proposed solution is already sufficiently complicated. Given the limited literature on Best Arm Identification under the nonstationarity (the only ones we know are [14] and [15]), we believe our work would be of great interest to the community and hope it can inspire more works towards this direction which encompass more complex scenarios or which produce tighter results.
>
> >The computational complexity for each major step of the proposed algorithm.
>
> We thank the reviewer for the suggestion to discuss the computational complexity of each subroutine within $PS\varepsilon BAI$.
>
> We consider the number of these operations: arithmetic (addition and multiplication) operations, logic operations and comparison operations and we also regard $\ln(\cdot)$, $\sqrt{\cdot}$ and sampling from a distribution as one step of operation.
>
> We decompose the main loop of Algorithm 1 as follows, where the lines with $O(1)$ operations are omitted:
> - Exploration phase (Lines 8 to 11): $O(1)$.
> - Change Detection phase (Lines 12 to 16):
>     - the $LCD$ subroutine in Line 16 needs $O(wd^2+Kd)$ operations.
> - Context Alignment phase (Lines 17 to 21):
>     - the $LCA$ subroutine in Line 19 needs $O((wd^2+Kd)N)$ operations, as the $LCD$ subroutine will be invoked $N$ times in the worst case in Line 4 of $LCA$.
>     - the reversion procedure in Line 21 needs $O(\gamma w d)$ operations.
> - The updating procedure and the stopping rule checking (Lines 25 to 32):
>     - updating $\hat{\theta}_{t,j}$ needs $O(d^2)$ operations, as we need to incorporate the latest sample into the estimate.
>     - updating $\hat{p}_{t,j}, j\in[N]$ requires $N$ operations.
>     - updating the confidence radius and the empirical best arm need $O(KN)$ operations.
>     - the stopping condition in Equation (3.4) requires $O(K)$ operations.
>
> We remark that some intermediate results can be stored to avoid repeated computations and we believe the algorithm is efficient overall. In particular, since the reward structure is linear, there are $K$ arms, and $N$ contexts, $O(d^2)$, $O(K)$, and $O(N)$ operations probably cannot be be avoided.
>
> From the above analysis, in the decreasing order of the number of operations:
> 1. the Change Detection phase requires the most budget as it will be invoked every $\gamma$ time steps.
> 2. the Context Alignment phase requires the second many operations. While it requires many operations when it is implemented, it will not be called during a stationary segment.
> 3. the updating procedure uses small portion of the operations, as $w$ is usually much greater than $K$ and $N$.
> 4. the exploration phase demands constant operations in each loop.
>
> We thank the reviewer for the suggestion again and we will incorporate these discussions into the revised manuscript.
>
> >Is $\theta_j^*$ sampled IID from $P_\theta$ each time? If so, it should be clearly stated "IID"
>
> We are sorry for if there was any confusion caused. At each changepoint, a new context is **independently** sampled from $P_\theta$. We will highlight this setup in Line 140 in the revised version.
>
> >Discussion on limitations can be improved, with designated section.
>
> We thank the reviewer for the suggestion. We will describe the limitations in a designated section in the revised version. We have some discussions in Section B on page 11. Similar to [10,21,22] in the literature, our work has introduced assumptions to provide theoretical guarantees for the algorithm, the algorithm is robust when the assumptions are violated though. We hope to weaken these assumptions via exploiting the latent clustering structure within the contexts, which can lead to future work.

---

> > ### Comment · Reviewer_comL · 2024-08-13
> >
> > Thank you very much for the detailed response. My concerns and questions are all adressed. I have no more questions.

---

### Official Review · Reviewer_5EBt · 2024-07-21

**Soundness:** 3
**Presentation:** 2
**Contribution:** 3
**Rating:** 6
**Confidence:** 3

**Summary:**

The paper proposes a new model called PSLB, where the underlying parameter $\theta$ in the common linear bandit model is sampled from an unknown distribution at fixed but unknown time steps. The authors propose a naive algorithm and a sophisticated algorithm for this problem, along with their sample complexity guarantees for best arm identification or $\epsilon$-best arm identification. A lower bound is also proposed, with a specific instance demonstrating its near-minimax optimality.

**Strengths:**

The new model makes a significant step towards making linear bandits more general, in the sense that it can model non-stationary environments, which is often true for many real-world applications. The analysis is very comprehensive from many perspectives. For example, it provides a naive algorithm whose sample complexity serves as a good comparison with the later, more complicated algorithm. The sample complexity of the more sophisticated algorithm shows an interesting dependency on the environment parameter $L_{max}$.

**Weaknesses:**

Typo: dynamics --> algorithm?

Motivating Example: I am not sure if investment is a reasonable example to motivate the model, as the model assumes fixed changing points, which clearly goes against the non-stationary market dynamics.

Presentations: Perhaps the biggest concern of mine is the presentation. The algorithm is overly tedious, and I am still not fully clear on it even after reading it multiple times.

It is not clear to me what $\tilde{O}$ hides or if it is intended to hide the same things throughout the paper.

**Questions:**

1. In most of the linear bandit best arm identification literature, for example, references 1-5, the sample complexities end up being instance-dependent quantities, which have been shown to be optimal in many cases. Is it fair to say that the sample complexity of this work is loose in the sense that it does not reveal much instance dependency?

2. It seems very critical to assume bounded noise and normalized rewards. What would happen if we assume general sub-Gaussian noise? I am considering a case where for one of the contexts, $\theta$ can result in a very high reward for certain arms, but the context has a very low probability of occurring. How would the algorithm behave in this scenario?

**Limitations:**

The work does not have any limitations due to negative societal impact. The main limitation is clarity, which has been mentioned in the weaknesses section.

---

> ### Author Rebuttal · Authors · 2024-08-05
>
> >Typo: dynamics --> algorithm?
>
> We are sorry for the confusion. There might be a cross citation problem in the manuscript which will be fixed in the revised version.
>
> The "Dynamics" refers to the dynamics of the problem. Specifically, Dynamics $1$ on page 43 describes the dynamics of the proposed problem. Dynamics 2 on page 44 and Dynamics 3 on page 49 are the dynamics of two easier problems, which are used in the proof of the lower bound result in Theorem 4.1 of Section 4.
>
> >Motivating Example
>
> Thanks for the question concerning the motivation for our problem setting. We believe our problem setting is sufficiently general to encompass a variety of practical situations. Indeed, we assume that the sequence of change points $c=(c_1, c_2,\ldots)$ is deterministic and fixed beforehand. Our results, however, do not depend on the exact sequence, but rather the lower and upper bounds on the lengths of each stationary phase, i.e., on $L_{\min}= \inf_{i \in \mathbb{N}} (c_{i+1}-c_i)$ and $L_{\max}= \sup_{i \in \mathbb{N}} (c_{i+1}-c_i)$. The investment example, or any other example, fits into this framework as long as $0<L_{\min}\le L_{\max}<\infty$. In fact, our results continue to hold even if the sequence of change points $c=(c_1, c_2,\ldots)$ is **random** as long as, almost surely, $0<L_{\min}\le L_{\max}<\infty$.
>
> Beyond specific applications and motivating examples, we believe that our model is fundamental in understanding the fundamental performance limits of best arm identification algorithms in certain controlled non-stationary environments.
>
> >Presentations
>
> We thank the reviewer for the feedback. As we wish to reveal all the details in the algorithm design, the presentation of Algorithm 1 may look complicated. We will provide a more high-level pseudocode of our algorithm in the revised version to resolve this issue (see the attached file in the global response).
>
> Based on the high-level pseudocode and Figure 1 in the manuscript, we provide a high-level explanation as follows:
> After the initialization, the algorithm design can be largely divided into 3 phases (as in Lines 177 to 179 of the manuscript):
> - **Exploration phase** (the thin blue line segments in Figure 1): the algorithm will be in the exploration phase in approximately $\frac{\gamma-1}{\gamma}$ of the total time steps. The algorithm collects samples which will be used to estimate the current latent vector $\theta_{\hat{j}\_t}^*$ and the distribution $P_\theta$ in Equation (3.2).
> - **Change Detection phase** (the diamonds in Figure 1): the algorithm enters this phase in every $\gamma$ time steps. It collects one sample each time and does changepoint detection via the $LCD$ subroutine.
> - **Context Alignment phase** (the thick brown line segment in Figure 1(b)): the algorithm enters this phase only when a change alarm is raised in the Change Detection phase. It does context alignment via the $LCA$ subroutine. The output $\hat{j}_t$ is the estimated index of the new context.
>
> Lastly, the algorithm updates the estimates and checks the stopping rule.
>
> The high-level pseudocode offers an overview of the algorithm design and we sincerely hope it can clarify the algorithm design.
>
>
> >It is not clear to me what $\tilde{O}$ hides.
>
> The $\tilde{O}$ notation hides the constants and logarithmic terms throughout the paper for easier comparison. Specifically,
> - for the upper bound of the naive algorithm in Proposition 3.1, the original non-asymptotic bound is presented in eqution (F.1) on page 22, where $T_0$ is defined in Line 650. The constants and the logarithmic term $\ln\frac{KL_{\max}}{(\Delta_{\min}+\varepsilon)^2}$ are hidden.
> - for the upper bound of $PS\varepsilon BAI$ in Theorem 3.2, the original non-asymptotic bound is presented in Equation (J.3) on page 35. The constants and logarithmic terms $\ln\frac{Kd}{(\Delta(x^*,x)+\varepsilon)^2}$, $\ln (NH_{DE}(x_\varepsilon,x))$ and $\ln\frac{KNL_{\max}}{\Delta(x^*,x)+\varepsilon}$ are hidden.
>
> >Instance-dependent sample complexity and optimality
>
> We ask the reviewer to kindly refer to the global response.
>
> >(1) Assumption on bounded noise and normalized rewards.
> (2) What would happen if we assume general sub-Gaussian noise?
> (3) Special case: high reward but low occurrence probability.
>
> We thank the reviewer for the questions.
>
> (1) The assumption on the normalized rewards (i.e., the rewards are within $[0,1]$) is common in the linear bandits literature [3] and the nonstationary bandits literature [10,14]. If the rewards are bounded in $[0,B]$, we can normalize the observed rewards $y$ by $\tilde{y}=y/B$.
>
> The assumption on the bounded noise is also common [1,14]. In our analysis, the bounded noise assumption is adopted to accommodate the randomness in the sampling rule in Line 9 of Algorithm 1, which simplifies the analysis.
>
> (2) In this case, Lemma E.1 does not hold as Bernstein's inequality has been used. In order to remedy this, one can choose the deterministic version of the G-optimal design like Figure 2 in [1], i.e., sample $x_t = \arg\min_{x\in\mathcal{X}} \max_{\tilde{x}\in\mathcal{X}} \tilde{x}^\top(A_{t-1,\hat{j_t}}+xx^\top)^{-1}\tilde{x}$ where the design matrix $A_{t,j} = \sum_{s\in\mathcal{T}_{t,j}}x_s x_s^\top$. Then Proposition 2 in [1] can replace Lemma E.1, which provides concentration guarantees for the empirical means. With some addition computational efforts, we believe we can obtain similar bounds as the current results (up to the sub-Gaussian parameter $\sigma$ factors).
>
> (3) Our algorithm indeed takes this critical scenario into account. We quantify the influence of the reward under each context and the corresponding occurrence probability of the context in $\bar{H}(x_\epsilon,x)$ in Theorem 3.2. The algorithm will stop when sufficient evidence is accumulated to infer an $\varepsilon$-best arm.
>
> We have provided more discussions and concrete examples in Lines 1453 to 1478 of the manuscript, where Case 1 is the scenario the reviewer is curious about.

---

> > ### Comment · Reviewer_5EBt · 2024-08-12
> >
> > Thanks for addressing my questions with great details. Would like to see this work being presented.

---

### Author Rebuttal · Authors · 2024-08-05

**We would like to thank all the reviewers for their valuable suggestions and we sincerely hope their concerns have been properly resolved by the detailed response provided below.**

>In most of the linear bandit best arm identification literature, for example, references 1-5, the sample complexities end up being instance-dependent quantities, which have been shown to be optimal in many cases. Is it fair to say that the sample complexity of this work is loose in the sense that it does not reveal much instance dependency?

We thank Reviewer 5EBt for the question.

Firstly, we emphasize that the sample complexity in our results (Theorem 3.2 and 3.3) are instance-dependent, in particular, the term $H_{DE}$ characterizes the difficulty in estimating the distribution of the context. Furthermore, the presence of the gaps $\Delta(x,x^*)$ also underscores that our upper bounds are functions of the instance.

Secondly, we acknowledge that there are algorithms that are instance-dependent optimal for the BAI problem in standard linear bandits literature, e.g., [24,32]. Our algorithm adopts the G-optimal design, which is minimax optimal for the BAI in standard linear bandits problem [1,2]. Although we have not proved it, we agree that the sample complexity of the proposed algorithm may not be instance-dependent optimal.

Additionally, we would like to make some remarks about the instance-dependent optimal algorithms:
- The G-optimal design is the cornerstone for the more efficient and adaptive rules like $\mathcal{X}\mathcal{Y}$-allocation and $\mathcal{X}\mathcal{Y}$-Adaptive Allocation in [1]. Therefore, our algorithm provides a framework for more sophisticated algorithms in the piecewise-stationary linear bandits problem with the BAI task. Empirically, one can attempt to replace the G-optimal design by the $\mathcal{X}\mathcal{Y}$-allocation during implementation, which can possibly yield empirical benefits.
- Currently, the lower bound is established with two simpler problems (see Section 4), which is sufficient to show our algorithm is minimax optimal. However, a tighter instance-dependent lower bound based on the original problem may be required if one wish to show an algorithm is instance-dependent optimal. This can be challenging because the distribution of the arms, the contexts and the changepoints are unknown, and the characterization of the "alternative instance" (i.e., the $\varepsilon$-best arm set is changed) is difficult.

We agree that an instance-dependent optimal algorithm is appealing and can lead to future research.

---

### Author Response · Authors · 2024-08-11

Dear reviewers,

Thank you for your assessment of our paper "Almost Minimax Optimal Best Arm Identification in Piecewise Stationary Linear Bandits". We have clarified your doubts and addressed your questions. Please let us know if our responses are satisfactory. We appreciate your time and efforts in assessing our paper and would be happy to engage in any discussions and further clarifications, if any.

Regards,

Authors of 4521

---

> ### Author Response · Authors · 2024-08-13
>
> Thanks reviewers for your acknowledgements of our rebuttals. Thanks for your positive assessment of our work.

---

### Decision · Program_Chairs · 2024-09-25

**Decision:**

Accept (poster)

**Comment:**

This paper proposes the piecewise stationary linear bandit (PSLB) model, where the weight parameter of the linear reward model changes at some unknown time step but is sampled from the same distribution. This model extends stochastic linear bandits to a nonstationary setting and is novel in the literature. The authors proposed an algorithm, PS$\epsilon$BAI+, to identify the best arm that has the highest return averaged over all context parameters. They presented an instance-dependent sample complexity analysis and showed the proposed algorithm is asymptotically optimal, validated by a lower bound on the sample complexity. Experiments are also provided to demonstrate the performance of the proposed BAI algorithm.

The reviewers unanimously voted to accept this paper due to its novelty and solid analysis, which I agree with. In the camera-ready version, the authors should address the most prominent questions brought up during the discussion, such as the discussion on computational complexity and motivating examples of the proposed model. It would also be beneficial to compare the proposed algorithm to nonstationary BAI algorithms, even though they deal with a larger class of problems, as they should still be applicable in the PSLB model.